# GeoDiv: Framework for Measuring Geographical Diversity in Text-to-Image Models

Abhipsa Basu[1*]          Mohana Singh[1*]          Shashank Agnihotri[2†]

Margret Keuper[2,3]          R. Venkatesh Babu[1]

[1]Vision and AI Lab, Indian Institute of Science, Bangalore, India

[2]Chair for Machine Learning, University of Mannheim, Mannheim, Germany

[3]Max Planck Institute for Informatics, Saarland Informatics Campus, Saarbrücken, Germany

## Abstract

Text-to-image (T2I) models are rapidly gaining popularity, yet their outputs often lack geographical diversity, reinforce stereotypes, and misrepresent regions. Given their broad reach, it is critical to rigorously evaluate how these models portray the world. Existing diversity metrics either rely on curated datasets or focus on surface-level visual similarity, limiting interpretability. We introduce GeoDiv, a framework leveraging large language and vision-language models to assess geographical diversity along two complementary axes: the Socio-Economic Visual Index (SEVI), capturing economic and condition-related cues, and the Visual Diversity Index (VDI), measuring variation in primary entities and backgrounds. Applied to images generated by models such as Stable Diffusion and FLUX.1-dev across 10 entities and 16 countries, *GeoDiv* reveals a consistent lack of diversity and identifies fine-grained attributes where models default to biased portrayals. Strikingly, depictions of countries like India, Nigeria, and Colombia are disproportionately impoverished and worn, reflecting underlying socio-economic biases. These results highlight the need for greater geographical nuance in generative models. *GeoDiv* provides the first systematic, interpretable framework for measuring such biases, marking a step toward fairer and more inclusive generative systems. Project page: `https://abhipsabasu.github.io/geodiv`

## 1 Introduction

As Text-to-Image (T2I) models gain traction in public and commercial applications, a central question arises: *whose world are they representing*? Trained on internet-scale data, these models often misrepresent regions and reinforce harmful socio-economic and regional biases (Basu et al., 2023). For instance, prompting Stable Diffusion (Rombach et al., 2022) with '`photo of a car in Africa`' often yields scenes with dusty, worn-out surroundings and damaged vehicles, overlooking the continent's visual and economic diversity. Recent studies confirm that these images frequently lack geographical diversity (Hall et al., 2023; 2024; Askari Hemmat et al., 2024). Moreover, early evidence also points to socio-economic skew (Turk, 2023): images from some countries like India overwhelmingly depict poverty or dilapidation, while others appear consistently polished or affluent (e.g., Japan). Such disparities challenge the aspiration of these models to function as faithful *world models* (Pouget et al., 2024; Astolfi et al., 2024).

With growing evidence that T2I models exhibit visual and socio-economic disparities across regions (see Figure 1), there is a need for an automated framework to capture fine-grained geo-diversity. Existing approaches, whether based on narrowly curated datasets (Hall et al., 2024; Ramaswamy et al., 2023; Gaviria Rojas et al., 2022) or low-level visual dissimilarity metrics (Friedman & Dieng, 2023), struggle to reveal such deeper, country-specific patterns. Although recent works use Large Language Models (LLMs) and Vision-Language Models (VLMs) to assess *realism* (Li et al., 2025), *prompt consistency* (Hu et al., 2023; Cho et al., 2023), or *concept diversity* (Rassin et al., 2024; Teotia et al., 2025), these formulations remain insufficient for geo-diversity, which spans economic, environmental,

---

*Equal contribution

†Work done while interning at the Vision and AI Lab, Indian Institute of Science, Bangalore

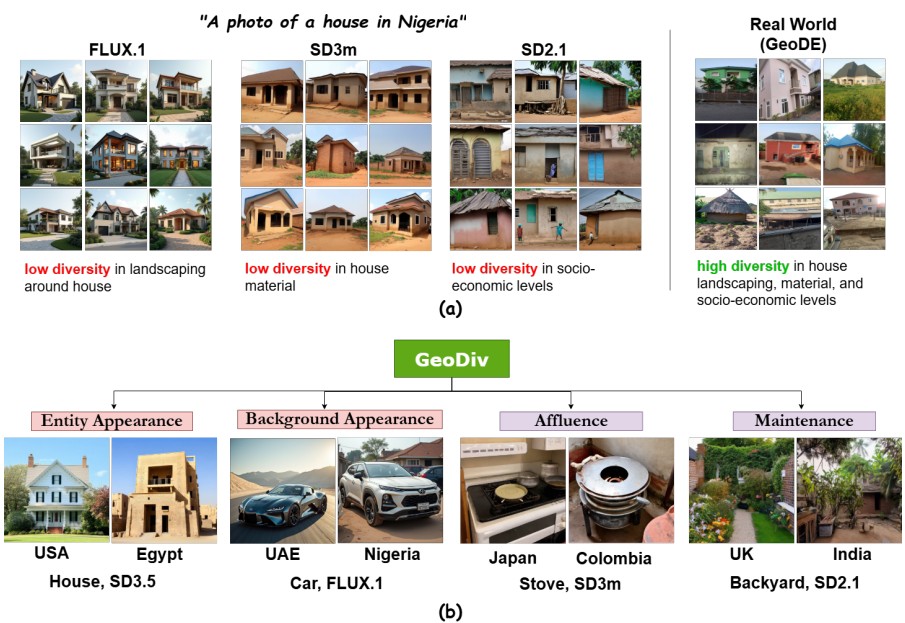

Figure 1: **Lack of Geographical Diversity observed in T2I Generations and the Need for GeoDiv.** (a) Text-to-image models produce systematically low visual diversity for the same prompt across countries (example: 'a photo of a house in Nigeria'), failing to reflect the rich variation seen in real-world images (Ramaswamy et al., 2023). (b) *GeoDiv* provides an automated, reference-free framework that can quantify such fine-grained geographical differences by evaluating images along four interpretable axes: Entity-Appearance (sloped/flat roof), Background-Appearance (paved/unpaved road), Affluence (luxury/modest settings), and Maintenance (manicured/unkempt). Examples show how the same entity type varies dramatically across countries and generative models.

and contextual variation. A single diversity metric cannot capture such multidimensional aspects, limiting interpretability and masking region-specific biases.

In this work, we propose *GeoDiv*, a framework for quantifying geo-diversity along two complementary axes. The **Socio-Economic Visual Index (SEVI)** captures socio-economic cues through two interpretable dimensions: (a) *Affluence*, ranging from impoverished to affluent depictions, and (b) *Maintenance*, measuring physical condition from worn to pristine. Both are rated on a 1–5 scale using VLM judgments and are closely tied to societal well-being (Awaworyi Churchill et al., 2025). The **Visual Diversity Index (VDI)** measures variation in (a) *Entity Appearance*, reflecting attributes such as shape, material, or color of the primary entity, and (b) *Background Appearance*, capturing contextual variability (e.g., type of roads visible). Fig. 1 illustrates how these dimensions differ across geographies and generative models. VDI employs LLMs to extract entity and background attributes, while VLMs aid in estimating their distributions with respect to images across countries. For each SEVI and VDI dimension, diversity is quantified using the interpretable *Hill Number*, defined as the exponential of the entropy of attribute value distributions (Leinster, 2021). While geo-diversity also encompasses cultural, historical, and aesthetic dimensions that remain difficult to measure at scale, *GeoDiv* is modular and can incorporate new axes as methods advance. Since our approach relies on the implicit world knowledge embedded in LLMs and VLMs, we validate both SEVI and VDI extensively through human studies.

Applied to $160,000$ images generated by Stable Diffusion v2.1 (SD2.1), v3 (SD3m), v3.5 (SD3.5) Rombach et al. (2022), and FLUX.1-dev (black-forest-labs, 2024), across 10 common entities (e.g., house, car, etc) and 16 countries, *GeoDiv* reveals several key insights. Images from countries like India, Nigeria, and Colombia are consistently found to be impoverished and worn out than those from USA, UK, or Japan, highlighting systemic socio-economic bias. Interesting country-level biases are also observed in case of Entity and Background appearance. For instance, SD3.5 shows $99\%$ Egyptian houses to be made of stones, while $88\%$ UK houses to be built of bricks. Across models, backgrounds of $77\%$ of car images from Nigeria show dirt/gravel road, compared

to US which generates paved roads $85\%$ of the time. Interestingly, FLUX.1 images score highly on SEVI but low on VDI, suggesting a trade-off between image polish and diversity. Thus, *GeoDiv* captures nuanced geographical biases and gaps in generative models, providing a systematic and interpretable framework for auditing geographical representation. Our key contributions are:

- We introduce *GeoDiv*, an interpretable evaluation framework for measuring geo-diversity in generative models along two complementary axes: **Socio-Economic Visual Index (SEVI)** and **Visual Diversity Index (VDI)**, quantifying socio-economic and visual diversity by leveraging the world knowledge of large language models (LLMs) and vision-language models (VLMs).
- We obtain and release structured attribute-value sets using LLMs, for evaluating the geo-diversity of 10 common entities (e.g., house) across both SEVI and VDI. We also provide the full prompts and filtering mechanisms needed to generate comparable evaluations for new entities.
- We curate a dataset of 160,000 synthetic images generated with four open-source diffusion models, covering 16 countries and 10 entities. For a subset, we collect VDI attribute annotations and country-level SEVI ratings from crowdworkers via crowdsourcing platforms. These human-annotated datasets are then used to evaluate multiple LLM-VLM combinations for implementing *GeoDiv*. All annotations and the codebase are released to support benchmarking of future models.
- *GeoDiv* uncovers regional biases and key limitations in current generative models, demonstrating its utility as an effective and interpretable diagnostic tool for assessing geographical diversity, compared to existing diversity measurement baselines. We release diversity scores across all *GeoDiv* dimensions for the curated synthetic dataset, enabling practitioners to systematically improve the geo-diversity of diffusion models.

## 2 RELATED WORK

**Metrics Measuring Image Diversity**: Image diversity metrics are typically categorized into two types. The first compares a given image set to a reference set, e.g., FID (Heusel et al., 2017), which compares feature distributions using a pre-trained Inception network (Szegedy et al., 2017). We exclude such metrics due to the absence of large-scale geo-diverse reference datasets (Gaviria Rojas et al., 2022; Ramaswamy et al., 2023). The second type assesses variation within the given set. Pairwise Distance Metrics (Fan et al., 2024; Boutin et al., 2023) compute average distances between image embeddings (e.g., Inception or CLIP (Radford et al., 2021)), while Vendi-Score (Friedman & Dieng, 2023) measures entropy over the eigenvalues of the feature kernel matrix. However, these approaches capture only visual variation. Because of their uninterpretable nature, the extent to which such metrics can capture the nuances of geo-diversity is unclear. On the contrary, our proposed framework *GeoDiv* measures the multiple dimensions of geo-diversity in an interpretable manner.

**Leveraging the World Knowledge of Large-Scale Models**: Trained on internet-scale data, LLMs and VLMs encode rich knowledge about global cultures and demographics, which many recent works have utilized to measure stereotypes, consistency, realism and diversity in images. OASIS (Dehdashtian et al., 2025) quantifies stereotypes in text-to-image generation by comparing real-world attribute distributions for different nationalities with those inferred from generated images via a VQA model. TIFA (Hu et al., 2023) and DSG (Cho et al., 2023) evaluate image-prompt consistency by generating questions from the LLM and finding corresponding answers for each image through a VLM, where the latter adopts a Davidsonian Scene Graph to avoid hallucinations, duplications, and omissions in the generated questions. REAL (Li et al., 2025) employs a VQA model to measure the realism of images from text-to-image models. The LLM-VLM paradigm has also been used by a few prior works to identify and measure biases in a given set of images (Chinchure et al., 2024; Mandal et al., 2024), whereas Basu et al. (2025) utilize diffusion models to augment existing biased training sets (Basu et al., 2024). GRADE (Rassin et al., 2024) is the first method that employs the LLM-VLM paradigm to assess visual diversity in everyday objects. However, geo-diversity being more complex, we first segregate it into multiple axes, and then propose metrics to measure each of them by leveraging the LLM-VLM approach in different ways.

**Geographical Biases in Text-to-Image Models**: Over the recent years, multiple works have uncovered harmful geographical biases in real and synthetic datasets. Such studies can be divided into two broad categories. The first category investigates the representation of countries within both real image datasets (De Vries et al., 2019; Shankar et al., 2017; Naggita et al., 2023; Wang et al., 2022; Faisal et al., 2022) and synthetic ones (Basu et al., 2023). The second category studies the the extent of

variations within a country in the images (Hall et al., 2023; 2024; Askari Hemmat et al., 2024), which show that existing metrics fail to capture geographical variations within a country. While our paper focuses on the second category, most of the previous works rely on existing geo-diverse datasets like GeoDE (Ramaswamy et al., 2023) to measure geo-diversity and similar aspects, constraining such metrics to concepts and countries covered in those datasets. Our paper attempts to mitigate this limitation, and introduces a framework that measures geo-diversity in a reference-independent and interpretable manner, extendable to any number of entities and countries.

## 3 PROPOSED FRAMEWORK: GEODIV

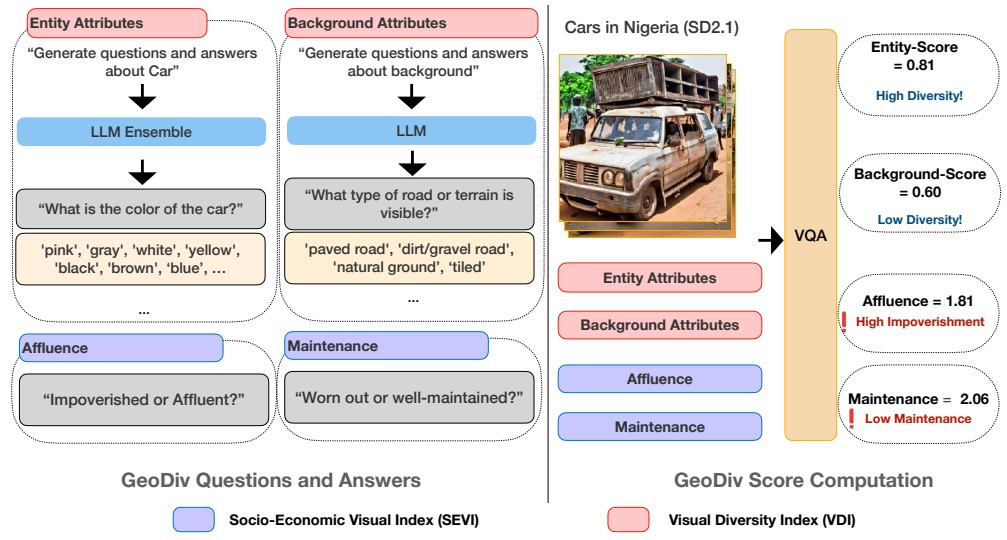

Figure 2: **GeoDiv Pipeline.** Given an entity $e$ and country $c$, LLMs generate attribute-based questions specific to $e$, and a fixed set of background-related questions applicable across entities. A VQA model predicts answer distributions over an image set for both question types, from which *GeoDiv* computes the Visual Diversity Index (VDI) via normalized Hill number. The VQA model also rates each image on Affluence and Maintenance to compute the Socio-Economic Visual Index (SEVI).

Motivated by clear human-identified disparities in how T2I models depict different regions, we aim to develop a principled method to quantify such geographical variation. We introduce *GeoDiv*, a systematic and interpretable framework for measuring the geo-diversity of images generated for a given entity and country. Given a collection of images $\mathcal{D}$, we extract a subset $\mathcal{D}_e^c$ corresponding to entity $e \in \mathcal{E}$ and country $c \in \mathcal{C}$. These images are synthetically generated using text-to-image models with prompts of the form 'a photo of a $\{e\}$ in $\{c\}$'. In this section, we first introduce the two core axes along which GeoDiv assesses geo-diversity, and then describe how it is quantitatively computed for each dimension.

### 3.1 VISUAL DIVERSITY INDEX (VDI)

To assess the visual variation of images across geographies, we define the **Visual Diversity Index (VDI)** along two axes: **Entity-Appearance** and **Background-Appearance**.

**Entity-Appearance** examines the visual attributes of entities (e.g., houses, cars) within a country. Manually defining a comprehensive set of attributes for each entity is infeasible, so we leverage multiple LLMs to generate candidate question-answer (Q&A) sets, and consolidate them into a unified list using an aggregator LLM. The same Q&A sets are applied across countries for comparability. Finally, a VQA model answers these questions for each image in the set $\mathcal{D}_e^c$, and the resulting distribution of answers across the images is used to compute per-question entity diversity.

**Background-Appearance** assesses the scene context (e.g., presence of modern infrastructure, type of roads, etc). We divide background into indoor and outdoor categories. An LLM first generates a

fixed set of contextual questions and answer choices for each category (an example outdoor-category question: '*What type of road or terrain is visible?*'). Each image is first classified by a VQA model as indoor or outdoor. Based on the prediction, category-specific questions and answers are input to the VQA model. The resulting answer distributions are then utilized to calculate background diversity.

## 3.2 SOCIO-ECONOMIC VISUAL INDEX (SEVI)

To capture economic status and visual cues of physical upkeep across geographies, we introduce the **Socio-Economic Visual Index (SEVI)** with two dimensions: **Affluence** and **Maintenance**. An attentive reader may enquire about the difference between the two. Affluence reflects the overall wealth depicted in an image, while Maintenance evaluates the physical condition of the primary entity, both crucial to understand societal well-being (Awaworyi Churchill et al., 2025). For each image, a Vision-Language Model (VLM) predicts scores for these dimensions on a 1-5 scale:

> **Affluence (1–5):** Impoverished → Low → Moderate → High → Luxury.
>
> **Maintenance (1–5):** Severely Damaged → Poor → Moderate → Well-Maintained → Excellent.

The VLM is prompted with detailed descriptions of these scales and scores each image individually to provide interpretable socio-economic visual signals. Finally, the distribution of the Affluence and Maintenance scores for an image set $\mathcal{D}_e^c$ is studied to assess socio-economic diversity.

*GeoDiv* integrates both SEVI and VDI dimensions for a comprehensive diversity assessment. All questions, and answers used are included in Appendix § H and prompts are included in the github repo (`https://github.com/moha23/geodiv`).

## 3.3 DIVERSITY COMPUTATION

Using the distributions obtained from the VDI and SEVI questions, we quantify the uniformity of answer distributions by computing the *Hill Number*. This is a biodiversity-inspired metric that represents the effective number of distinct categories (or "species") in a community and is calculated by exponentiating Shannon's entropy, which captures the uniformity of the distribution. Consider a question $q_k$ (related to either SEVI or VDI attributes), having a set of answers denoted by $\mathcal{A}_k$. Given that the values of an attribute can be too large to enumerate exhaustively, we generate an approximate set of answers per question by leveraging the world knowledge of the LLMs, denoting the same as $\hat{\mathcal{A}}_k$ (see our codebase for prompt details). Hill numbers represent the "*effective number of answers*" represented in the distribution and range from 1 (when a single answer class is over-represented, yielding zero entropy) to $|\hat{\mathcal{A}}_k|$ (when all provided answers are equally well-represented, yielding maximum diversity). Since the number of plausible answers can vary across different questions, we compute a *Normalized Hill Number* (ranging between 0 and 1) to enable fair comparison between questions with varying answer-set sizes, as defined below:

$$\text{Diversity-Score} = \frac{\exp(H(\hat{P}_k)) - 1}{|\hat{\mathcal{A}}_k| - 1} \tag{1}$$

where $\hat{P}_k$ is the answer distribution for $q_k$, and $H(\cdot)$ denotes Shannon entropy. Diversity for **Affluence** and **Maintenance** are computed directly using Diversity-Score. The **Entity-Appearance** and **Background-Appearance** Diversity are calculated by averaging Diversity-Score over all related questions for the individual dimensions.

**On Computing Socio-Economic Diversity.** When evaluating socio-economic diversity in synthetic images, a key question arises: *should the ideal scenario emphasize affluence and high physical upkeep, or represent the full spectrum of socio-economic conditions?* We adopt the latter to promote inclusivity, additionally reporting the mean Affluence and Maintenance ratings (on a 1–5 scale, subsection 3.2) per country or dataset. This reveals systematic biases, with models disproportionately generating affluent or impoverished images depending on the country prompted.

Table 1: **Performance of various VQA models in identifying VDI answers and SEVI scores compared against human annotations**. `Gemini-2.5-flash` achieves the highest accuracy on entity and background questions, as well as the strongest correlation with human ratings on the SEVI metrics. `Qwen2.5-VL` is competitive, while `LLaVA` underperforms substantially.

| Models | VDI Answers (Accuracy) | | | SEVI Scores (Spearman's $\rho$) | |
|---|---|---|---|---|---|
| | Entity | Background | Overall | Affluence | Maintenance |
| `Gemini-2.5-flash` | 0.87 | 0.85 | 0.86 | 0.76 | 0.69 |
| `gpt-4o` | 0.85 | 0.81 | 0.83 | 0.76 | 0.76 |
| `Qwen2.5-VL` | 0.85 | 0.77 | 0.81 | 0.69 | 0.71 |
| `llava-v1.6-mistral-7b-hf` | 0.70 | 0.66 | 0.68 | 0.65 | 0.68 |

## 4 EXPERIMENTAL SETUP AND VALIDATION FOR GEODIV

### 4.1 DATASET DETAILS

**Entities**. We evaluate geo-diversity of images belonging to 10 entities commonly studied in prior works (Hall et al., 2024), as well as represented in well-known geo-diverse datasets (Ramaswamy et al., 2023): *backyard*, *bag*, *car*, *chair*, *cooking pot*, *dog*, *house*, *plate of food*, *shopfront* and *stove*.

**Countries**. Our analysis spans 16 countries across diverse regions: the United States (USA), Mexico, Colombia, the United Kingdom (UK), Italy, Spain, Japan, South Korea, Indonesia, China, India, the UAE, Turkey, Philippines, Egypt, and Nigeria.

**Generative Models**. We measure the geo-diversity of images generated by models such as SDv2.1, v3m, v3.5 (Rombach et al., 2022), and FLUX.1-dev (black-forest-labs, 2024). For each entity-country pair, we generate 250 images per model, resulting in $40,000$ images per model. Thus, our synthetic dataset comprises of $160,000$ images overall. Further dataset details can be found in Appendix § M, and samples can be observed in Appendix Fig. 23, 24, 25 and 26.

### 4.2 VALIDATING GEODIV COMPONENTS

**VQA Accuracy for Entity and Background Diversity.** The VDI dimensions depend on the VQA model's ability to correctly recognize visual attributes. We evaluate this by sampling 12 images per entity (randomly chosen from the four T2I models), each paired with one entity- and one background-based question, yielding 240 image-question pairs. Each pair is annotated by three (Prolific, 2024) crowd-workers using LLM-generated answer choices, with majority voting for the final label. The questions are deliberately generic, requiring minimal region-specific knowledge to avoid bias. Table 1 reports the accuracy of the VQA model's predictions when compared against human annotations during the validation study. Among the four VLMs tested, `gemini-2.5-flash` performs best with $86\%$ overall accuracy ($87\%$ for entity, $85\%$ for background), while `Qwen2.5-VL` and `gpt-4o` achieve comparable results but slightly lag on background questions.

**Validating the SEVI Metrics.** The Affluence and Maintenance dimensions of SEVI capture nuanced aspects of wealth and physical condition. To evaluate alignment with human judgment, we conduct a country-wise study: for each country, 4 images per concept (40 total) are sampled across all T2I models, yielding 80 image-question pairs. Owing to participant unavailability, Nigeria and Turkey are excluded. Native annotators (via (Prolific, 2024)) rate each image on the SEVI scale, with three ratings per image, producing 1120 ratings overall. On this benchmark, `Gemini-2.5-flash` achieves high Spearman correlations with human scores ($\rho = 0.76$ for Affluence, $\rho = 0.69$ for Maintenance), with similar performances by the other models. Overall, the open-source `Qwen2.5-VL` can be seamlessly used for implementing *GeoDiv* (see Appendix Section I.3), though we adopt `Gemini-2.5-flash` for its slightly superior performance on VDI.

Further details on the human studies (remuneration, instructions, etc), country-wise correlation coefficients for the SEVI dimensions, and a robustness analysis of the metric across all axes are shared in the Appendix § I.

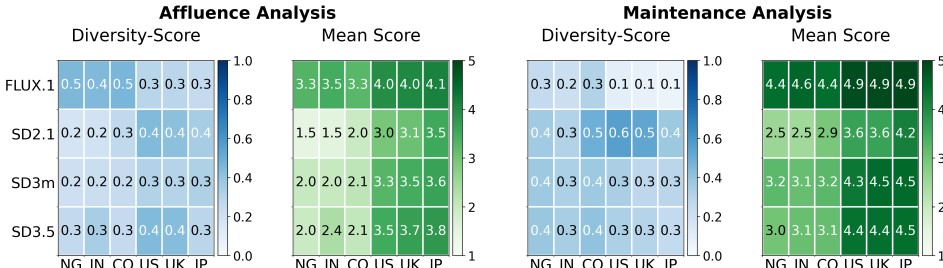

Figure 3: **SEVI Diversity and Mean Ratings across Datasets and Countries**. India (IN), Nigeria (NG), and Colombia (CO) are seen to receive lower SEVI ratings, while the US, UK, and Japan (JP) rank highest—revealing strong socio-economic biases in country-level image representations. Strikingly, none of the models generate images spanning *diverse* socio-economic strata.

### 4.3 IMPLEMENTATION STEPS

We use the `Gemini-2.5-flash` model for all experiments due to its superior performance (§4.2). The hyperparameter details are provided in Appendix A.4. SEVI scores are obtained by directly prompting the VLM to rate images on Affluence and Maintenance. The *VDI analysis* involves several steps, detailed below:

**Question and Answer Generation.** For Entity-Appearance, diverse attribute-related questions are generated by an ensemble of five LLMs, and consolidated using a separate aggregator LLM (see Appendix A.3 for full model versions). This ensures comprehensive attribute coverage for entities whose characteristics may vary widely. In contrast, background questions (e.g., crowded vs. quiet) are generally applicable across scenes and do not require per-entity customization. Therefore, a fixed set of background questions is generated using `Gemini`. Answers for all questions are obtained from `Gemini` and further cleaned by the same to remove redundant or problematic responses (see our codebase for prompts and § H for the resulting questions and answers).

To reduce the effects of the intrinsic biases of the VLM, we perform the following *control steps*:

**Visibility Step for Undetectable Attributes.** After generating question–answer pairs for background and entities, the VLM filters out images where the questioned attribute is not visually detectable (Cho et al., 2023) to reduce hallucinations in the VQA step (Appendix B.1 shows the rejection percentages).

**Multi-Select Responses.** This allows selecting multiple valid answers and avoids distortions from forced single-choice formats.

**None Of The Above (NOTA).** To account for any missing answer in those generated by the LLM, we append a special NOTA option before querying the VQA model. Only $2.6\%$ image-question pairs obtained NOTA as the answer. This lets the model abstain when no option fits, reducing hallucinations due to forced *guessing* instead of acknowledging *uncertainty* (Kalai et al., 2025). See Appendix B.2 for finer-grained analysis.

## 5 WHAT DOES GEODIV REVEAL ABOUT GEO-DIVERSITY?

The *GeoDiv* framework is applied to images from four T2I models, spanning 10 entities and 16 countries (see Section 4). Overall SEVI and VDI trends are shown in Figures 3 and 4, with detailed analyses across **datasets** (§ 5.1) and **countries** (§ 5.2) below.

### 5.1 DIVERSITY COMPARISON ACROSS DATASETS

**FLUX.1 Images Appear the Richest, Yet No Dataset Offers Balanced SEVI Coverage.** The average Affluence Diversity-Score is similar across the T2I models ($0.35 \pm 0.01$). While the average Maintenance Diversity-Score is $0.34 \pm 0.12$, FLUX.1 images show a severe lack of variation in the physical conditions of the entities depicted, with a low score of $0.15$. This indicates that no model provides balanced coverage across all socio-economic strata. FLUX.1 tends to generate polished,

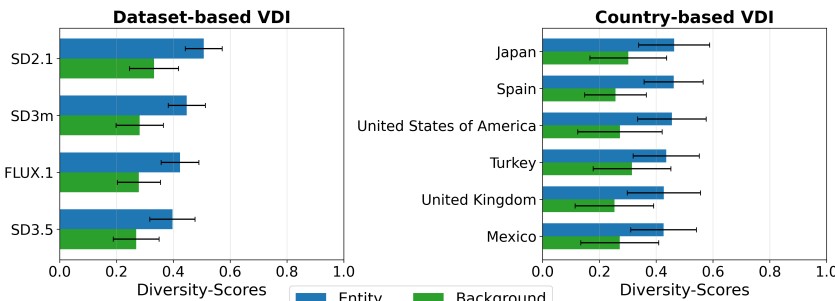

Figure 4: **VDI Scores across (a) Datasets, (b) Countries**. Model-wise VDI diversities are similar, with SD2.1 achieving higher scores than the others. Mexico and the UK show low entity and background diversity, while Japan scores highest.

aesthetically pleasing images, achieving mean Affluence and Maintenance ratings of 3.82 and 4.73 respectively (on the $1-5$ scale defined in Section 3). In contrast, the remaining models show similar, lower scores, with SD2.1 scoring the lowest: mean Affluence of 2.41 and Maintenance of 3.23. These observations are demonstrated for a selected group of countries in Fig. 3. Aggregating across all entities and countries, Affluence and Maintenance show a moderate positive correlation ($\rho = 0.5$): more affluent items tend to appear better maintained. Yet this pattern varies by entity, sometimes even reversing. *GeoDiv* highlights such cases; for instance, a Nigerian clay pot on muddy ground scores low on affluence (score: 1) but high on maintenance (score: 4), while an Egyptian luxury sports car scores high on affluence (score: 5) but low on maintenance (score: 2) due to visible dust on the hood.

**Synthetic Images Lack Visual Diversity.** The Entity-Appearance Diversity-Score is highest for SD2.1 (0.51), followed by SD3m (0.45), FLUX.1 (0.42), and SDv3.5 (0.40) (see Fig. 4). While these scores indicate a general lack of diversity in entity appearances, the issue is more pronounced for background appearance, where all datasets score low (0.31 on average). Overall, the limited variation in both dimensions highlights a clear opportunity for improvement by data curators and model developers. In particular, FLUX.1 exhibits very low VDI diversity while achieving the highest SEVI ratings, suggesting it produces consistently polished, yet overly similar-looking images.

**Overall Geo-Diversity Tends to Decrease in Newer Diffusion Model Versions.** Averaged across SEVI and VDI, FLUX.1 shows the lowest scores, while SD2.1 ranks highest among T2I models, consistent with prior findings (Rassin et al., 2024; Hall et al., 2023) (Appendix Table 6). Though differences are modest, they underscore the need to improve both visual and socio-economic diversity in synthetic image generation and demonstrate *GeoDiv*'s utility in assessing geo-diversity.

## 5.2 COUNTRY-BASED GEO-DIVERSITY

**India, Nigeria, and Colombia Portrayed as Poorest; Japan, UAE, and UK as Wealthiest.** Across datasets, the mean Affluence and Maintenance diversity scores per country are low, 0.36 and 0.38, highlighting a severe lack of socio-economic inclusivity, with India and Japan exhibiting the least diversity. Strong biases emerge (see Fig. 3): India, Nigeria, and Colombia are consistently portrayed as the poorest (average Affluence: 2.31, Maintenance: 3.34), while Japan, UAE, and UK appear as the wealthiest (Affluence: 3.53, Maintenance: 4.30). This trend is less apparent in FLUX.1 images, as it generates polished images uniformly. These results expose a pronounced socio-economic bias in synthetic image generation, entrenching narrow and stereotypical socio-economic portrayals.

**Entity-Appearance Diversity Low Across Countries; But Are The Distributions Similar?** The mean Diversity-Score across countries is only 0.47, indicating limited variation in entity attributes and exposing both global and country-specific biases (see Fig. 4). For example, models consistently fail to generate *chairs* without backrests irrespective of countries. On the other hand, country-specific biases emerge, for example, SD3m shows very few cushioned chairs for Nigeria and the Philippines, whereas the UK and USA samples rarely depict hard-seated chairs (see Appendix D.2 for more examples). Beyond absolute diversity, we compute Jensen-Shannon Distance (JSD) to capture distributional differences between countries. Fig. 5 reports the maximum JSD averaged across questions for each model and entity, showing sharp divergences in some cases. For instance, Egyptian houses generated

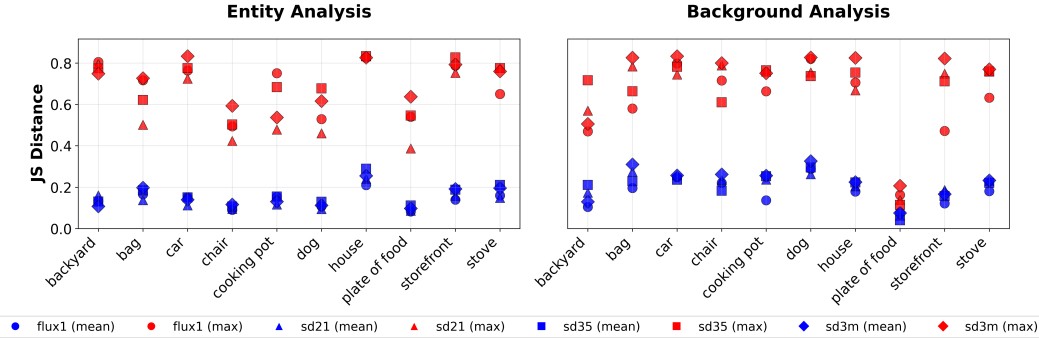

Figure 5: **Country-wise maximum and mean JS Divergence across Entities and Models**. High maximum values for both Entity and Background indicate high cross-country variations in the respective attribute value distributions.

by SD3.5 differ markedly from others (Appendix Fig. 9), caused by distinct exterior materials and adjoining ground cover. Thus, *GeoDiv* reveals both global biases and substantial cross-country variation, while offering a framework readily extendable to new entities, countries, and models. To enhance interpretability, we release full per-question distributions in the supplementary, enabling practitioners to prompt underrepresented attributes explicitly (see Appendix § G for an example).

**Background Diversity Strikingly Lower than Entity Diversity.** The average Background Diversity-Score across countries is 0.33, significantly lower than that of Entity-Appearance, indicating severe lack of variations in the generated backgrounds. Irrespective of countries and models, most backgrounds tend to be quiet and empty, without significant crowd presence, even in case of entities like cars and houses. Similarly, mountains and hills are depicted only 12% times on average across countries–least depicted in Nigeria (1.1%), and most depicted in Turkey (24%), indicating underrepresentation of a crucial natural feature. Waterbodies are depicted even lesser, in only 3.4% images. We plot the maximum JSD averaged across questions for each model and entity across distributions of country-pairs in Fig. 5. The high values are caused by cross-country variations: for instance, across models, backgrounds of 77% of car images from Nigeria show dirt/gravel road, compared to US which generates paved roads 85% of the time. Similarly, 57% of Indian images show dense buildings in the background, compared to only 17% for the UAE.

**Egypt Most Geo-Diverse Country, India The Least.** Averaging across all four *GeoDiv* scores, we find Egypt, Colombia, Turkey, and Spain to be among the most geo-diverse countries, whereas Japan, the UK, the US, and India rank among the least. The mean *GeoDiv* score per country is 0.39, a predictably low value that underscores the need to improve the diversity of generative models across all analyzed dimensions. Country-level scores are reported in Appendix Table 5. Interestingly, we also observe a weak negative correlation between *GeoDiv* scores and both GDP nominal and per capita ($\rho = -0.27$ and $-0.28$, respectively), suggesting that generative models tend to produce less diverse imagery for wealthier countries.

Detailed visualizations of the SEVI and VDI scores across models, entities and countries, along with crucial examples of observed biases are presented in Appendix § E, § D and § J respectively. The variation in SEVI and VDI scores per entity is further discussed in Appendix C.1.

## 6 DISCUSSION

**Comparison with Existing Baselines.** Vendi-Score (Friedman & Dieng, 2023) measures visual diversity within image sets, but overlooks key aspects of geo-diversity that *GeoDiv* measures. For example, *GeoDiv*'s SEVI axis on Affluence and Maintenance reflects socio-economic context that Vendi-Score cannot detect. To assess the relationship between Vendi-Score and *GeoDiv* (combined across all axes), we compute their correlations. Only Entity Diversity has a high correlation (Pearson's $\rho = 0.56$) while the others are lower (e.g., $\rho = 0.06$ for maintenance). This shows that although entity specific diversity can be measured by Vendi score, it lags behind in multidimensional diversity computations. Detailed results are in Appendix Table 15. We discuss another method DIMCIM (Teotia et al., 2025) in Appendix K.

**Geo-Diversity of a Real-World Dataset.** To benchmark synthetic images against a geographically representative real-world dataset, we evaluate GeoDE (Ramaswamy et al., 2023) using *GeoDiv*. Clear differences emerge: GeoDE achieves substantially higher Entity-Appearance Diversity (0.60 vs. 0.44 for synthetic images). Background-Appearance Diversity is closer but still higher in GeoDE (0.42 vs. 0.31). On the SEVI axis, GeoDE exhibits markedly greater diversity in Maintenance (0.61), while its Affluence diversity, though the highest among all datasets, remains comparable to others. These findings highlight GeoDE is consistently more geo-diverse than synthetic datasets, particularly in Entity-Appearance and Maintenance, likely because it is crowd-collected, and thus, it is expected to reflect real-world variations better. Detailed entity- and country-level scores are provided in Appendix Fig. 13 and § F.

## 7 CHALLENGES AND LIMITATIONS

We analyze the geo-diversity of four T2I models across 16 countries and 10 entities, but extending this evaluation to a broader set of regions and entity types may uncover additional patterns and biases. To support such extensions, we publicly release the question and answer distributions for every country–entity–model combination used in this study. We also provide all prompts in our codebase, enabling researchers to easily adapt our framework to new entities, countries, and generative models.

For the VDI axis, the questions and their corresponding answer sets are generated using the world knowledge of LLMs, since exhaustively enumerating all possible entity or background attributes and their values is infeasible. For the SEVI axis, we explicitly define the levels of affluence and maintenance due to the absence of any established or standardized scales for these socio-economic cues. Furthermore, our diversity score estimates also rely on LLMs and VLMs, which may carry inherent biases. To mitigate this, we restrict questions to generic entity and background attributes, avoiding region-specific knowledge. The goal is to reveal how model generations vary even on basic attribute distributions across entities and countries. Large-scale human studies (including country-wise studies for the SEVI metrics) reinforce *GeoDiv*'s reliability, while the visibility and NOTA checks further reduce hallucinations.

An important aspect of geo-diversity is cultural representation; whether generative models capture local cultural contexts or default to globalized visuals. We quantify this using a Cultural Localization score via our VQA-based pipeline, analogous to the Affluence and Maintenance scores. We observe higher disagreement between the VQA model and human annotators for countries like the USA and UK, while Japan and Colombia show better alignment, reflecting regional variations in model–human agreement. Full results are in Appendix § L.

Another limitation is reliance on `Gemini-2.5-Flash`, a closed-source model; despite strong quality and alignment with human judgments, budget constraints limit large-scale evaluations across entities and countries. As noted in Section 4.2, open-source `Qwen2.5-VL` is a practical alternative, showing high agreement with `Gemini` on all four diversity axes (average correlation $\rho = 0.83$) across two datasets and six entities (Appendix I.3). Continued progress in open-source VLMs will enable broader, richer, and more cost-effective assessments of global diversity.

## 8 CONCLUSION

In this work, we introduced *GeoDiv*, a multidimensional framework that leverages the world knowledge of LLMs and VLMs to quantify geographical diversity in image datasets. To capture disparities in socio-economic status, physical upkeep, and variations in entities (e.g., houses, cars) and their contexts, we proposed two axes: (a) the **Socio-Economic Visual Index (SEVI)**, which uses a VLM to assess affluence and maintenance, and (b) the **Visual Diversity Index (VDI)**, which evaluates entity and background diversity with LLM-VLM guidance. Applying *GeoDiv* to images from four T2I models across 16 countries and 10 entities, we found systematic gaps: diversity in entities and backgrounds declines in newer models, while SEVI scores consistently mark India, Nigeria, and Colombia as impoverished and poorly maintained. By contrast, FLUX.1 generates more affluent depictions but with low visual diversity, revealing a trade-off between sophistication and inclusivity. *GeoDiv* provides a first step toward interpretable audits of T2I geographical inclusivity with minimal human oversight, and we hope it inspires efforts to build generative systems that are not only visually appealing but also globally representative.

ACKNOWLEDGEMENTS

This work is supported by a grant from Google Research and the Kotak IISc AI-ML Centre (KIAC). The authors are grateful to Soumya Dutta (LEAP Lab, IISc) for their valuable feedback. Shashank Agnihotri and Margret Keuper acknowledge support by the DFG Research Unit 5336 - Learning to Sense (L2S). The authors further acknowledge support by the state of Baden-Württemberg through bwHPC and the German Research Foundation (DFG) through grant INST 35/1597-1 FUGG. The authors are grateful for the computing time provided on the high-performance computer HoreKa by the National High-Performance Computing Center at KIT (NHR@KIT). This center is jointly supported by the Federal Ministry of Education and Research and the Ministry of Science, Research, and the Arts of Baden-Württemberg, as part of the National High-Performance Computing (NHR) joint funding program (https://www.nhr-verein.de/en/our-partners). HoreKa is partly funded by the German Research Foundation (DFG).

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

# GeoDiv: A Multidimensional Framework for Measuring Geographical Diversity in Images

## Supplementary Material

*GeoDiv*, introduced in the main paper, is a framework for assessing dataset geo-diversity across multiple dimensions. This supplementary material provides extended details in support of the main results. The following sections outline the details and additional analyses.

## A   Implementation Details

### A.1   Implementation Details For Text-to-Image Generative Models

All synthetic datasets were generated using publicly available models from the Hugging Face Hub. Default generation settings provided by the respective model repositories were used unless otherwise specified. Image generation was performed via the `diffusers` library, using standard inference pipelines. Prompts were constructed per entity-country pair using the template: "`A photo of a/an <entity> in <country>`". Each model was queried to generate 250 images per entity-country pair, totaling 40,000 images per model.

The models used are:

- **Stable Diffusion 2.1** (SD2.1)[1]
- **Stable Diffusion 3** (SD3m)[2]
- **Stable Diffusion 3.5** (SD3.5)[3]
- **FLUX.1-dev** (FLUX.1)[4]

SD2.1 images were generated at a resolution of $768 \times 768$, while SD3m, SD3.5, and FLUX.1 used $1024 \times 1024$. For reproducibility, generation was performed with fixed seeds for each batch. No further post-processing was applied to the generated images.

### A.2   Compute Resources

Image generation experiments were conducted on an NVIDIA RTX 6000 GPU (48GB VRAM). For all LLM and VLM-based tasks, including question-answer generation and VQA, we use `Gemini-2.5-Flash` (Google, 2024) accessed via the Vertex AI API (Cloud, 2024) with dynamic thinking enabled for optimal token efficiency as well as batch processing for cost and time

---

[1]`https://huggingface.co/stabilityai/stable-diffusion-2-1`
[2]`https://huggingface.co/stabilityai/stable-diffusion-3-medium`
[3]`https://huggingface.co/stabilityai/stable-diffusion-3.5-large`
[4]`https://huggingface.co/black-forest-labs/FLUX.1-dev`

efficiency. The estimated cost for computing the VDI component of our diversity score, including visibility checks and VQA for both entity and background analysis, is approximately $58.64 per entity-country-question combination (across 250 images per set). The SEVI score computation for the same combination costs approximately $9.46, resulting in a total cost of $68.10 per complete diversity assessment. On the other hand, experiments using Qwen2.5-VL-32B-Instruct-AWQ, performed locally using an NVIDIA RTX A5000 (24GB VRAM), incur no additional computational costs but require significantly longer processing times.

### A.3 LLMs used for Entity-Appearance Attribute-Value Generations

As each entity may have its own distinct features, we generate questions and answers inquiring about its various attributes using an ensemble of 5 LLMs, later consolidating them using a neutral one (`claude-opus-4-1@20250805` [5]). Here, we specify the names and model versions of each such LLM for reproducibility ease.

- `gemini-2.5-pro` (Google, 2024)
- `gpt-4o-2024-08-06` [6]
- `Qwen2.5-VL-32B-Instruct` (Bai et al., 2025)
- `Mistral-Small-3.2-24B-Instruct-2506` [7]
- `Llama-3.2-11B-Vision-Instruct` [8]

The prompts used for these models can be found in the publicly released codebase.

### A.4 Hyperparameter Details

We use the `Gemini-2.5-flash` model for all our experiments due to its strong empirical performance (§4.2). Across all stages, the LLM and VLM are configured with a temperature of $0.0$, top-p value of $0.01$, and top-k value of $1$ to enforce deterministic generations. The maximum number of output tokens is set to $4000$, while thinking budget is set to dynamic mode. All experiments are executed using batch-processing mode for computational efficiency.

### A.5 Indoor-Outdoor Distribution of Images

For calculating background diversity, we classify whether each image depicts an indoor or an outdoor scene (see subsection 3 in the main paper). Table 2 details the indoor-outdoor distribution achieved from this step before conducting the remaining VQA steps of the pipeline. Since our chosen entities are inspired by those analyzed in the GeoDE dataset (Ramaswamy et al., 2023), we further mention the groups (indoor common, indoor rare, outdoor common, outdoor rare) to which each of the chosen entities belong to, as assigned by the authors. Notably, while most of the GeoDE images adhere to their assigned indoor/outdoor groups, synthetic datasets display major deviations in depiction of typically indoor entities like bags, chairs, stoves, and cooking pots, frequently generating them in outdoor settings.

## B Visibility Failures and NOTA Statistics

### B.1 Percentage of Images Failing the Visibility Check

Most entity-question pairs fail the visibility check for fewer than 5% of images. Table 3 highlights few of those with higher failure rates. All findings are qualitatively verified through image inspection to confirm the reasons for non-answerability. Below, we list our observations for each entity.

*Stove* images that are traditional wood-fired or charcoal-fired, fail for questions inquiring about the type of stove, and those with hidden/distorted cooktops fail for questions querying about the cooktop

---

[5]`https://www.anthropic.com/news/claude-opus-4-1`
[6]`https://platform.openai.com/docs/models/gpt-4o`
[7]`https://huggingface.co/mistralai/Mistral-Small-3.2-24B-Instruct-2506`
[8]https://huggingface.co/meta-llama/Llama-3.2-11B-Vision-Instruct

Table 2: **Indoor-Outdoor Distribution**.

| Group | Object | Indoor | | | | | | Outdoor | | | | | |
|---|---|---|---|---|---|---|---|---|---|---|---|---|---|
| | | GeoDE | SDv2 | SDv3 | SDv3.5 | FLUX.1 | Avg | GeoDE | SDv2 | SDv3 | SDv3.5 | FLUX.1 | Avg |
| *Indoor common* | bag | 95.71 | 3.78 | 10.07 | 13.34 | 35.16 | 31.61 | 4.29 | 96.22 | 89.93 | 86.66 | 64.84 | 68.39 |
| | chair | 88.11 | 1.41 | 10.86 | 12.38 | 84.09 | 39.37 | 11.89 | 98.59 | 89.14 | 87.62 | 15.91 | 60.63 |
| *Indoor rare* | cooking pot | 95.96 | 0.87 | 26.61 | 10.57 | 57.29 | 38.26 | 4.04 | 99.13 | 73.39 | 89.43 | 42.71 | 61.74 |
| | plate of food | 94.98 | 95.00 | 98.47 | 94.07 | 98.95 | 96.29 | 5.02 | 5.00 | 1.53 | 5.93 | 1.05 | 3.71 |
| | stove | 93.14 | 16.37 | 67.18 | 57.64 | 87.74 | 64.41 | 6.86 | 83.63 | 32.82 | 42.36 | 12.26 | 35.59 |
| *Outdoor common* | backyard | 0.06 | 0.00 | 0.00 | 0.00 | 0.02 | 0.02 | 99.94 | 100.00 | 100.00 | 100.00 | 99.98 | 99.98 |
| | car | 2.27 | 0.00 | 0.05 | 0.07 | 0.08 | 0.49 | 97.73 | 100.00 | 99.95 | 99.93 | 99.92 | 99.51 |
| | house | 0.00 | 0.00 | 0.02 | 0.00 | 0.00 | 0.00 | 100.00 | 100.00 | 99.98 | 100.00 | 100.00 | 100.00 |
| *Outdoor rare* | dog | 28.87 | 0.23 | 0.69 | 2.38 | 2.69 | 6.97 | 71.13 | 99.77 | 99.31 | 97.62 | 97.31 | 93.03 |
| | storefront | 8.59 | 0.00 | 0.23 | 0.93 | 0.46 | 2.04 | 91.41 | 100.00 | 99.77 | 99.07 | 99.54 | 97.96 |

type. The latter is higher for SD2.1, which shows depictions of distorted renderings of traditional or repurposed stoves with no discernable cooktop, and FLUX.1 which has similar depictions of wood-burning compartments with no visible cooktops.

*House* images fail for questions about *doors* when the *door* features are obscured. *Car* images where the roof is not clearly visible fails for the question on roof types. Daylight images of cars often fail for the question on whether the lights are on or off due to difficulty in observing the head and tail lights. *Chairs* in which the back is fully covered with fabric or obscured by cushions tend to fail on the question about the type of backrest (solid, slatted, or woven). Interestingly, this failure rate is lower for SD2.1 and SD3.5, suggesting a lower proportion of cushioned chairs in these datasets, a pattern corroborated by the responses to the question on cushioned versus hard seats. *Storefront* images with only display window visible or shutters fail for the question on type of entrance.

We define a question as "low-coverage" if the visibility checks retain fewer than $50\%$ of the original image set. Such questions are excluded from further processing. Among the 111 unique questions considered for entity diversity, we identify two that fall into this category: "What kind of controls are visible on the stove: knobs, buttons, or a touchscreen display?" and "Does the bag have a zipper, buckle, or flap closure?". The first is inherently difficult to answer using synthetic images, while in the second case, bag images often do not clearly reveal the type of closure.

Table 3: **Visibility Check Failure Rates for Selected Entity-Question Pairs Across Datasets.**

| Entity | Question | SD2.1 | SD3m | SD3.5 | FLUX.1 | GeoDE |
|---|---|---|---|---|---|---|
| Bag | Is the bag's closure type visible or identifiable in the image? | 44.5 | 50.5 | 43.05 | 28.95 | 25.22 |
| Car | Is it visible or detectable from the image if the car's lights are turned on or off? | 22.8 | 8.55 | 11.48 | 1.42 | 18.22 |
| | Is the car's roof type visible or identifiable in the image? | 14.92 | 30.47 | 20.8 | 22.32 | 11.12 |
| Chair | Is the construction style of the chair's backrest visible or identifiable in the image? | 2.28 | 22.7 | 1.95 | 15.67 | 23.25 |
| House | Is it visible or detectable from the image whether a door on the house is open or closed? | 17.15 | 9.72 | 9.9 | 2.03 | 30.11 |
| Storefront | Is the type of the storefront entrance visible or identifiable in the image? | 27.28 | 16.25 | 7.53 | 2.5 | 19.26 |
| Stove | Is the stove's cooktop type visible or identifiable in the image? | 36.05 | 10.2 | 12.0 | 26.25 | 0.92 |

## B.2    PERCENTAGE OF IMAGES WITH (NONE OF THE ABOVE) NOTA OPTIONS

During the VQA stage (i.e., the stage of obtaining answers to the questions from images before calculating the VDI scores) we add a 'None of the Above' option to the answer list for each question, as discussed in Subsection 4 (main paper). Table 4 details the NOTA percentages across datasets for all questions per entity. We qualitatively verify these cases by visually inspecting the images and the VQA model's reasoning for selecting NOTA.

- **Stove** has the highest NoTA percentage at $5.51\%$. The first question with high NOTA is *"What is the primary material of the stove's body: stainless steel or enamel/painted metal?"* It is comparatively higher for SD3.5, with a lot of rustic representations of stove with iron / stone / corrugated metal bodies, except for the UK, USA, and Japan. The other question with high NOTA is *"What type of cooktop does the stove have: gas burners, electric coils, or a flat glass/ceramic top?"* which again fails for images with representations of traditional stoves.
- For **storefront**, all datasets show similar NOTA (avg. $4.32\%$), mostly due to two questions: *"Is the facade primarily made of brick, wood, or glass?"* and *"Is the storefront entrance a single door, double doors, or a revolving door?"*. For the first, option Concrete/Stone may be missing. For the second question, open entrance (like in malls) and accordion-style metal gates are absent. In SD3m, both questions show stronger geographical disparities with lower NOTA for UK, USA, and Italy.
- For **bag**, the higher NOTA rate is observed to be a result of question on *"Does the bag have a zipper, buckle, or flap closure?"* which examines an attribute that is inherently open-ended. Thus, bags with drawstrings, open-topped totes, plastic bags with tied handles are not represented by this question.
- The slightly high NOTA rate for **car** results from *"Is the car a sedan or SUV?"* which does not cover all types of cars, missing options like hatchbacks.
- For **Cooking pot**, **backyard**, **chair**, **house**, **plate of food**, and **dog**, NOTA rate is consistently $< 3\%$ across all datasets.

For questions where more than $30\%$ of images result in NOTA, we include an **'Others'** as an option in the distribution.

Table 4: **NOTA percentages** per entity across datasets, with per-entity average.

| Entity | SD2.1 | SD3m | SD3.5 | FLUX.1 | GeoDE | Entity Avg. |
|---|---|---|---|---|---|---|
| Bag | 6.05 | 3.84 | 4.63 | 0.99 | 2.73 | 3.65 |
| Backyard | 0.22 | 0.48 | 0.19 | 0.52 | 0.33 | 0.35 |
| Car | 2.02 | 2.34 | 4.41 | 3.63 | 5.17 | 3.51 |
| Chair | 0.95 | 1.25 | 1.00 | 1.45 | 1.66 | 1.26 |
| Cooking Pot | 1.24 | 0.79 | 0.27 | 0.06 | 3.65 | 1.20 |
| Dog | 0.12 | 0.03 | 0.07 | 0.68 | 0.06 | 0.19 |
| House | 3.55 | 2.35 | 1.76 | 1.53 | 2.04 | 2.25 |
| Plate of Food | 2.12 | 0.74 | 1.29 | 0.98 | 3.07 | 1.64 |
| Storefront | 4.16 | 5.48 | 6.90 | 1.08 | 3.96 | 4.32 |
| Stove | 6.15 | 3.20 | 9.31 | 3.54 | 5.34 | 5.51 |
| **Dataset Avg.** | 2.66 | 2.98 | 2.05 | 2.80 | 1.44 | |

## C GEODIV DIVERSITY - EXTENDED ANALYSIS

### C.1 GEODIV DIVERSITY COMPARISON ACROSS ENTITIES

In section 5 of the main paper, we discuss the SEVI and VDI diversities across datasets and countries. In this section, we perform similar analyses, but based on the entities we chose for this paper. Our observations are noted below:

**SEVI Diversity Analysis.** The overall SEVI diversity is predictably low across entities, with average scores of $0.36$ for Affluence and $0.39$ for Maintenance. Among the entities, stove and chair images exhibit the highest diversity across both SEVI dimensions, while plate of food images are the least diverse. In terms of Affluence ratings (on a $1 - 5$ scale), backyard and house images receive the highest average scores ($3.34$), whereas cooking pot and stove images are rated as more impoverished (average rating: $2.50$). The trends for Maintenance ratings differ slightly: plate of food and dog images receive the highest ratings (average $4.66$), while cooking pot and stove images are rated lowest, mirroring the pattern observed for Affluence (average $3.20$). Overall, we observe not only a lack of diversity in the SEVI dimensions at the entity level but also significant differences in SEVI ratings, suggesting that models generate images reflecting varying socio-economic conditions depending on the entity prompted. These trends are demonstrated in Fig. 6.

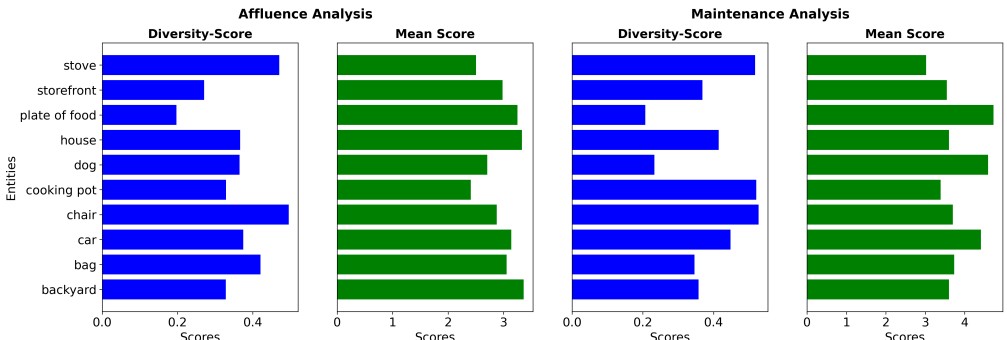

Figure 6: **Affluence and Maintenance (SEVI) Scores across Entities.** Chair and Stove images show the highest variance in Affluence, whereas Cooking Pot and Stove images appear the least affluent. For Maintenance, Stove, Cooking Pot and Chair turn out to be the most diverse, though the mean ratings are low for each of them.

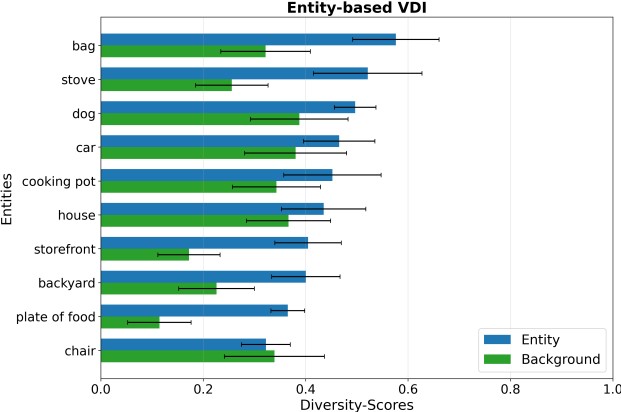

Figure 7: **Entity and Background Appearance (VDI) Scores across Entities.** While Bag and Stove images demonstrate considerably higher entity diversity, Chair and Plate of Food are the least diverse. The Background-Diversity for these Entities vary considerably, and are distinctly lower than the Entity Diversity-Scores. Plate of Food images understandably are the least diverse, as most of them are closeups of the entity itself, whereas dogs, cars and houses demonstrate variation in the background to some extent.

**VDI Diversity Analysis.** The Entity Appearance diversity, while low in general for most entities (with an average Diversity-Score of $0.44$), varies significantly among the same. For instance, Chair, Storefront and Plate of Food are the least diverse, owing to similar answers getting generated across countries (mean score of $0.36$). On the other hand, Bags and Stoves vary the most in their attribute values (with a mean score of $0.55$). The Background Diversity-Scores are considerably lower than those for the entities (mean score of $0.29$ across entities). While these scores are similar for 7 out of the 10 studied entities, the images belonging to Plate of Food, Storefront and Stove have strikingly low background variation, with a mean score of only $0.18$. While Plate of Food images are primarily closeups, Storefront and Stove images are also mostly placed in country-wise similar backgrounds. These trends are shown in Fig. 7.

### C.2 ANALYSIS ON OVERALL GEODIV SCORES

**GeoDiv Scores Across Countries.** GeoDiv comprises of four dimensions - Affluence and Maintenance (SEVI), with Entity and Background Diversity (VDI). We combine the Diversity-Scores obtained under each dimension by averaging, to compute a final geo-diversity score per country. The

scores can be seen in Table 5, where we find countries like the UK, US, Japan and India to have lower scores, in comparison with those like Egypt and Colombia.

Table 5: Average GeoDiv scores across countries.

| Country | GeoDiv Score |
|---|---|
| Egypt | 0.4106 |
| Colombia | 0.4079 |
| Turkey | 0.4049 |
| Spain | 0.4046 |
| Indonesia | 0.3999 |
| China | 0.3967 |
| Italy | 0.3942 |
| South Korea | 0.3932 |
| Philippines | 0.3915 |
| United Arab Emirates | 0.3878 |
| Nigeria | 0.3877 |
| Mexico | 0.3817 |
| United States | 0.3681 |
| United Kingdom | 0.3645 |
| Japan | 0.3623 |
| India | 0.3372 |

**GeoDiv Scores Across Datasets.** We further combine the SEVI and VDI scores by averaging, and report the final dataset-wise geo-diversity values, as estimated by GeoDiv in Table 6. While all datasets appear similarly diverse, SD2.1 images dominate the overall scores, whereas FLUX.1 images achieve the least scores. Overall, all datasets have low values, indicating the urgent need to enhance the geographical nuances in the generative models.

Table 6: Average GeoDiv Scores across models.

| Model | GeoDiv Score |
|---|---|
| SD2.1 | 0.4251 |
| SD3m | 0.3655 |
| SD3.5 | 0.3455 |
| FLUX.1 | 0.3153 |

## D ENTITY AND BACKGROUND DIVERSITY SCORES

### D.1 ENTITY DIVERSITY SCORES

Figure 8 presents heatmaps of entity-diversity scores across entities and countries.

**Dataset Level.** SD2.1 achieves the highest dataset-level average ($0.51$) and SD3.5 the lowest ($0.40$), as is evident in Figure 8. The variance across countries per dataset is generally $\approx 0.01$ across all T2I models, and across entities is in range $[0.001, 0.008]$. Variance is relatively small, reflecting homogeneous generations.

**Entity Level.** The average diversity across all datasets and countries varies notably by entity type. *Bags* show the highest average diversity at about $0.58$, followed by *stoves* ($0.52$) and *dogs* ($0.50$). *Chairs* have the lowest average diversity at around $0.32$, and *plate of food* also scores low at about $0.36$. *House* exhibits the highest variance across dataset ($0.004$). *Chair* and *dog* show the lowest dataset variances ($\approx 0.0007$), indicating consistent diversity levels across datasets for these entities. Variance across countries within an entity is generally higher than variance across datasets, with *cooking pots* showing the highest geographic variance ($\approx 0.02$), followed by *stoves* ($\approx 0.02$) and *dogs* ($\approx 0.01$). *Plate of food* and *cars* have the lowest country-level variances ($\approx 0.002$ and $\approx 0.004$, respectively), suggesting more consistent diversity worldwide for these categories.

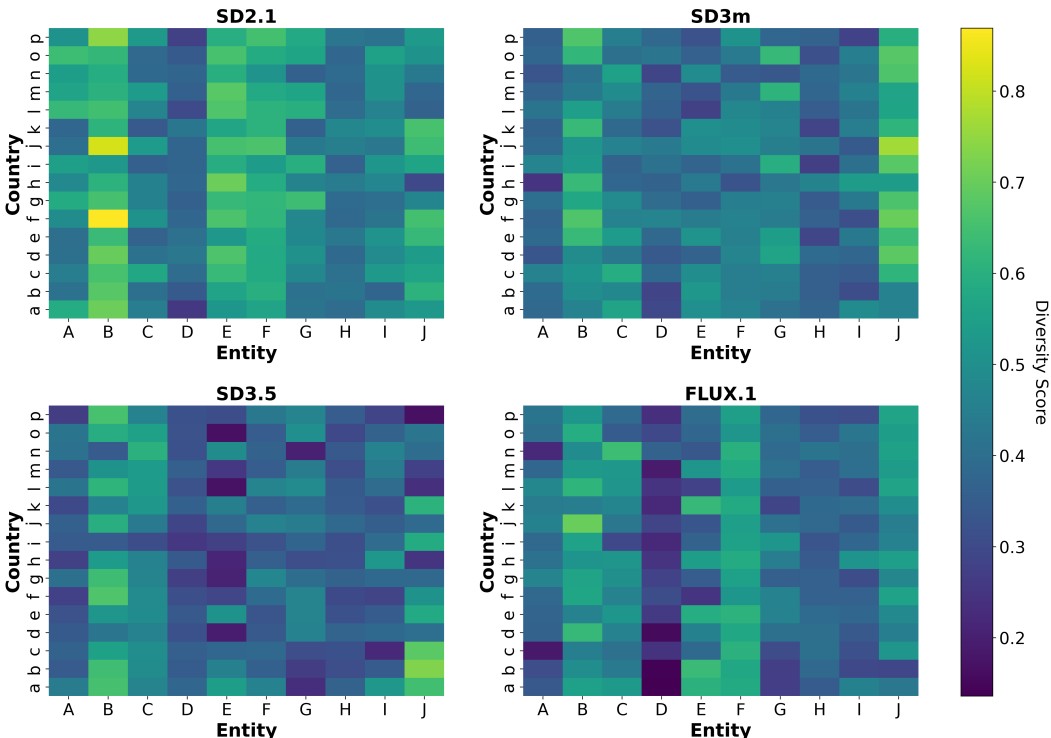

Figure 8: **Entity diversity scores across generative models.** *Countries (a-p)*: a) USA b) UK c) UAE
d) Turkey e) Spain f) South Korea g) Philippines h) Nigeria i) Mexico j) Japan k) Italy l) Indonesia
m) India n) Egypt o) Colombia p) China. **Entities (A-J)**: A) Backyard B) Bag C) Car D) Chair
E) Cooking pot F) Dog G) house H) Plate of food I) Storefront J) Stove. The dataset with highest
average diversity is SD2.1 and lowest is FLUX.1.

**Country Level** spread is narrow, from $0.43$ (Mexico) to $0.47$ (Japan), indicating that the country-
level differences are subtle compared to dataset and entity-level differences. This is evident from
Figure 8 which shows higher variation horizontally (along Entity) than vertically (along Country).
Cross-country stability (coefficient of variation across country means) indicate SD3.5 is the most
polarized by country (SD2.1 ($\approx 0.04$), SD3m ($\approx 0.04$), FLUX.1 ($\approx 0.05$), and SD3.5 ($\approx 0.07$)).

## D.2 BIAS PATTERNS IN ENTITY ATTRIBUTES REVEALED BY GEODIV

As discussed in § 5 in the main paper, we observe both global and cross-country biases within model
generations. The below observations relate to most countries, making the biases in global in nature.

1. *Chairs* without backrests are absent in SD3.5 and FLUX.1, chairs with single central bases
   never appear; exclusively multi-legged designs are visible in all the synthetic datasets. SD3.5
   and FLUX.1 have a bias towards **brown** coloured, **cushioned**, and **solid**-backed chairs,
   while SD2.1 defaults to **slatted**-backed, **wooden** chairs.

2. *Backyard* images in FLUX.1 are almost always **grass**-only, with distinct **pathways** and
   **plants and shrubs**. Interestingly, while all datasets hardly show any **grass** cover for Nigeria,
   FLUX.1 images for Nigeria are largely biased *towards* the same.

3. SD2.1 images of *Bag* default to **non-geometric/unstructured** shaped bags, FLUX.1 defaults
   to **brown**-coloured, **leather** bags.

4. While majority of SD2.1 *car* images do not have **logos or brand badges**, SD3m and FLUX.1
   almost always do.

5. Single-handled **Cooking pots** or those without handles are hardly generated, defaulting to
   only multiple-handled variations across all the datasets.

6. SD3m images only show *dogs* with **folded** ears unlike the other datasets which show higher diversity.

7. *Plate of food* displays one of the lowest diversities across datasets, always depicting **vegetables**, dense with **multiple types** of food in the plate, almost always with some **garnish**, and **white**, **round** plates.

8. The cooktop type of *Stove* images in SD3.5 and FLUX.1 are only **gas burners**.

9. FLUX.1 always depicts multi-storeyed *houses* with chimneys, porches, grass and paving ground-cover (except Egypt), trees.

Such biases vary in severity across datasets, but others reveal alarming geographic variations. Here we note some examples of such biases across countries:

1. *Chair*: SD3m shows very few **cushioned** chairs for Nigeria and the Philippines, and images for Egypt show an over-representation of chairs with **woven** backrests, whereas the UK and USA samples rarely depict hard-seated chairs.

2. *Backyard*: SD2.1 and SD3m images for Nigeria show no **patio / deck**, while for Spain it is always present. There is a striking bias in depiction of primary **ground cover** in most datasets, for Nigeria (only **dirt/gravel**), India and Egypt (no **grass**), USA (only **grass**). UK and USA images are always depicted with **outdoor furniture**, while it is biased towards absence for Nigeria.

3. *Bag* images show country-specific biases for **material**: SD2.1 and SD3.5 bags are biased towards **fabric** in general, but Nigeria has a higher proportion of **plastic**, while the UK, USA, Italy and Japan are the only countries showing **leather**; SD3m shows only **fabric** bags for India; Mexico shows higher proportion of **patterned** and **fabric** bags, even in FLUX.1 which is otherwise biased towards **leather**. SD3.5 images for Egypt, India, Mexico and Turkey do not have any visible **brand logo or label**.

4. *Car* images show a consistent bias towards **unpaved** surfaces for Nigeria and Egypt across most datasets, including in FLUX.1 which otherwise defaults to paved surfaces. SD3.5 images for Mexico do not show **logos or brand badges**, while defaulting to always showing for most other countries.

5. *Storefront* images in FLUX.1 always have **lights on** except in Nigeria. SD3m shows higher diversity for presence of **sidewalk** only for Nigeria, leaning towards 'no', whereas it defaults to 'yes' for other countries.

6. *Stove* shows high disparity in representation across countries, especially in SD3.5. In SD3.5, UK and USA only have **multiple burner** stoves while India, Nigeria, and Egypt only show **single burner** ones. In fact, SD3.5 has disproportionately chooses cooktop type as *others* for almost all countries, especially Egypt ($> 93\%$), while UK and USA are equally biased towards gas burners. SD2.1 doesn't show ovens along with the stoves for most countries, except in USA where it exclusively shows those with ovens.

7. *House* images for Egypt and UAE show a bias towards being depicted solely as **flat**-roofed. Ground cover for Egypt, Nigeria, India never show grass, while USA always shows only grass. SD3m doesn't even show paving as ground cover for Egypt, Nigeria, India, only dirt/gravel. For SD3.5, house images of Egypt share distinct features compared to the other countries, owing to its overrepresentation of stones as the primary construction material, and dirt/gravel as the ground cover (see Fig. 9).

There are also some country-specific patterns that seem to be consistent across datasets and entities. For example, there is an apparent correlation between China and the colour **red**. While FLUX.1 *bag* images are biased towards **brown** colour, in case of China it is biased towards **red**. Some other entities and datasets that show red-colour bias for China include *Chairs* and *Bags* in SD3m, and *Storefront* in SD2.1, SD3.5, and FLUX.1.

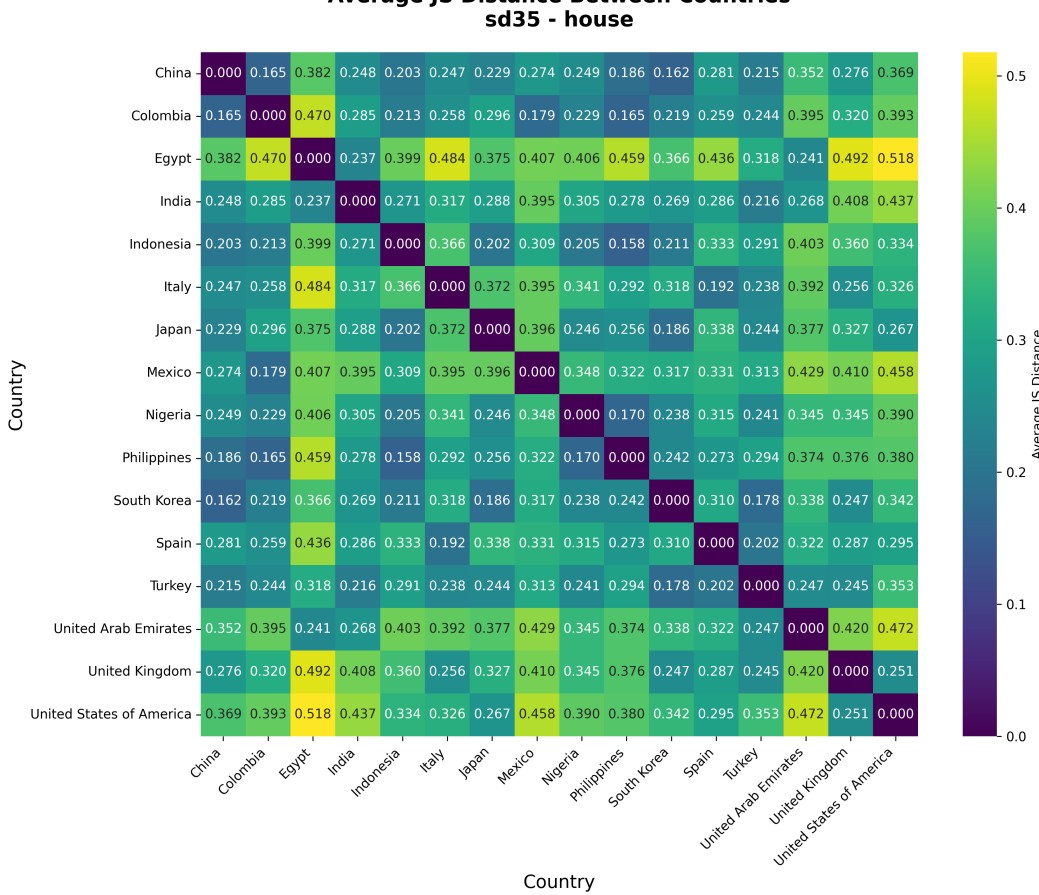

Figure 9: **Jensen Shannon Distances (JSD) of Entity Attribute Distributions Across Countries for SD3.5 images of House.** We note the higher JSD values for countries like Egypt, Mexico and USA, signalling that they possess distinct features compared to the other studied countries.

### D.3 BACKGROUND DIVERSITY SCORES

Figure 10 illustrates the entity- and country-wise background diversity score heatmaps. Compared to Entity Diversity Scores, the Background Diversity Scores are lower.

**Dataset Level.** The overall average background diversity across all synthetic datasets and entities is $0.31$. As with entity diversity, SD2.1 scores the highest at $0.35$, followed by FLUX.1 ($0.32$) and SD3m ($0.31$). SD3.5 records the lowest at $0.28$. The variance across countries per dataset is approximately $0.02$, and across entities is in range $[0.001, 0.009]$.

**Entity Level.** Highest background diversity is for dogs ($0.42$) and cars ($0.40$), reflecting naturally varied scenes. Lowest are plate of food ($0.10$) and storefront ($0.21$), both entities with limited background depictions across generated images. As Figure 10 shows, *plate of food* images with no background context were dropped from the VQA pipeline through the visibility checks. Largest dataset-to-dataset disagreement occurs for bags and cars ($0.01$), suggesting models differ most in how they situate these objects. Chairs ($0.03$) and dogs ($0.02$) show the highest cross-country variability, implying strong geographic differences in their backgrounds.

**Country Level.** Highest background diversity is seen in Indonesia ($0.37$), Nigeria ($0.36$), and Colombia ($0.35$). Lowest diversity appears in Italy ($0.25$), Spain ($0.27$), and the UK ($0.27$). Overall, developing regions (Nigeria, Indonesia, Philippines) tend to show richer background variation, while European countries (Italy, Spain, UK) exhibit more uniform contexts. China ($0.08$) and Italy ($0.08$) show the lowest variance, suggesting more consistent backgrounds across different objects.

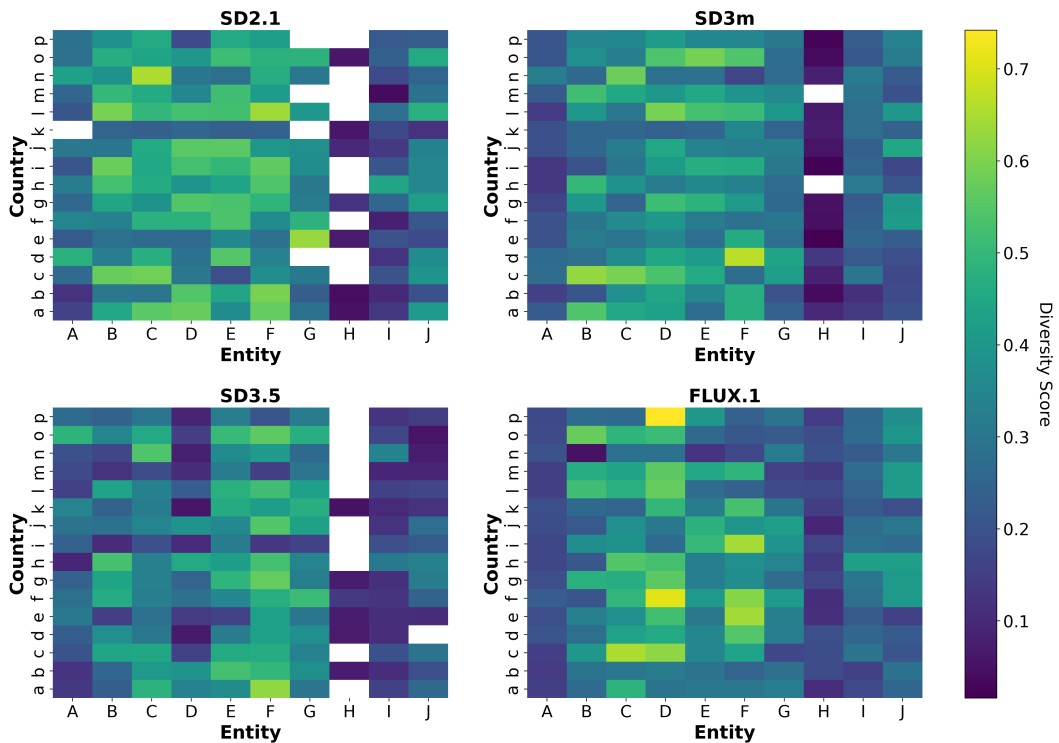

Figure 10: **Background diversity scores across generative models.** *Countries (a-p)*: a) USA b) UK c) UAE d) Turkey e) Spain f) South Korea g) Philippines h) Nigeria i) Mexico j) Japan k) Italy l) Indonesia m) India n) Egypt o) Colombia p) China. **Entities (A-J)**: A) Backyard B) Bag C) Car D) Chair E) Cooking pot F) Dog G) house H) Plate of food I) Storefront J) Stove.

Table 7: **Comparison of entity and background diversity across datasets.** SD21 ranks first for entity and background diversity but with notable variance. FLUX.1 and SD3m are more consistent but less diverse. [†]Dataset rank is based on number of entities for which it had highest diversity.

| Dataset | Entity Diversity | | | Background Diversity | | |
|---|---|---|---|---|---|---|
| | Rank[†] | Mean | Std | Rank[†] | Mean | Std |
| SD21 | 1 (7/10) | 0.508 | 0.114 | 1 (7/10) | 0.354 | 0.149 |
| SD3m | 2 (2/10) | 0.448 | 0.104 | 3 (0/10) | 0.306 | 0.137 |
| FLUX.1 | 3 (0/10) | 0.424 | 0.117 | 2 (3/10) | 0.317 | 0.141 |
| SD35 | 4 (1/10) | 0.397 | 0.114 | 4 (0/10) | 0.283 | 0.143 |

**Summary.** We evaluate both entity- and background-level diversity across datasets by comparing average diversity scores, entity-wise rankings, and per-country variation (see Table 7).

# E SEVI SCORES - COUNTRY AND ENTITY-WISE DETAILS

## E.1 AFFLUENCE SCORES

Figure 11 details the country-entity wise affluence Diversity-Scores. It clearly shows which for which entities and T2I models, which countries show least variance in Affluence level, whereas overall, the diversity across T2I models appears similar.

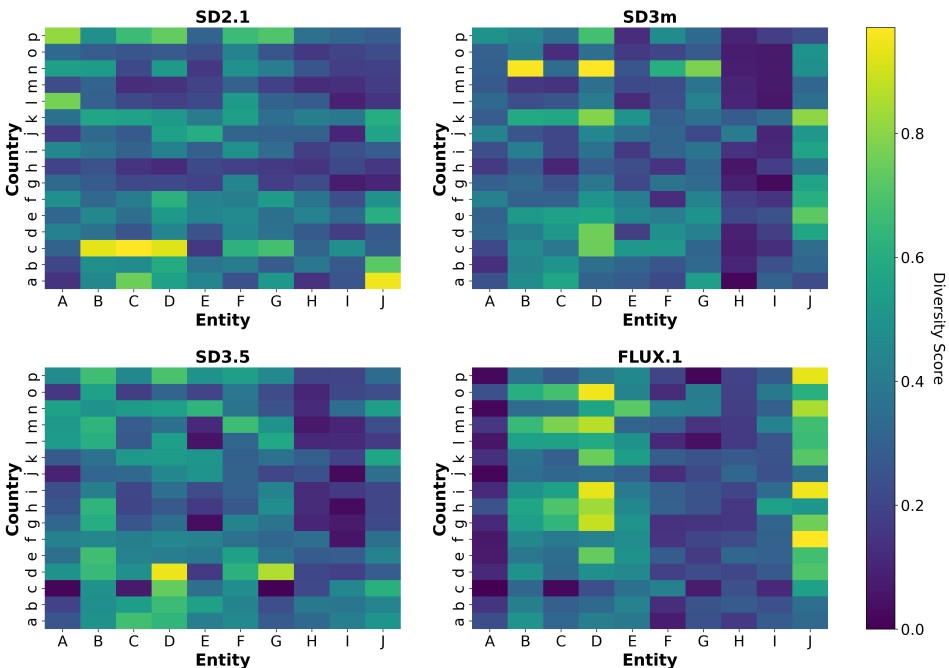

Figure 11: Affluence diversity scores across generative models. **Countries (a-p)**: a) USA b) UK c) UAE d) Turkey e) Spain f) South Korea g) Philippines h) Nigeria i) Mexico j) Japan k) Italy l) Indonesia m) India n) Egypt o) Colombia p) China. **Entities (A-J)**: A) Backyard B) Bag C) Car D) Chair E) Cooking pot F) Dog G) house H) Plate of food I) Storefront J) Stove.

## E.2 MAINTENANCE SCORES

Figure 12 details the country-entity wise maintenance Diversity-scores. FLUX.1 has remarkably low diversity in terms of its maintenance, across countries and entities.

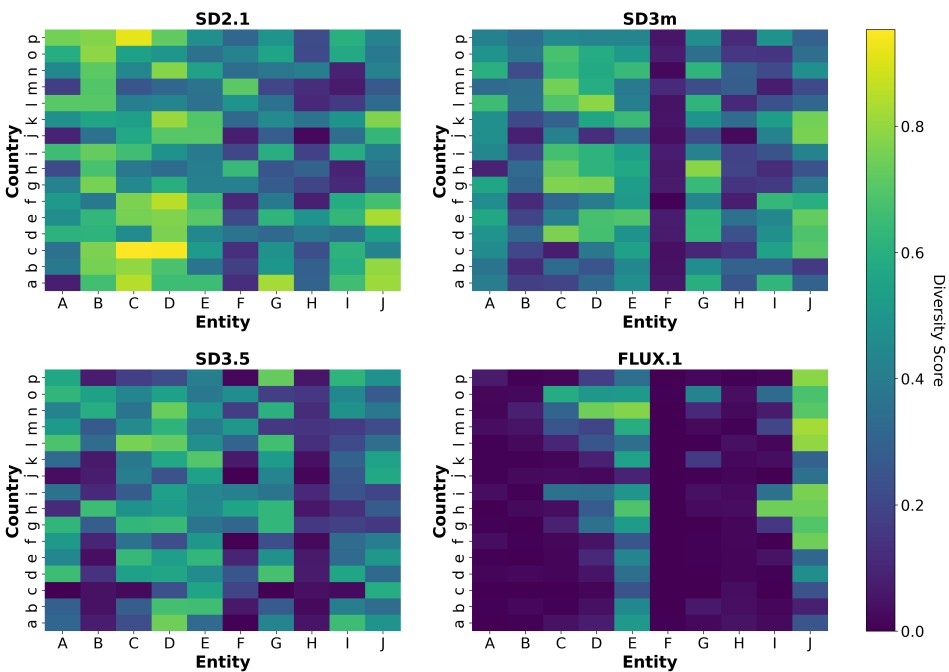

Figure 12: Maintenance diversity scores across generative models. **Countries (a-p)**: a) USA b) UK c) UAE d) Turkey e) Spain f) South Korea g) Philippines h) Nigeria i) Mexico j) Japan k) Italy l) Indonesia m) India n) Egypt o) Colombia p) China. **Entities (A-J)**: A) Backyard B) Bag C) Car D) Chair E) Cooking pot F) Dog G) house H) Plate of food I) Storefront J) Stove.

# F    GeoDE: Observations on a Real-World Dataset

## F.1    Data Distribution

Table 8 provides the entity-country wise counts of images in the GeoDE dataset used in this work.

Table 8: **GeoDE entity-country distribution**. In the table below, we show the entity counts by country.

| Entity | UK | Nig | Tur | Indo | Col | Jap | Ind | Chi | USA | Mex | UAE | SKor | Spa | Ita | Egy | Phil |
|---|---|---|---|---|---|---|---|---|---|---|---|---|---|---|---|---|
| backyard | 60 | 352 | 192 | 125 | 64 | 153 | 0 | 13 | 0 | 63 | 13 | 23 | 33 | 74 | 13 | 59 |
| bag | 103 | 176 | 178 | 312 | 126 | 212 | 0 | 73 | 0 | 87 | 26 | 77 | 124 | 154 | 75 | 208 |
| car | 92 | 203 | 161 | 136 | 97 | 139 | 0 | 80 | 0 | 58 | 35 | 51 | 106 | 137 | 54 | 45 |
| chair | 84 | 137 | 177 | 270 | 143 | 183 | 0 | 66 | 0 | 68 | 45 | 74 | 96 | 142 | 121 | 175 |
| cooking pot | 75 | 116 | 162 | 87 | 110 | 177 | 0 | 25 | 0 | 56 | 23 | 35 | 95 | 97 | 54 | 59 |
| dog | 24 | 93 | 214 | 24 | 79 | 142 | 0 | 27 | 0 | 77 | 0 | 21 | 43 | 85 | 27 | 161 |
| house | 63 | 307 | 150 | 117 | 108 | 168 | 0 | 12 | 0 | 74 | 20 | 32 | 58 | 53 | 20 | 40 |
| plate of food | 38 | 235 | 154 | 203 | 74 | 216 | 0 | 25 | 0 | 103 | 30 | 47 | 83 | 94 | 95 | 84 |
| storefront | 38 | 143 | 161 | 133 | 86 | 116 | 0 | 40 | 0 | 66 | 21 | 35 | 70 | 60 | 71 | 45 |
| stove | 46 | 256 | 137 | 140 | 90 | 161 | 0 | 15 | 0 | 66 | 31 | 25 | 68 | 128 | 73 | 73 |
| Total | 623 | 2018 | 1686 | 1547 | 977 | 1661 | 0 | 432 | 0 | 634 | 278 | 476 | 792 | 1114 | 647 | 953 |

## F.2    Entity-Appearance Diversity

The GeoDE real-world dataset has an average diversity score of $0.60$, noticeably higher than that of the synthetic datasets analyzed (which range roughly between $0.40$ to $0.51$). Despite this higher diversity, GeoDE still exhibits inherent biases. For example, while generated images of cars display reasonable variability in viewing angles, GeoDE car images tend to be biased towards side views. This highlights how even carefully curated real datasets have distributional skew.

## F.3    Background-Appearance Diversity

GeoDE shows the strongest background variation ($0.41$), higher than the T2I models. However, background diversity is still significantly lower than entity diversity-score ($0.60$). Figure 13 shows the heatmaps for GeoDE across all four axes of diversity.

# G    Improving Geo-Diversity Using GeoDiv: An Application

Based on our discussion in § 3, GeoDiv assesses the geo-diversity of a set of images belonging to a certain entity and country. Applied to images from multiple diffusion-based models, the proposed framework uncovers significant lack of visual and socio-economic diversities. In this section, we demonstrate how the insights it provides can be directly applied to improve inclusivity in practice. As the GeoDiv framework produces detailed distributions over answer categories and socio-economic traits, it enables identification and correction of geographical imbalances for data curators. Similarly, model creators can use these metrics to uncover and mitigate model biases–something we illustrate with a concrete example.

Building on findings from prior work (Basu et al., 2023; Askari Hemmat et al., 2024), which suggest that prompt design can reduce generative model biases, we apply a simple mitigation strategy using our Affluence scores. We observe that the Affluence ratings for India were among the lowest across countries when using a default prompt (e.g., "photo of a house"). To counter this, we design new prompts that explicitly specify different affluence levels, and generated images accordingly. The number of images generated per affluence level was inversely proportional to the distribution predicted by the VQA model on the original image set. To assess the impact of this intervention, we ask human annotators from India to label both the original and the balanced image sets, and computed the diversity-score of the resulting distributions. We found that this prompt-based balancing strategy leads to an increase in diversity for every model evaluated, with an **average increase of 0.33**, indicating improved diversity in the generated outputs (see Table 9).

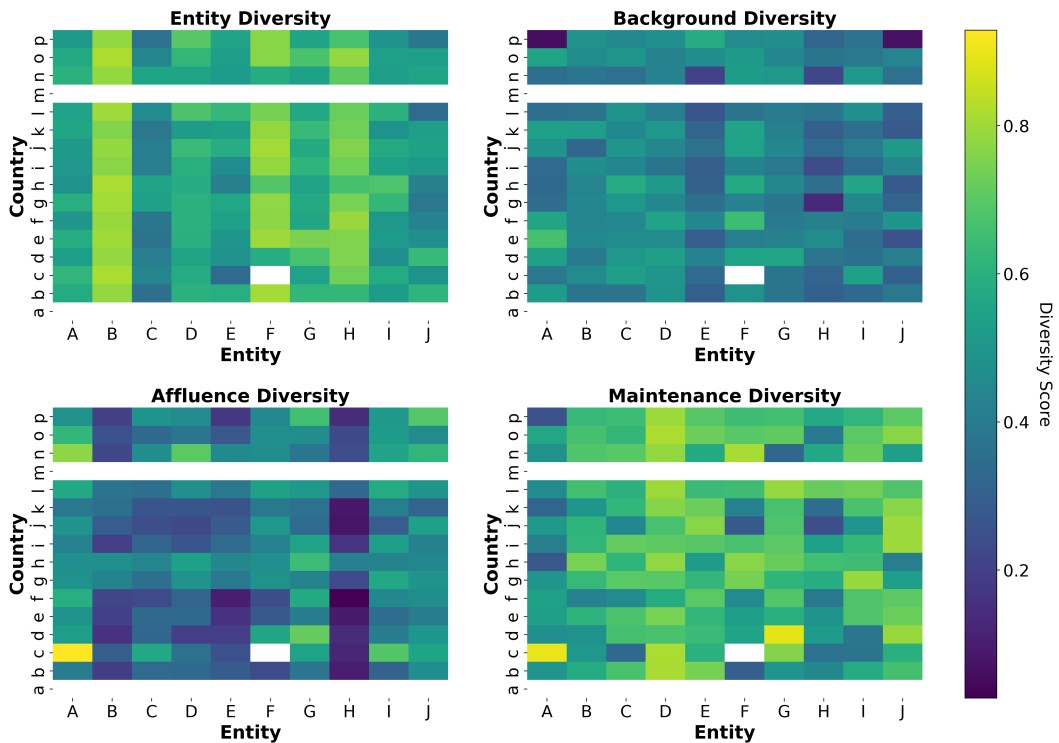

Figure 13: Diversity scores across the four axes for GeoDE. **Countries (a-p)**: a) USA b) UK c) UAE d) Turkey e) Spain f) South Korea g) Philippines h) Nigeria i) Mexico j) Japan k) Italy l) Indonesia m) India n) Egypt o) Colombia p) China. **Entities (A-J)**: A) Backyard B) Bag C) Car D) Chair E) Cooking pot F) Dog G) house H) Plate of food I) Storefront J) Stove.

Table 9: Improvement in Affluence Diversity achieved by utilizing GeoDiv's Affluence Scores

| Model | Original | Balanced | Difference |
|-------|----------|----------|------------|
| SD2.1 | 0.56 | 0.94 | +0.38 |
| SD3m | 0.62 | 0.87 | +0.25 |
| FLUX.1 | 0.52 | 0.88 | +0.36 |

While our mitigation strategy is simple, it demonstrates that once our metrics reveal underlying biases, they can be used to guide actionable interventions that enhance fairness and representation in generated content.

## H QUESTION-ANSWER (QA) SET FOR VDI SCORES

### H.1 QA SET FOR ENTITY DIVERSITY PART OF VDI SCORES.

Table 10 provides the entity-wise question and answer-lists used for calculating entity diversity.

Table 10: Entity-wise questions and their corresponding answer lists.

| Entity | Question | Answer List |
|---|---|---|
| Backyard | 1. Are there any animals or pets in the backyard? | Yes, No |
| | 2. Are there any distinct pathways or walkways visible in the backyard? | Yes, No |
| | 3. Are there any plants, trees, or shrubs in the backyard? | Yes, No |
| | 4. Are there any structures (e.g., a shed, playhouse) in the backyard? | Yes, No |
| | 5. Are there any visible recreational items (e.g., a swing set, trampoline, basketball hoop) in the backyard? | Yes, No |
| | 6. Is a body of water (e.g., a pool, pond, or fountain) visible in the backyard? | Yes, No |
| | 7. Is there a garden or vegetable patch in the backyard? | Yes, No |
| | 8. Is there a patio or deck attached to the house in the backyard? | Yes, No |
| | 9. Is there a visible grill or outdoor kitchen area in the backyard? | Yes, No |
| | 10. Is there any outdoor furniture (e.g., a table, chairs) in the backyard? | Yes, No |
| | 11. What is the primary ground cover in the backyard: grass, paving (concrete/tiles/stone), or dirt/gravel? | Grass, Paving, Dirt/Gravel |
| Bag | 1. Is a brand logo or label visible on the bag? | Yes, No |
| | 2. Does the bag have any visible external pockets or compartments? | Yes, No |
| | 3. Does the bag have a zipper, buckle, or flap closure?? | Zipper, Buckle, Flap |
| | 4. Does the bag have handles, a shoulder strap, or both? | Handles, Both, Shoulder strap |
| | 5. Is the bag a backpack, handbag, or tote bag? | Backpack, Tote bag, Handbag |
| | 6. Is the bag being carried by a person, placed on a surface, or hanging? | Carried by a person, Placed on a surface, Hanging |
| | 7. Is the bag's overall shape best described as rectangular, circular, trapezoidal, or non-geometric/unstructured? | Circular, Unstructured, Rectangular, Trapezoidal |
| | 8. Is the bag made of fabric, leather, or plastic? | Plastic, Fabric, Leather |
| | 9. Is the bag's surface a solid color or patterned? | Solid color, Patterned |
| | 10. What is the main color of the bag? | White, Black, Purple, Blue, Green, Orange, Red, Yellow, Brown, Pink, Gray |
| Car | 1. Are any wheels visible on the car? | Yes, No |
| | 2. Are there any logos or brand badges on the car? | Yes, No |
| | 3. Are any of the following modifications visible on the car: a spoiler, a roof rack, or custom rims? | Yes, No |
| | 4. Is there a license plate on the car? | Yes, No |

*(continued from previous page)*

| Entity | Question | Answer List |
|---|---|---|
| | 5. Is the car a convertible or does it have a fixed roof? | Convertible, Fixed roof |
| | 6. Is the car viewed from the front, side, or rear? | Front, Rear, Side |
| | 7. Does the car appear modern or vintage? | Modern, Vintage |
| | 8. Are the car's lights turned on or off? | On, Off |
| | 9. Is the car a sedan or SUV? | Sedan, Suv |
| | 10. Is the car moving or stationary? | Stationary, Moving |
| | 11. Is the car on a paved surface (like a street or driveway) or an unpaved one (like grass or dirt)? | Unpaved surface, Paved surface |
| | 12. What is the primary color of the car? | White, Black, Blue, Orange, Brown, Red, Yellow, Green, Beige, Gray |
| Chair | 1. Does the chair have a backrest? | Yes, No |
| | 2. Does the chair have armrests? | Yes, No |
| | 3. Does the chair have wheels? | Yes, No |
| | 4. Is the seat of the chair cushioned or hard? | Hard, Cushioned |
| | 5. Does the chair have multiple distinct legs or a single central base? | Multiple distinct legs, A single central base |
| | 6. Is the chair designed for a single person or multiple people? | Multiple people, Single person |
| | 7. Is the chair's seat primarily square or round? | Round, Square |
| | 8. Is the backrest of the chair solid, slatted, or woven? | Slatted, Woven, Solid |
| | 9. What is the primary material of the chair (e.g., wood, metal, plastic, fabric, woven)? | Stone, Metal, Leather, Wood, Plastic, Fabric |
| | 10. Is the chair's backrest straight or curved? | Straight, Curved |
| | 11. What is the primary color of the chair? | White, Black, Purple, Blue, Orange, Brown, Red, Yellow, Green, Gray |
| Cooking Pot | 1. Are there any visible markings, patterns, or logos on the cooking pot? | Yes, No |
| | 2. Does the pot have a lid on it? | Yes, No |
| | 3. Is the pot placed on a cooking surface (e.g., stove, burner, or fire)? | Yes, No |
| | 4. Is the pot taller than it is wide? | Yes, No |
| | 5. Is any food or liquid visible inside the pot? | Yes, No |
| | 6. Does the pot have a single handle or multiple handles? | No handles, Single handle, Multiple handles |
| | 7. What material does the pot appear to be made of? | Copper, Cast iron, Stainless steel, Ceramic, Enamel |
| | 8. What is the primary color of the cooking pot? | Blue, Red, Copper, Silver, Green, Brown, Orange, White, Black |
| Dog | 1. Are there any objects like toys, a leash, or food near the dog? | Yes, No |
| | 2. Is the dog wearing an accessory (e.g., collar, harness)? | Yes, No |
| | 3. Is the dog alone or with other animals or people? | With other animals, Alone, With people, With other animals and people |

*(continued on next page)*

*(continued from previous page)*

| Entity | Question | Answer List |
|---|---|---|
| | 4. What is the dog's primary activity or posture (e.g., standing, sitting, lying down, in motion/playing, eating, sleeping)? | Walking, Eating, Running, Playing, Standing, Lying down, Sitting |
| | 5. Is the dog in an indoor or outdoor setting? | Outdoor, Indoor |
| | 6. Does the dog's fur appear predominantly as a single solid color, or does it have multiple distinct colors/patterns (e.g., spots, patches)? | Multiple colors/patterns, Single solid color |
| | 7. Does the dog have short, medium, or long fur? | Medium, Long, Short |
| | 8. Are the dog's ears floppy (whole ear droops down), erect, or folded (ear starts upright but bends partway)? | Erect, Folded (ear starts upright but bends partway), Floppy (whole ear droops down) |
| | 9. Is the dog's mouth open or closed? | Closed, Open |
| House | 1. Are there any trees visible near the house? | Yes, No |
| | 2. Do the windows on the house have shutters? | Yes, No |
| | 3. Does the house have a porch or a balcony? | Yes, No |
| | 4. Is there a chimney on the house? | Yes, No |
| | 5. Is there a fence on the property? | Yes, No |
| | 6. Is there a garage visible, attached to the house? | Yes, No |
| | 7. What is the main color of the house's exterior? | White, Yellow, Brown, Beige, Gray |
| | 8. Is the house single-storey or multi-storey? | Multi-storey, Single-storey |
| | 9. What is the primary ground cover around the house: grass, paving (concrete/tiles/stone), or dirt/gravel? | Grass, Paving, Dirt/gravel |
| | 10. Is the roof of the house flat or sloped? | Flat, Sloped |
| | 11. What is the primary exterior material of the house? | Concrete, Stone, Metal, Wood, Glass, Brick |
| | 12. Is a door on the house open or closed? | Closed, Open |
| Plate of Food | 1. Are any vegetables visible on the plate? | Yes, No |
| | 2. Is there any food item on the plate that visually resembles meat, fish, or eggs? | Yes, No |
| | 3. Is a sauce or liquid topping visible on the food? | Yes, No |
| | 4. Is any cutlery (e.g., fork, knife, spoon) visible next to the plate? | Yes, No |
| | 5. Is more than half of the plate's surface covered by food? | Yes, No |
| | 6. Is the food on the plate topped with any garnish, like fresh herbs or seeds? | Yes, No |
| | 7. Is the plate a single solid color? | Yes, No |
| | 8. Is there any food item on the plate that visually resembles rice, bread, pasta, or potatoes? | Yes, No |
| | 9. Are there smaller dishes or bowls visible along with the main plate of food? | Yes, No |
| | 10. Is the plate primarily white or black? | White, Black |

*(continued on next page)*

*(continued from previous page)*

| Entity | Question | Answer List |
|---|---|---|
| | 11. Is the plate round or square? | Square, Round |
| | 12. Is the plate made up of a single kind of food (e.g., only cookies) or multiple different types (e.g., rice, curry, and vegetables)? | Single, Multiple |
| | 13. Is the plate of food on a table, placemat, or countertop? | Placemat, Table, Countertop |
| | 14. Is the food on the plate solid, liquid, or a mix of both? | A mix of both, Solid, Liquid |
| Storefront | 1. Are there any items placed outside the storefront, such as displays, furniture, or plants? | Yes, No |
| | 2. Are there any lights on inside or on the exterior of the storefront? | Yes, No |
| | 3. Are there any signs or logos identifying the store visible on the storefront? | Yes, No |
| | 4. Are there products or displays visible in the storefront window? | Yes, No |
| | 5. Does the storefront have an awning or a canopy? | Yes, No |
| | 6. Is there a sidewalk in front of the storefront? | Yes, No |
| | 7. Is there an 'Open' or 'Closed' sign on the storefront? | Yes, No |
| | 8. Is the storefront entrance a single door, double doors, or a revolving door? | Single door, Double doors, Revolving door |
| | 9. Is the storefront part of a larger building or a standalone structure? | Part of a larger building, Standalone structure |
| | 10. Is the facade primarily made of brick, wood, or glass? | Glass, Wood, Brick |
| | 11. Is the main entrance door to the storefront open or closed? | Closed, Partially open, Open |
| | 12. What is the primary color of the storefront's facade? | Blue, Red, Pink, Purple, Gray, Green, Yellow, Orange, Brown, White, Beige |
| Stove | 1. Are there multiple burners or heating zones visible on the cooktop? | Yes, No |
| | 2. Does the stove have a backguard or splash guard? | Yes, No |
| | 3. Does the stove's oven door have a glass window? | Yes, No |
| | 4. Is there a digital clock or timer display on the stove? | Yes, No |
| | 5. Is there a range hood or vent above the stove? | Yes, No |
| | 6. Is there an oven integrated below the cooktop? | Yes, No |
| | 7. Is there any cookware, such as a pot or pan, on the stove? | Yes, No |
| | 8. What kind of controls are visible on the stove: knobs, buttons, or a touchscreen display? | Touchscreen display, Buttons, Knobs |
| | 9. What is the primary material of the stove's body: stainless steel or enamel/painted metal? | Stainless steel, Enamel/painted metal |

*(continued on next page)*

*(continued from previous page)*

| Entity | Question | Answer List |
|---|---|---|
| | 10. What is the primary color of the stove? | Red, Blue, Cream, Gray, Silver, Green, White, Black |
| | 11. What type of cooktop does the stove have: gas burners, electric coils, or a flat glass/ceramic top? | Gas burners, Electric coils, Flat glass/ceramic top |
| | 12. Is the stove freestanding or built into the surrounding counter? | Built-in, Freestanding |

## H.2 QA set for Background Diversity part of VDI scores.

Table 11 provides the question-answer list set (common across all entities) for calculating background diversity.

Table 11: Questions and their corresponding answer lists for Background Diversity Scores.

| Scene | Question | Answer List |
|---|---|---|
| Indoor | 1. Which main elements are visible in the background? | Walls, Windows, Furniture, Appliances (e.g., fridge, microwave, washing machine), Electronic equipment (e.g., tvs, computers, speakers), Plain / solid color background |
| | 2. What type of floor or ground is visible? | Tiled floor, Wooden floor, Carpeted floor, Concrete floor |
| | 3. What type of environment is visible? | Residential, Commercial / public, Plain / solid color background |
| | 4. What best describes the visual order in this image? | Organized (several elements present, but neat, intentional arrangement), Cluttered (many elements, visually noisy, no clear order), Minimalist (very few or no elements at all, mostly empty or plain) |
| Outdoor | 1. What natural features, if any, are visible in the background of the image? | Trees / forest / plants, Mountains / hills, Waterbody, Open ground / fields |
| | 2. What type of modern infrastructure is visible in the background? | Transport-related (paved roads, vehicles, bridges, rail tracks), Utility-related (electric poles, wires, water tanks, pipelines), High-rise / industrial (skyscrapers, factories, construction sites, large machinery) |
| | 3. How dense is the built environment in the background? | Sparse / open (fields, wide spaces, few or no buildings), Moderate (some houses/buildings, not crowded), Dense / crowded (clustered buildings, narrow streets, crowded interiors) |
| | 4. What type of road or terrain is visible? | Paved road, Dirt / gravel road (man-made), Natural ground / grass (wild, non-constructed), Tiled / courtyard-style surface |
| | 5. What type of background elements are most visible? | Natural (trees, sky, soil, water, mountains), Built structures (walls, windows, houses, buildings, fences), Mixed (both natural and built elements visible) |
| | 6. How busy does the background appear, crowded (many people, vehicles, signs of activity), moderately busy (some human activity), or quiet / empty (few or no people or vehicles)? | Crowded, Moderately busy, Quiet / empty |

---

**A Study on Image-based Question Answering in Japan**

You will be shown a number of images, and each such image will be accompanied by **THREE questions**. Each image will primarily portray an entity. The questions will enquire about three things:

- **Affluence**: Whether the **overall** image reflects **impoverished** or **affluent conditions**.
- **General Condition**: The **physical state** of the **depicted entity** (e.g., worn, damaged, or pristine).
- **Cultural Localization**: The extent to which **culturally specific symbols** (e.g., religious motifs, traditional architecture) of **your country** are present versus globalized visual cues in the **entity**.

Answer **ALL** questions.
**Total time: 30 minutes**

**Instructions:**

1. See the image very carefully before answering a question.
2. Each question can be answered on a scale of 1 to 5.
3. We will define the scores within each scale for each question. **READ them carefully**.

---

**Figure 14:** Instructions for the SEVI-based Human Annotation Task

# I  VALIDATING GEODIV - EXTENDED DETAILS

## I.1  SURVEY DETAILS

We validate the SEVI and VDI components of *GeoDiv* by conducting rigorous human studies, as shown in subsection 4.2 (main paper). For studies conducted on each axes, we utilize the Prolific platform (Prolific, 2024). For the **SEVI component**, we enquire the crowdworkers (hired from 14 different countries, excluding Nigeria and Turkey) about the Affluence and Maintenance in the images shown, on the same scale of 1 to 5 as defined for these two dimensions in Section 3 (main paper). For the VDI scores, we hire 3 crowdworkers per country, totaling 42 participants, and report the Spearman's rank correlation coefficient $\rho$ between the LLM-predicted Affluence and Maintenance scores with the corresponding average human scores. Each annotator is allowed a span of 30 minutes to complete the survey. The instructions for the study, specific to Japan as an example, including a question to assess the Cultural Localization of images (discussed in Appendix L) are shown in Figure 14. A screenshot of instructions can be seen in Figure 15.

For VDI, instead of directly asking for diversity scores, we validate the performance of the VQA model by obtaining answers for a subset of image-question pairs from the crowdworkers, where equal number of questions enquire about the entity and the background respectively. Three crowdworkers are randomly hired for this task, and the overall annotation requires around 45 minutes to complete. In addition to the VQA questions, we ask every user to rate the images on a) their realism (on a Likert-scale of 1 to 5), where a high score denotes high realism, and b) the confidence of the user in answering the question (on a Likert-scale of 1 to 5), where a high score denotes high confidence. The exact instructions for annotation are described in Figure 16. A screenshot of an image and the questions asked for it can be seen in Figure 17.

This image is of the entity: a chair. Rate this image on the level of *affluence* depicted in the *entire* image. The options are a scale of 1 to 5, where each score is defined within the options.

○ 1 – **Impoverished**: Severe visible decay; disrepair, dirt, broken infrastructure, minimal economic activity.

○ 2 – **Low Affluence**: Basic but aging structures; modest upkeep; informal or patchy development visible.

○ 3 – **Moderate Affluence**: Clean and functional spaces; organized but simple environments; middle-income indicators.

○ 4 – **High Affluence**: Well-maintained, vibrant areas; professional storefronts; signs of prosperity and civic care.

○ 5 – **Very High Affluence (Luxury)**: Sleek, modern, or designer elements; upscale brands; spotless, elite environments.

This image is of the entity: a chair. Rate this image on the *general condition* of the *entity*. The options are a scale of 1 to 5, where each score is defined within the options.

○ 1 – **Severely Damaged**: Major disrepair, heavy rust, breakage, or abandonment visible.

○ 2 – **Poor Condition**: Noticeable wear, aging, dirt, minor missing parts, but still recognizable.

○ 3 – **Moderately Maintained**: Functional, intact, but with small flaws like scuffs or fading.

○ 4 – **Well Maintained**: Clean, organized, minor cosmetic wear only, no functional damage.

○ 5 – **Excellent Condition**: Polished, pristine, flawless; appears new or recently serviced.

This image is of the entity: a chair. Rate this image on the *cultural localization* of the *entity* with respect to *your* country. The options are a scale of 1 to 5, where each score is defined within the options.

○ 1 – **Highly globalized**: The subject displays no distinct cultural markers and appears universally generic or global in design.

○ 2 – **Slightly localized**: The subject shows minor cultural hints, but these are subtle and easily overshadowed by global aesthetics.

○ 3 – **Moderately localized**: The subject blends global and cultural elements, suggesting a recognizable yet not dominant cultural identity.

○ 4 – **Strongly localized**: The subject prominently features distinctive cultural elements that are clearly tied to the local context.

○ 5 – **Deeply rooted in culture**: The subject embodies cultural uniqueness through highly characteristic and tradition-rich visual cues.

Next

Figure 15: **Sample questions for the SEVI dimensions, including a question on measuring Cultural Localization** for a given image. For each image-question pair, the scales for each of these dimensions are defined.

**A Study on Image-based Question Answering**

You will be shown a number of images, and each such image will be accompanied by **FOUR questions**. Answer **ALL** questions. **Total time: 45 minutes**
**Instructions:**

1. See the image very carefully before answering a question.

2. Each question will be associated with options.

3. **Multiple options can be correct for the first two questions.**

4. If you do not feel any of the options is correct, select **None of the above**.

5. You can refer to the internet in case you want to know more about certain options.

6. The bottom two questions are **single-options only**.

**Figure 16:** Instructions for the Image-based Question Answering Task

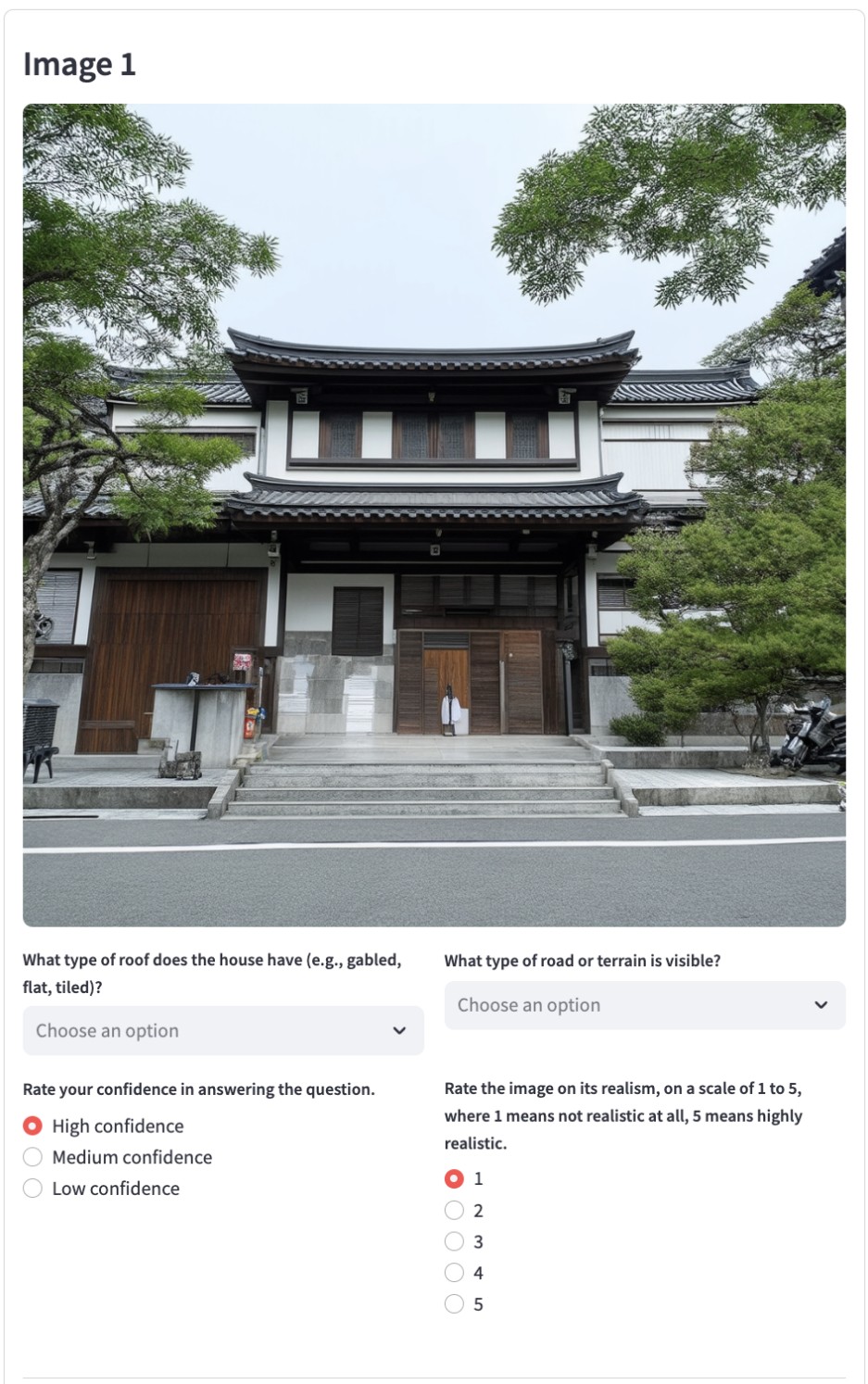

Figure 17: **Sample questions for the VDI-based VQA model Validation.** Along with the VDI questions (one entity and background question for each image), we also ask the users about the Realism of the given image, as well as their confidence in answering the question.

Each crowdworker is paid at a rate of $**8 per hour**.

Table 12: **Country-wise Spearman's Correlation Coefficient between human and model ratings for SEVI dimensions**. `Gemini-2.5` outperforms the open-source variants.

| Country | Qwen2.5-VL | | llava-v1.6 | | Gemini-2.5 | |
|---|---|---|---|---|---|---|
| | Affluence | Maintenance | Affluence | Maintenance | Affluence | Maintenance |
| India | 0.87 | 0.72 | 0.84 | 0.81 | 0.89 | 0.80 |
| China | 0.76 | 0.78 | 0.76 | 0.75 | 0.88 | 0.80 |
| USA | 0.72 | 0.67 | 0.53 | 0.79 | 0.69 | 0.62 |
| Colombia | 0.84 | 0.85 | 0.84 | 0.74 | 0.88 | 0.81 |
| Egypt | 0.66 | 0.86 | 0.73 | 0.76 | 0.62 | 0.58 |
| UAE | 0.82 | 0.89 | 0.81 | 0.67 | 0.69 | 0.83 |
| UK | 0.44 | 0.53 | 0.45 | 0.61 | 0.67 | 0.37 |
| South Korea | 0.54 | 0.70 | 0.75 | 0.71 | 0.66 | 0.62 |
| Mexico | 0.76 | 0.75 | 0.82 | 0.74 | 0.90 | 0.86 |
| Japan | 0.59 | 0.56 | 0.50 | 0.55 | 0.71 | 0.68 |
| Philippines | 0.64 | 0.72 | 0.69 | 0.63 | 0.70 | 0.64 |
| Indonesia | 0.56 | 0.49 | 0.35 | 0.62 | 0.75 | 0.72 |
| Italy | 0.74 | 0.70 | 0.67 | 0.73 | 0.72 | 0.60 |
| Spain | 0.67 | 0.68 | 0.64 | 0.68 | 0.70 | 0.68 |
| Average | 0.69 | 0.71 | 0.65 | 0.68 | 0.76 | 0.69 |

## I.2 COUNTRY-WISE CORRELATION ANALYSIS FOR SEVI SCORES

Expanding on Table 1 (main paper), which shows the SEVI correlations with human ratings for `Qwen2.5-VL`, `llava-v1.6-mistral-7b-hf` and `Gemini-2.5`, we show the country-wise Spearman's correlation coefficient $\rho$ for each model in Table 12. The Affluence and Maintenance rating correlations for `Gemini-2.5` remains similar to the other models for most countries.

## I.3 COMPARISON BETWEEN CLOSED AND OPEN SOURCE MODELS

While our VQA pipeline employs a closed-source model (Gemini 2.5 Flash), it can be substituted with any efficient open-source alternative. In this section, we examine the correlation between the diversity scores produced by Gemini 2.5 Flash and those obtained from Qwen2.5-VL-32B-Instruct-AWQ across the four diversity axes. The analysis is conducted on one synthetic dataset (FLUX.1) for six entities (Bag, Chair, Cooking Pot, House, Plate of Food, and Storefront) spanning all countries considered in the main study. We additionally report the correlation for real-world dataset (GeoDE) as well (see Table 13).

Both the closed and open model shows high agreement across all four diversity axes on both the synthetic (FLUX.1) and real (GeoDE) datasets, indicating broadly consistent scoring behavior. The average correlation across entities and diversity axes is 0.831 for FLUX.1 and 0.826 for GeoDE, respectively.

## I.4 STATISTICAL ROBUSTNESS OF GEODIV

The previous section shows the robustness of GeoDiv to varying models. We further analyse the statistical behaviour of GeoDiv scores across prompt and seed variations.

**Robustness to Varying Image Generation Prompts**. In this work, we evaluate the geo-diversity with respect to 'default (minimal) prompts' to analyse what attribute values the T2I model associates to certain geographies without explicit mention. For this analysis, we try the following prompt variations which have minimal semantic changes to generate *100 images for the USA, Colombia, India and Egypt across the 3 entities (house, chair, stove) using 2 models (SD2.1 and FLUX.1)*.

Table 13: Correlation between Flash and Qwen across all four axes of *GeoDiv* scores.

| Dataset | Entity Name | Affluence | Maintenance | Background | Entity |
|---------|-------------|-----------|-------------|------------|--------|
| FLUX.1 | Bag | 0.924 | 0.316 | 0.412 | 0.921 |
| | Chair | 0.926 | 0.979 | 0.978 | 0.986 |
| | Cooking pot | 0.923 | 0.982 | 0.889 | 0.985 |
| | House | 0.796 | 0.819 | 0.885 | 0.875 |
| | Plate of food | 0.813 | 0.532 | 0.727 | 0.697 |
| | Storefront | 0.955 | 0.836 | 0.910 | 0.887 |
| GeoDE | Bag | 0.931 | 0.874 | 0.379 | 0.811 |
| | Chair | 0.962 | 0.894 | 0.387 | 0.870 |
| | Cooking pot | 0.976 | 0.949 | 0.555 | 0.924 |
| | House | 0.981 | 0.971 | 0.839 | 0.803 |
| | Plate of food | 0.953 | 0.896 | 0.673 | 0.800 |
| | Storefront | 0.945 | 0.921 | 0.708 | 0.812 |

---

**Original Prompt**

*a photo of a <entity> in <country>*

**Prompt Variants**

*1. an image of a <entity> in <country>*
*2. a <entity> in <country>*
*3. a <entity> located in <country>*

---

Table 14: Effect of VLM prompt perturbations on SEVI scores.

| Entity | Dataset | Affluence (Orig) | Affluence (Pert) | Maintenance (Orig) | Maintenance (Pert) |
|--------|---------|------------------|------------------|--------------------|--------------------|
| house | SD2.1 | 0.36 | 0.36 | 0.41 | 0.39 |
| | FLUX.1 | 0.22 | 0.22 | 0.05 | 0.07 |
| chair | SD2.1 | 0.41 | 0.42 | 0.60 | 0.57 |
| | FLUX.1 | 0.63 | 0.62 | 0.20 | 0.23 |

As our original prompt, Variant 1 and Variant 2 are very neutral, with the only difference to our original prompt being that the generated image does not have to be a photo but could also be a drawing or cartoon. Variant 3 additionally uses more sophisticated wording, "located in" instead of "in", potentially preconditioning the models in a specific way. We discuss our observations below:

- *SD2.1 exhibits high rank-consistency among the prompt variations across all four axes*, indicating that its country-level diversity scores are largely insensitive to them. The diversity scores obtained from every prompt variant achieves strong agreement with the scores from the original images, with high overall Spearman correlations at $\rho = 0.80$ (variant 1), $\rho = 0.85$ (variant 2) and $\rho = 0.80$ (variant 3). This shows that the underlying diversity patterns learned by SD2.1 remain stable even when prompt phrasing is slightly altered.

- *FLUX.1 is more sensitive to prompt changes than SD2.1*. The correlations are $\rho = 0.65$ (variant 1), $\rho = 0.80$ (variant 2), and $\rho = 0.45$ (variant 3), which are still significantly high. This observation crucially indicates that different image-generative models exhibit differing levels of sensitivity to prompts.

We observe that the correlation scores for FLUX are most affected by changes along the background axis. Even small modifications to the prompt induce different semantic directions in the diffusion model. For example, the phrase "A photo of" pushes the model toward more realistic and commonly

photographed environments, whereas "an image of" broadens the modality to include stock-image-like compositions, studio setups, or cleaner, more curated scenes.

These shifts are visible in our empirical distributions. For instance, variant 1 images of stove show a marked increase in clean, organized kitchen layouts, consistent with a stock-photo bias triggered by the more generic "image" phrasing. Similarly, for chair, variant 3 increases the frequency of courtyard or tiled surfaces while reducing natural ground textures, suggesting that the word "located" pushes the model to place objects within more explicitly constructed or architectural contexts.

Taken together, these examples illustrate that different variations of the prompt introduce distinct semantic steering behaviours that can subtly shift the generated distributions. To avoid introducing unintended stylistic biases and to remain grounded in realistic depictions, we therefore adopt the most neutral form of the prompt as our standard.

**Robustness to Variation of VQA Prompts.** We perturb the SEVI prompts to the VLM for both affluence and maintenance via GPT. The results in Table 14 show that the scores change negligeably from the original (orig) with the VLM prompt perturbation (pert).

**Robustness to Varying Image Budgets.** To assess the statistical stability of our diversity metric, we evaluated how diversity scores change as a function of the number of generated images. For each image-budget $n \in 10, 50, 100, 150, 200, 250$, we generated three independent samples using different random seeds. For each axis of our metric (affluence, maintenance, entity, and background), we calculate the normalized hill numbers for each country-entity-dataset triplet and take the average across the three seeds. We list our observations below:

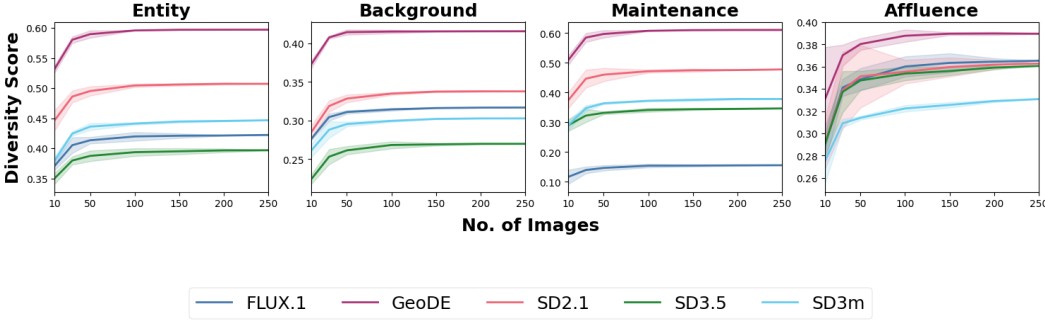

Figure 18: Effect of image budget on GeoDiv estimates. All axes show rapid convergence of diversity scores and stable model ranking, indicating statistical robustness of the metrics.

- *A budget of 100-150 images per concept-country pair is sufficient for stable and reproducible metric estimation.* Across all axes, diversity scores converge smoothly as the number of images increases. Large fluctuations are visible at $< 50$ images, which diminish substantially, and finally become negligible after 100 images. For 150-250 images, confidence intervals are extremely narrow, indicating high reliability (see Figure 18).

- *Consistent Model and Country Ranking Across Image Budgets.* The real-word dataset GeoDE still exhibits the highest diversity scores, and Flux the lowest. This pattern persists across all 4 diversity axes and all values of $n \geq 50$ (see Figure 18), and holds true for the ranking of the studied countries as well (see Figure 19).

- These results suggest that the metric is statistically well-behaved and convergent, suitable for large-scale quantitative evaluation. The width of $95\%$ confidence intervals decreases monotonically with the number of images. This indicates that seed-induced randomness vanishes with larger sample sizes, and the metric's uncertainty is well-behaved and predictable.

**Robustness to Re-runs and Different Seed Image Sets** We rerun the full pipeline three times on the same set of 250 SD3m-generated Indian house images and observe at most a 0.01 standard deviation in the resulting scores (Entity: 0.009, Background: 0.001, Affluence: 0.013, Maintenance: 0.006, overall: 0.007). We additionally generate three independent sets of SD3m images for the same

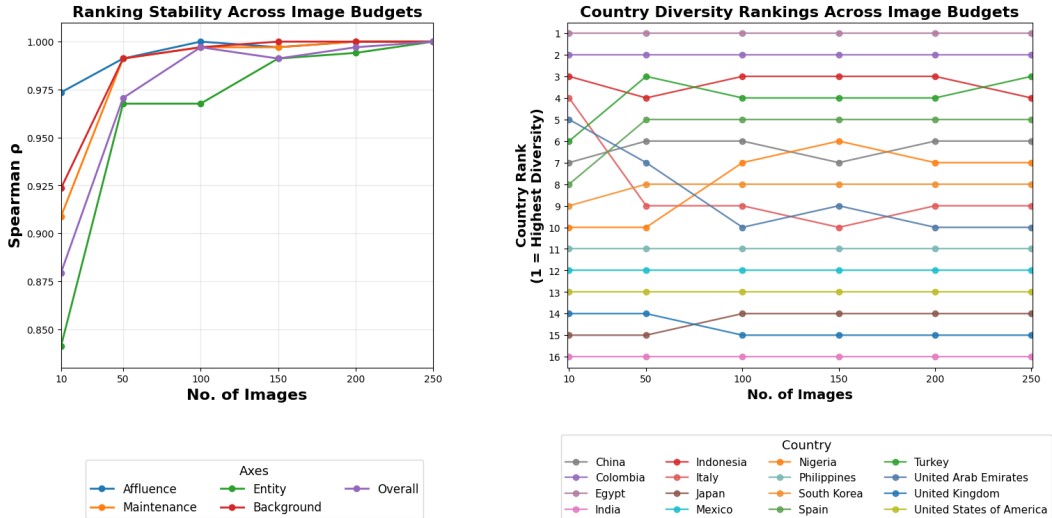

Figure 19: **Left**: Spearman rank correlation between country rankings obtained at each image budget and the 250-image baseline. Across all diversity axes the rankings converge quickly and remain stable once $\geq 100$ images are used. **Right**: Country diversity rankings for the overall GeoDiv score across different image budgets (1 = highest diversity).

entity-country pair using different seeds and find a maximum standard deviation of only 0.05 across all GeoDiv axes (Entity: 0.018, Background: 0.044, Affluence: 0.023, Maintenance: 0.008, overall: 0.023).

## I.5 INTER-ANNOTATOR AGREEMENT ACROSS SEVI AND VDI AXES

Our human validation exhibits strong inter-annotator agreement across both axes, demonstrating the reliability of the collected scores. For the SEVI axis, majority consensus is reached in $85\%$ of the $1,120$ annotated images, and the ordinal consistency is robust: Kendall's $\tau = 0.54$ for Affluence and $0.53$ for Maintenance, with Spearman's $\rho = 0.61$ for both, levels comparable to or exceeding agreement reported in prior work (e.g., Cho et al. [1]). For the VDI axis, annotators show high pairwise agreement, with $87\%$ agreement on entity-diversity and $80\%$ on background-diversity questions. These results indicate substantial annotator consensus and confirm that the human annotations provide a stable and reliable foundation for validating GeoDiv's axes.

## J    QUALITATIVE EXAMPLES

We show examples of **house images of Nigeria**, sampled from each dataset in Figure 20. While GeoDE shows a variety of houses, of different architectures and levels of affluence and maintenance, we can notice a striking lack of diversity in all levels in the generated images. While FLUX.1 images look highly affluent and polished (affluence score: 4.15, maintenance score: 4.99), SD2.1 represents ruralized images of impoverished, ill-maintained houses (affluence score: 1.74, maintenance score: 2.02), and SD3m depict well-maintained houses with consistent bare-earth landscapes (affluence score: 2.18, maintenance score: 3.81). These examples further motivate the need for frameworks that can quantify this lack of diversity within images. *GeoDiv* can quantify geo-diversity on multiple dimensions like affluence, maintenance, background and entity-diversity separately, making it a useful tool that can distinguish among images from datasets, and even entities and countries.

Figure 21 presents a cross-dataset visual comparison of **car images** for Indonesia, an entity–country pair exhibiting the highest cross-country variance in entity diversity scores. GeoDE shows a relatively low entity diversity score (0.49), with real-world images capturing mid-range, commonly used vehicles in typical Indonesian urban contexts. SDv2 yields the highest diversity score for cars (0.850), showcasing a wide range of types, colors, and settings. However, it records the lowest maintenance (2.04) and affluence (1.9) scores for cars in Indonesia, well below the dataset average, frequently depicting rustic, vintage, and even deteriorated vehicles. SD3m exhibits moderate entity diversity (0.714) but very low background diversity (0.303), capturing mostly street-level scenes (urban, paved roads, moderately busy backgrounds) and low contextual variance. FLUX.1 scores lower in entity diversity (0.540), heavily skewed toward polished, high-end SUVs and sedans in modern, affluent-looking neighborhoods, reflecting a synthetic bias toward suburban affluence. The comparative visualization illustrates how real and synthetic datasets differ not only in realism but in the socio-cultural and contextual representation of common entities.

While the UK and USA rank among the lowest on the VDI (Visual Diversity Index), and India and Nigeria score among the lowest on the SEVI (Socio-Economic Visual Indicators), FLUX.1 consistently assigns high scores to all four countries, exceeding 4 on the affluence axis and close to 5 on the maintenance axis. Figure 22 displays FLUX.1's generation of 'houses' across these countries. FLUX.1 consistently generates upscale, multi-storey houses with manicured lawns, porches, and lush green surroundings across all countries. This uniform aesthetic, often resembling Western suburban affluence, reflects a bias toward idealized, high-end housing. As a result, while the images are visually appealing, they lack cultural and structural diversity, demonstrating high affluence but low geo-specific realism.

## House in Nigeria

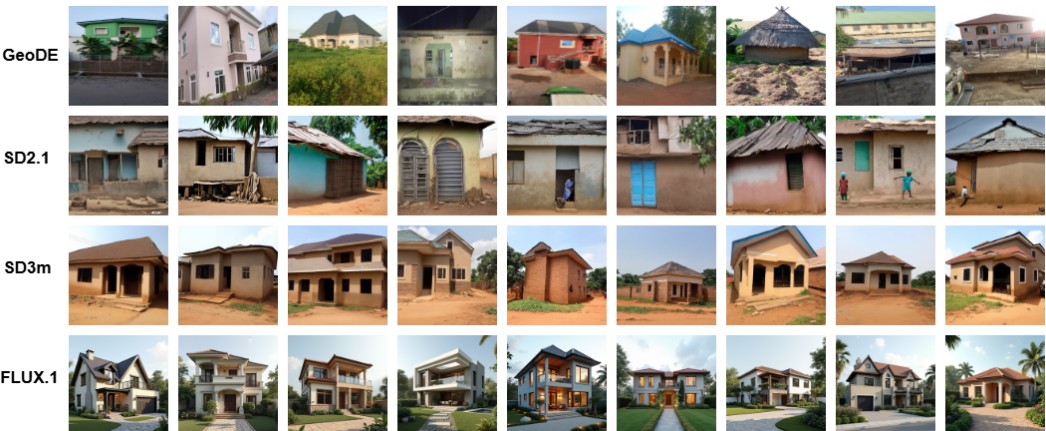

Figure 20: Qualitative examples of house images from Nigeria across datasets. GeoDE shows balanced rural, suburban and urban scenes, while SD2.1 and SD3m show strong rural bias and FLUX.1 shows suburban bias. Each column shares the same generation seed across synthetic models for controlled comparison.

## Car in Indonesia

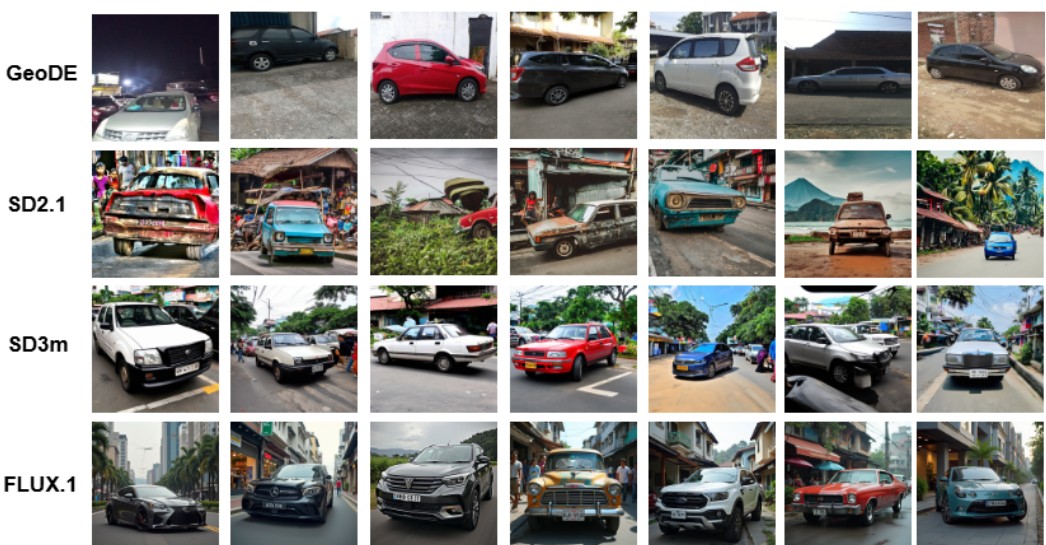

Figure 21: Comparison of car images for Indonesia across datasets. Rows: GeoDE (Entity diversity = 0.49), SD2.1 (0.85), SD3m (0.71), FLUX.1 (0.54). SD2.1 shows highest entity diversity with varied car types and contexts; FLUX.1 skews toward affluent suburban scenes. Indonesia shows the highest cross-country variance (0.03) for the *car* entity.

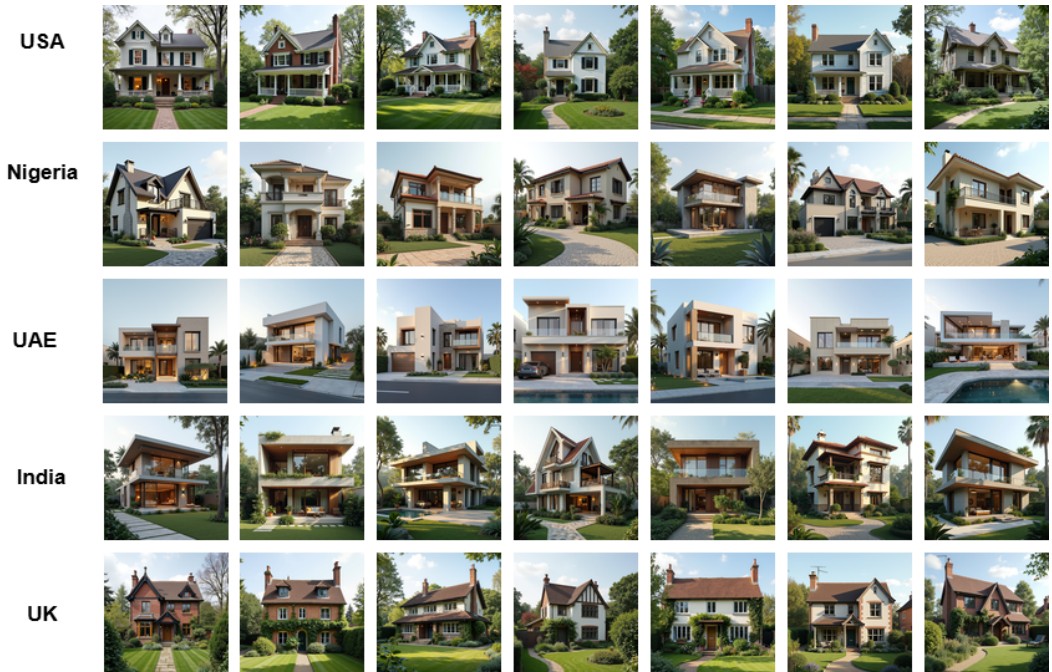

Figure 22: Comparison of house images generated by FLUX.1 across countries.

In Fig. 23, 24, 25 and 26, we provide samples from each of the chosen T2I models, from each of the 10 entities, and 6 selected countries (due to space constraint).

# FLUX.1

Figure 23: **Dataset Samples from the FLUX.1 model.** across 6 countries and 10 entities. We note distinct country-wise features for each image.

Figure 24: **Dataset Samples from the SD2.1 model.** across 6 countries and 10 entities. We note distinct country-wise features for each image.

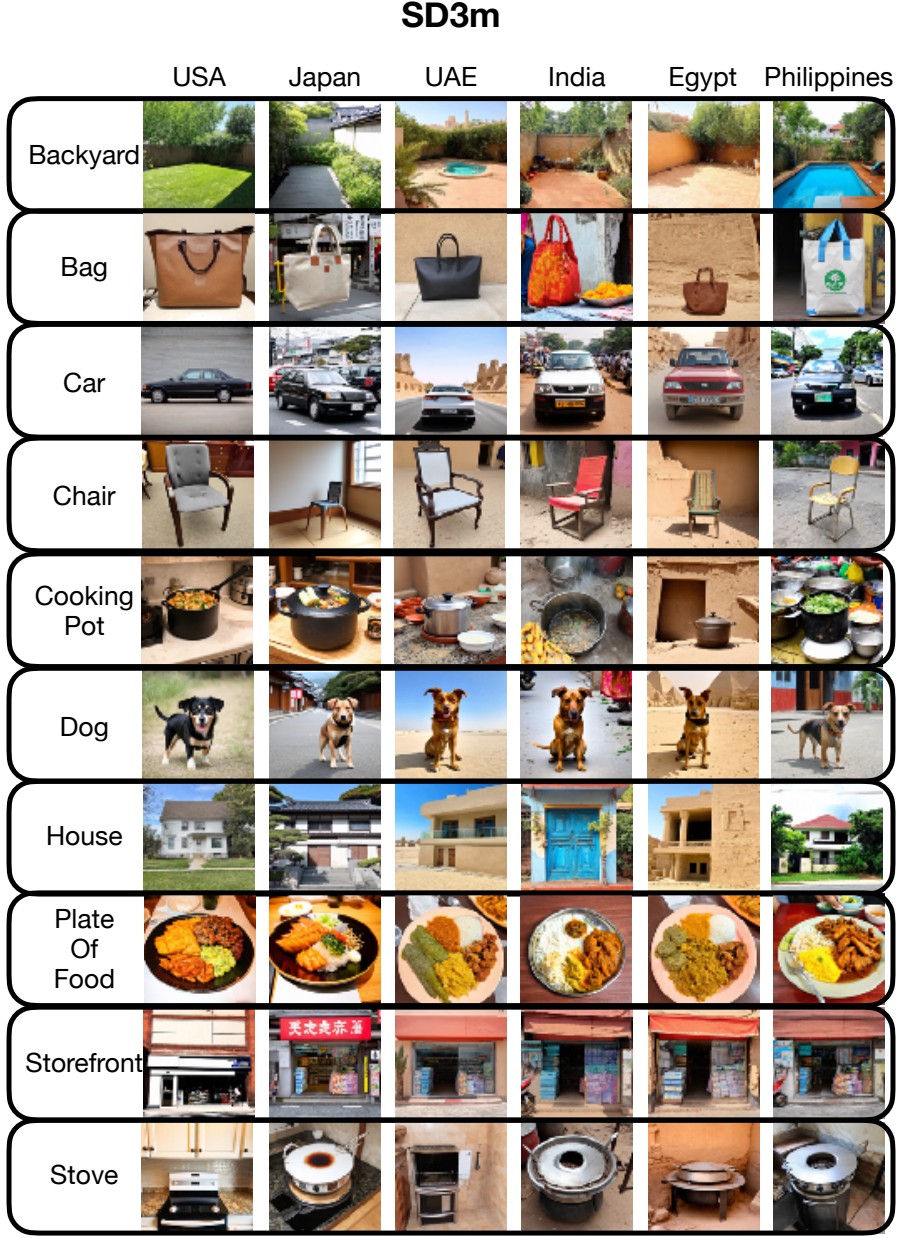

Figure 25: **Dataset Samples from the SD3m model.** across 6 countries and 10 entities. We note distinct country-wise features for each image.

**SD3.5**

Figure 26: **Dataset Samples from the SD3.5 model.** across 6 countries and 10 entities. We note distinct country-wise features for each image.

# K COMPARISON OF GEODIV WITH EXISTING BASELINES - EXTENDED DISCUSSION

## K.1 VENDI-SCORE VS GEODIV SCORES

In the main paper, we analyze the relationship between the proposed VDI metrics and the Vendi-Score (Friedman & Dieng, 2023), a measure of visual diversity within image sets. Specifically, we compute the Pearson correlation between the Vendi-Score and the four aspects of GeoDiv: (a) Entity Diversity, (b) Background Diversity, and (c) Affluence Diversity, (d) Maintenance Diversity. The country-wise correlations, averaged across datasets and entities vary ($\rho = 0.56, 0.23, 0.37$ and $0.06$ respectively, as shown in Table 15). We find a moderate correlation for Entity-Appearance, and weak to very weak correlation for Affluence, Background-Appearance and Maintenance, showing that Vendi-Score focuses mostly on the primary entity, and that our metrics capture aspects of image diversity that go beyond general visual dissimilarity.

Importantly, while Vendi-Score offers a quantitative estimate of diversity, it is non-interpretable, making it difficult to explain why a particular image group receives a high or low score. In contrast, the SEVI and VDI metrics are inherently interpretable: they are grounded in entropy computed from VQA-derived answers to specific semantic questions, allowing for a more transparent understanding of what drives a diversity score.

Table 15: **Pearson's Correlation Coefficient** ($\rho$) between *Vendi-Score* and a) Entity Diversity (Entity-Div), b) Background Diversity (Background-Div), c) Affluence Diversity (Affluence-Div) and d) Maintenance Diversity (Maintenance-Div). Correlations across datasets is very weak, showing that the VDI scores capture features beyond visual diversity.

| Model_name | Entity-Div | Background-Div | Affluence-Div | Maintenance-Div |
|---|---|---|---|---|
| FLUX.1 | 0.59 | 0.03 | 0.11 | 0.20 |
| SD2.1 | 0.63 | 0.42 | 0.41 | 0.14 |
| SD3m | 0.61 | 0.31 | 0.45 | $-0.01$ |
| SD3.5 | 0.43 | 0.18 | 0.51 | $-0.09$ |

## K.2 COMPARISON WITH DIMCIM

Teotia et al. (2025) measure image diversity by querying reliable VQA models on entity attributes and use VQA-Score (Lin et al., 2024) to estimate diversity. However, there are key differences from our GeoDiv approach. First, it ignores geo-diversity, focusing solely on entity-appearance variation. Second, unlike GeoDiv, which collects separate attribute-value sets per entity, DIMCIM uses a fixed set of attributes shared across all entities, making it highly restrictive. Its scores are also less interpretable, often appearing uniform across T2I generations, for example, for the question "*What material is the bag made of?*", DIMCIM assigns a high score (0.8) even if all bags are leather, whereas GeoDiv gives a more realistic score (0.18). Overall, GeoDiv provides a more comprehensive framework for evaluating geo-diversity.

# L CULTURAL LOCALIZATION

The dimensions we measure within the SEVI axis of GeoDiv are: a) Affluence, and b) Maintenance. One other important aspect of Socio-Economic Visual Index is **Cultural Localization**, i.e., the extent of representation of the cultures prevalent in a country in the images generated from the same. We define it on a scale of 1 to 5 in the following way: 1 = Highly Globalized, 2 = Slightly Localized, 3 = Moderately Localized, 4 = Strongly Localized, 5 = Deeply Rooted in Culture. With such definitions of Cultural Localization, we conduct a country-wise human study (see Appendix I.1) on the same, where the models are asked to output a score on a scale of 1 to 5 using the same definitions. The prompt used for computing Cultural Localization is shown below:

```
You are a visual-language model tasked with evaluating how culturally
aligned a generated image of an object is within the context of a
specific region.  You will be provided with the object, the region,
and an image depicting the object situated in that region.
Your task is to assess visual elements, including both the object and
its background features.
Definitions:
        • Cultural Alignment:  visible cues that reflect traditional,
          recognizable features specific to the region's or country's
          identity.
        • Globalization:  cues that are generic, globally common, or lack
          strong ties to local culture.
Your decision must be based only on visible cues present in the image.
Use the following continuous 5-point scale to rate the cultural
alignment:
1  Highly globalized:  No distinct cultural markers;
   universally generic or global in appearance.
2  Slightly localized:  Minor cultural hints are
   present but overshadowed by global aesthetics.
3  Moderately localized:  A mix of global and local
   cues; suggesting a recognizable yet not dominant
   cultural identity.
4  Strongly localized:  Clear and prominent cultural
   elements tied to the local or regional identity.
5  Deeply rooted in culture:  Embodiment of the
   cultural uniqueness through highly characteristic
   and tradition-rich visual cues.
Provide your answer in JSON format:

reasoning_steps: ['Step 1', 'Step 2', ...],
answer: {1, ..., 5}

What is the cultural alignment of the generated image based on
visual cues alone?  Respond only with a single integer between 1
(highly globalized) and 5 (deeply rooted in culture), and provide the
reasoning.
Object:  {entity} Region:  {country} Selection:
```

The average Spearman's rank correlation coefficient $\rho$ across countries (0.41 for `Gemini-2.5`, 0.40 and `Qwen2.5-VL` turns out to be much lesser than those of Affluence and Maintenance. We hypothesize that this happens as the aspect of "Cultural Localization" demands specific knowledge for people residing in each country, and it is often not trivial to rate images on the same due to subjectivity. The only countries for which `Gemini-2.5` has a moderate-to-high positive correlation (i.e, $\rho \geq 0.4$) with the human scores are: India, UK, South Korea, Mexico, Japan and Italy. Across all datasets studied in this paper, we thus assess the Cultural Localization scores of these 6 countries, and find that suprisingly, GeoDE images have a much lower average score (1.87), while SD2.1 images have the highest average score (3.28). SD3m and FLUX.1 images score similarly (2.79 and 2.66). This shows that GeoDE images are relatively more globalized, with less references to country-wise cultures, while the trend is opposite for SD2.1 (as shown in Figure 27).

## M  DATASET DETAILS - EXTENDED DISCUSSIONS

**Choice of Countries.** The countries chosen (USA, UK, India, Japan, Spain, Italy, Mexico, Philippines, Egypt, Nigeria, Colombia, South Korea, China, Indonesia, Turkey and UAE) represent multiple continents like North and South America, Europe, Asia and Africa. They were chosen to understand how differently generative models depict a large spectrum of countries including the US as well as Nigeria, and they have been inspired by previous works that have studied similar countries (Ramaswamy et al., 2023; Basu et al., 2023; Gaviria Rojas et al., 2022; Hall et al., 2023).

**Choice of Entities.** Our selection of entities follows the protocol established in prior studies examining geographical disparities in image datasets (Basu et al., 2023; Hall et al., 2023; 2024).

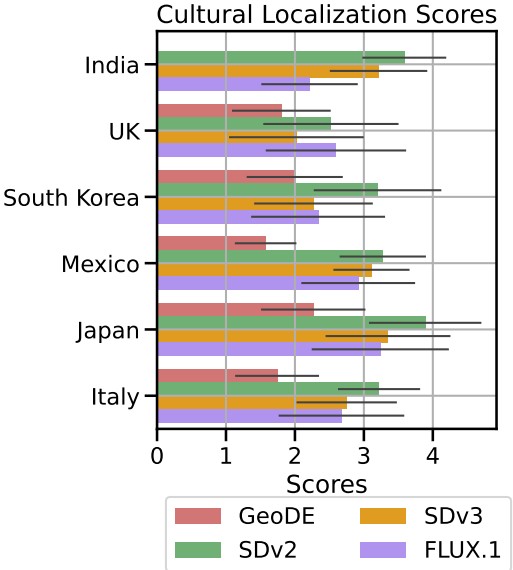

Figure 27: **Assessing Cultural Localization.** In general, we find India, Mexico and Japan to have more culturally localized images per model (including the real-world GeoDE dataset), with SD2.1 (SDv2) achieving higher scores than SD3m (SDv3) and FLUX.1.

Specifically, we adopt all six entities used by Hall et al. (2024)—bag, car, cooking pot, dog, plate of food, and storefront—and supplement these with four additional entities commonly studied in the literature: chair, stove, backyard, and house (Ramaswamy et al., 2023; Gaviria Rojas et al., 2022; Hall et al., 2023; 2024). These ten entities represent everyday objects with wide socio-cultural relevance. Furthermore, GeoDE provides a loose grouping of entities into four categories: Indoor common, Indoor rare, Outdoor common, and Outdoor rare. As shown in Table 2 in the Appendix, our selected entities collectively provide good coverage of all these categories.

# N   BROAD SOCIETAL IMPACT OF GEODIV

Our proposed framework, *GeoDiv*, measures geographic diversity in image datasets by evaluating images of a given entity from different countries. We believe this can positively impact the community by highlighting over- or under-representation of visual attributes across regions. A potential limitation lies in the fixed answer lists generated by the LLM for measuring background and entity diversity as these may not capture the full global spectrum, potentially reinforcing existing biases. To mitigate this, we incorporate a 'None of the Above' option during the VQA stage, allowing the model to flag missing answers specific to certain countries and entities.

