# OpenReview forum: "GeoDiv: Framework for Measuring Geographical Diversity in Text-to-Image Models"
_ICLR.cc/2026/Conference — ICLR 2026 Poster_

### Official Review · Reviewer_UsfU · 2025-10-15

**Soundness:** 4
**Presentation:** 3
**Contribution:** 3
**Rating:** 8
**Confidence:** 3

**Summary:**

This paper introduces GeoDiv, a framework for measuring geographical diversity in Text-to-Image (T2I) models . The authors argue that existing diversity metrics are often uninterpretable or rely on curated datasets, failing to capture nuanced geographical and socio-economic biases. GeoDiv leverages LLMs and VLMs to assess diversity along two primary axes: the Socio-Economic Visual Index (SEVI), which quantifies cues like 'Affluence' and 'Maintenance', and the Visual Diversity Index (VDI), which measures variation in entity and background appearance. Applying this framework to 160,000 images generated by four T2I models across 10 entities and 16 countries, the authors found a consistent lack of diversity and significant socio-economic biases. Notably, depictions of countries such as India, Nigeria, and Colombia were found to be disproportionately impoverished and worn compared to others, demonstrating the framework's utility in identifying specific, interpretable biases.

**Strengths:**

**1. A Novel Evaluation Framework**

The paper introduces GeoDiv, a novel and well-structured framework for the critical task of measuring geographical diversity in text-to-image models. By decomposing the complex concept of "geo-diversity" into two clear and complementary axes, the framework provides a highly interpretable and systematic way to diagnose model biases. This multi-dimensional approach is a significant contribution over existing monolithic or uninterpretable diversity metrics.

**2. A Methodologically Rigorous and Nuanced Definition of Diversity**

A standout strength of this work is its thoughtful and rigorous approach to defining socio-economic diversity. The authors adeptly navigate the paradox of whether to evaluate against real-world inequality or an idealized "fair" world. Their solution, which separately measure the *breadth* of representation (Diversity Score) and the *central tendency* of that representation (Mean Score), is an elegant and powerful way to identify stereotyping without demanding unrealistic portrayals.

**3. Comprehensive Validation and Demonstrated Practical Application**

The paper's claims are supported by a large-scale empirical study and rigorous validation. The authors not only generated a massive dataset of 160,000 images but also conducted extensive human studies to validate that their VLM-based metrics correlate well with human judgments. Crucially, the paper includes a proof-of-concept experiment, demonstrating how the diagnostic insights from GeoDiv can be directly used to guide a simple prompt-based mitigation strategy and measurably improve diversity.

**4. Rich, Granular, and Actionable Insights from the Analysis**

The paper does not just present high-level scores; it provides a rich and granular analysis of the results that yields specific, actionable insights. For example, it uncovers concrete biases like the overrepresentation of "stone" for Egyptian houses and "dirt roads" for Nigeria, and identifies an interesting trade-off in the FLUX.1 model between aesthetic polish and visual diversity. This level of detailed analysis demonstrates the diagnostic power of the GeoDiv framework and provides concrete targets for model improvement.

**5. Clear Writing and High-Quality Presentation**

The paper is well-written and clearly structured, presenting a complex, multi-component framework in an intuitive and easy-to-follow manner. The arguments are logically laid out, and the claims are well-supported by the evidence presented. Furthermore, the figures and tables are of high quality, effectively visualizing the data and making the key findings—such as the country-level SEVI disparities—easy to understand.

**Weaknesses:**

**1. Limited Scope of Evaluated Entities and Countries**

While the study is large in scale, its conclusions are based on a curated set of 10 common entities and 16 countries. These entities are mostly globally common objects, and the countries, while diverse, still represent a fraction of global diversity. The framework's effectiveness and the specific biases it uncovers might not generalize to more culturally-specific entities (e.g., traditional clothing, regional architecture) or to the many countries not included in the analysis.

**2. Attribute Definitions are Constrained by LLM Capabilities**

The Visual Diversity Index (VDI) relies on an ensemble of LLMs to generate the question-answer sets that define the visual attributes for each entity. This approach is fundamentally limited by the collective "imagination," knowledge, and inherent biases of the LLMs used. The framework may fail to generate questions about important, culturally-nuanced visual features that are outside the LLMs' training data distribution, potentially leading to an incomplete or skewed measurement of diversity.

**3. Lack of Reported Inter-Annotator Agreement for Human Validation**

The paper's validation of its core SEVI and VDI metrics relies on human studies, but it fails to report standard Inter-Annotator Agreement (IAA) metrics. While the authors mention using three annotators and resolving disagreements via majority vote or averaging, they do not quantify the level of agreement among these human annotators. This omission makes it difficult to fully assess the consistency and reliability of the human-generated ground truth data used to validate the framework's components.

**Questions:**

**1.** Could the authors discuss the limitations of this VDI approach, particularly the risk that the LLM's own biases and limited knowledge may prevent it from generating a truly comprehensive and culturally aware set of visual attributes?

**2.** The high failure rate of the visibility check for certain questions (e.g., over 50%) is a concern. Could the authors comment on how this aggressive filtering might affect the validity of the final VDI scores? Does removing a large portion of the images for a given question introduce a form of selection bias into the measurement?

**3.** For the human studies used to validate the framework, could the authors please report the IAA scores?

---

> ### Author Response · Authors · 2025-11-20
>
> We thank the reviewer for the thoughtful and positive assessment of our work. We appreciate the recognition of GeoDiv’s novelty, the rigor in defining geographically grounded diversity, and the value of separating breadth and central tendency in our analysis. We are also grateful for the acknowledgement of our large-scale validation, the actionable insights enabled by the framework, and the clarity of the paper’s presentation. We address the reviewer’s questions and concerns below, and all updates in the paper with respect to the reviewer’s comments are presented in **purple color**.
>
> **Limited Scope of Evaluated Entities and Countries**
>
> We have updated this point in the limitations section. The countries are chosen to be diversely spread across the globe, and all the prompts and steps to calculate geo-diversity are clearly listed in the Appendix (section H), along with the hyperparameters, thus enabling future users to extend the study to other countries as well. The entities that we have chosen are global, and the attributes we calculate diversity over are intentionally generic, as LLMs and VLMs have limited cultural knowledge (i.e., they may not recognize traditional clothings from all countries, but recognize if a chair has a single leg or multiple legs).
>
> **Attribute Definitions are Constrained by LLM Capabilities**
>
> The ensemble of LLMs are used to capture as many visually detectable attributes as possible. These attributes are intentionally generic, avoiding cultural nuances. This is because our preliminary analysis finds current LLMs and VLMs to be still deficient in human correlation on cultural localization ($\rho$ is only 0.4, as noted in Appendix M). Our goal in GeoDiv is therefore to measure how model generations vary on basic, universally recognizable attributes across entities and countries, attributes that LLMs/VLMs can reliably identify. However, documenting distributions for all attributes of an entity may not be feasible, and we update this in the limitation section within the revised version.
>
> **Lack of Reported Inter-Annotator Agreement for Human Validation.**
>
> We thank the reviewer for highlighting this important aspect. Our human validation exhibits strong inter-annotator agreement across both axes, demonstrating the reliability of the collected scores. For the SEVI axis, majority consensus is reached in 85% of the 1,120 annotated images, and the ordinal consistency is high: Kendall’s $\tau$ = 0.54 for Affluence and 0.53 for Maintenance, with Spearman’s $\rho$ = 0.61 for both, levels comparable to or exceeding agreement reported in prior work (e.g., Cho et al. [1]). For the VDI axis, annotators show high pairwise agreement, with 87% agreement on entity-diversity and 80% on background-diversity questions. These results indicate substantial annotator consensus and confirm that the human annotations provide a stable and reliable foundation for validating GeoDiv’s axes. We add this point in the updated version in Appendix J.5.
>
> [1] Cho et al., Davidsonian Scene Graph: Improving Reliability in Fine-grained evaluation for Text-to-Image Generation, ICLR 2024
>
> **Could the authors discuss the limitations of this VDI approach, particularly the risk that the LLM's own biases and limited knowledge may prevent it from generating a truly comprehensive and culturally aware set of visual attributes?**
>
> We have discussed these limitations in the current version. Keeping in mind that VLMs and LLMs have their own biases, the attributes have been kept mostly generic. To limit hallucinations, we have added some safety measures: a) including a visibility check to ensure the queried attribute is actually visible in the image, reducing hallucinated responses, b) including a “None of the Above” (NOTA) option in the answer list so that the model can abstain from answering. The high agreement between human and VLM outputs demonstrates the ability of the VLMs to reliably compute VDI diversity. VLMs still struggle with cultural knowledge, making them unsuitable for measuring cultural localization, hence we do not report the same as part of the GeoDiv framework. However, we realize that the attributes are indeed limited by the LLM’s knowledge of the entities, which we have updated in the Limitations section.

---

> > ### Author Response · Authors · 2025-11-20
> > **Official Comment by Authors (Continued)**
> >
> > We continue with our responses here.
> >
> > **The high failure rate of the visibility check for certain questions (e.g., over 50%) is a concern. Could the authors comment on how this aggressive filtering might affect the validity of the final VDI scores? Does removing a large portion of the images for a given question introduce a form of selection bias into the measurement?**
> >
> > The visibility check is applied per question, not per image overall, and is designed to ensure that a VDI score reflects only attributes that are actually observable. Many questions naturally apply to only a subset of images (e.g., “Is the chair upholstered?” is not meaningful if the chair is occluded or only its legs are visible). In such cases, filtering prevents the VLM from hallucinating answers. A single image may be filtered out for one question but still contribute to many others. Importantly, this filtering does not introduce selection bias because the Diversity Score is computed only over the subset of images where the attribute is visually determinable, which aligns with the semantic intent of the metric. This filtering removes unsuitable questions from the analysis and in turn adds to the robustness of our final scores.

---

> ### Comment · Reviewer_UsfU · 2025-11-21
>
> Thanks for the response. Most of my concerns are resolved. Although I still have somewhat unsure in the limitation, I believe it's a good work. I will keep my positive score.

---

> ### Author Response · Authors · 2025-11-23
>
> We thank the reviewer for acknowledging our responses. We are glad that their concerns have been addressed, and we appreciate that they find our work to be strong and maintain their positive score. We have revised the Limitations section to clearly and transparently reflect the points raised, including: (i) the limited set of entities and countries studied, (ii) the reliance on LLM-generated attributes for VDI, (iii) the use of a 1–5 Likert scale for SEVI due to the absence of standardized measures, and (iv) the inherent constraints of current LLMs/VLMs and the explicit steps we take to mitigate them. These updates ensure that the scope and claims of GeoDiv are well-bounded and fully aligned with our discussion. We greatly value the reviewer’s careful evaluation and supportive assessment of our contribution.

---

> ### Author Response · Authors · 2025-12-02
> **Discussion Summary**
>
> We thank the reviewer once again for the thoughtful and positive assessment of our work. We appreciate the recognition of GeoDiv’s novelty, the rigor in defining geographically grounded diversity, and the value of separating breadth and central tendency in our analysis. We are also grateful for the acknowledgement of our large-scale validation, the actionable insights enabled by the framework, and the clarity of the paper’s presentation. We summarize our discussions below:
>
> Changes made to manuscript (in purple color):
> * A note on the number of entities and countries studied is added in the Limitation section, along with the challenge of being dependent on LLMs for attribute and their value generations.
> * Details of Inter-annotator agreement have been added to Appendix J.2
>
> Other clarifications made:
> * Justification of visibility-based filtering
>
> The reviewer maintained their score, mentioning that they believe it to be a **good work**.

---

### Official Review · Reviewer_oxKU · 2025-10-28

**Soundness:** 3
**Presentation:** 4
**Contribution:** 4
**Rating:** 6
**Confidence:** 4

**Summary:**

This paper introduces GeoDiv, an interpretable framework using LLMs and VLMs to measure geographical diversity in text-to-image (T2I) models, addressing limitations of existing metrics. It assesses images along two axes: the Socio-Economic Visual Index (SEVI) for affluence/maintenance, and the Visual Diversity Index (VDI) for entity/background variety. The authors validate GeoDiv against human judgments, confirming its reliability. Applying GeoDiv reveals significant geographical biases in T2I models, such as stereotypical portrayals of certain countries.

**Strengths:**

1. The paper addresses the important and novel problem of quantifying geographical bias in T2I models by decoupling diversity into interpretable socio-economic and visual axes.
2. The proposed GeoDiv framework is rigorously validated against human judgments, building trust in its reliability.
3. The framework produces strong empirical results that highlight specific biases and demonstrate its advantages over existing diversity metrics.
4. The paper is clearly written, well-organized, and effectively presented with helpful figures.

**Weaknesses:**

The paper's main weakness lies in its narrative structure, which could more clearly delineate the problem from the proposed solution. While the paper successfully demonstrates (1) that T2I models exhibit geographical bias and (2) that the GeoDiv framework is a valid VLM-based method to measure this, the presentation intertwines these two major points.

A potentially stronger narrative might be:
1. First, establish the core problem: Demonstrate unequivocally that significant geographical bias exists in T2I model outputs. This could potentially leverage some of the human-annotated scores upfront to ground the problem in human perception, making the need for an automated metric clear.
2. Then, introduce GeoDiv as the solution: Present the framework as the proposed method to systematically and interpretably measure the established bias. The strong correlation between VLM and human scores (Sec 4.2) then serves as the crucial validation for this specific methodological contribution.

This revised structure would create a clearer separation between motivating the problem and validating the proposed tool, potentially strengthening the paper's overall argument by first concluding "T2I image generation is biased (as perceived by humans)" before concluding "and our VLM-based GeoDiv is a reliable way to measure this." The current organization mixes the validation of the tool (correlation) with the findings from the tool (bias results) without this clear sequential flow.

**Questions:**

I have a few questions, primarily related to the calculation and presentation of the Visual Diversity Index (VDI), which seem connected to the paper's organizational structure:
- Placement of VDI Calculation: Section 3.1 introduces VDI and mentions using VQA, but the actual calculation method (Normalized Hill Number, Eq. 1) appears later, after Section 3.2. Could the authors confirm this placement is intentional and perhaps clarify the VDI pipeline within Section 3.1 itself for better flow?
- Role of Q&A Pairs: If the final VDI score is based on the diversity of the VQA model's output distribution (measured by the Hill Number), what is the specific role of generating detailed Question-Answer pairs beforehand? How do these pre-defined pairs contribute to the final diversity score if the score reflects the diversity of the VQA outputs themselves?
- VDI Metric in Table 1: Table 1 lists "Accuracy" under the VDI column. Given VDI is ultimately a 0-1 diversity score, could the authors clarify what this "Accuracy" refers to? Is it measuring the accuracy of the VQA model's intermediate categorical answers against human annotations (as part of the framework validation), rather than the final VDI score itself? Clarifying this distinction would be helpful.


Moreover, one high-level question. Due to GDPR, the open-sourced generative models might not be trained on EU countries' images. Do you think this can be a potential reason why "European countries (Italy, Spain, UK) exhibit more uniform contexts"?

---

> ### Author Response · Authors · 2025-11-20
>
> We thank the reviewer for recognizing the importance and novelty of quantifying geographical diversity in T2I models, and for highlighting the rigor of our validation, the strength of the empirical findings, and the clarity of the presentation. We address the reviewer’s concerns below. As per the reviewer's suggestion, we have also updated the paper in **blue color**.
>
> **The paper's main weakness lies in its narrative structure, which could more clearly delineate the problem from the proposed solution...**
>
> We completely agree with the reviewer, and thank them for the thoughtful suggestions. We have updated the paper (particularly Section 1 (Introduction and Figure 1) and Section 3 (The Proposed Framework: GeoDiv) to separate the motivation from the proposed approach in blue color, and request the reviewer to check the relevant sections of the updated paper. We are open to further suggestions, and will change our presentation accordingly to improve the quality of the manuscript.
>
> **Placement of VDI Calculation: Section 3.1 introduces VDI and mentions using VQA, but the actual calculation method (Normalized Hill Number, Eq. 1) appears later, after Section 3.2. Could the authors confirm this placement is intentional and perhaps clarify the VDI pipeline within Section 3.1 itself for better flow?**
>
> We thank the reviewer for pointing this out. In Section 3, we first define what constitutes geographical diversity: VDI in Section 3.1 and SEVI in Section 3.2. It is to be noted that the method of measuring diversity is common across both axes. The Normalized Hill Number (Eq. 1) therefore appears after Section 3.2 because it applies uniformly to both VDI and SEVI.
>
> We agree that this structure could be clearer. In the revised version, we introduce a new subsection that explicitly presents the diversity computation (Normalized Hill Number). This reordering makes the flow more intuitive.
>
> **Role of Q&A Pairs: If the final VDI score is based on the diversity of the VQA model's output distribution (measured by the Hill Number), what is the specific role of generating detailed Question-Answer pairs beforehand? How do these pre-defined pairs contribute to the final diversity score if the score reflects the diversity of the VQA outputs themselves?**
>
> We apologize for any confusion that may have led to this misunderstanding. We would like to clarify that for the VDI axis, the LLM-generated Question-Answer (Q&A) pairs define the attribute space over which diversity is measured. Each question corresponds to an entity or background attribute, and the LLM-generated answer set specifies the possible values that attribute can take. For each image, the VQA model is asked these questions with the answer options explicitly provided, and the final diversity score is computed from the frequency distribution of these answers over all images for a given (model, entity, country) combination. To minimize inter-LLM variation in the generated answers, we design questions around generic, visually detectable attributes whose values are typically binary or narrowly scoped (e.g., “Does the chair have multiple distinct legs or a single central base?”). A “None of the Above” (NOTA) option is included to allow abstention and reduce hallucinations. The resulting diversity scores and corresponding observations are therefore grounded in this curated, consistent answer space, which already reveals substantial skewness across geographies, reflecting limited variability in the generated images. We have added this with more clarity to the revised manuscript.
>
> **VDI Metric in Table 1: Table 1 lists "Accuracy" under the VDI column. Given VDI is ultimately a 0-1 diversity score, could the authors clarify what this "Accuracy" refers to? Is it measuring the accuracy of the VQA model's intermediate categorical answers against human annotations (as part of the framework validation), rather than the final VDI score itself? Clarifying this distinction would be helpful.**
>
> We thank the reviewer for pointing this out. The “Accuracy” reported in Table 1 refers not to the VDI diversity-score itself, but to the *accuracy of the VQA model’s categorical predictions* given a set of question-image pairs, when compared against human annotations during the validation study. This validates that the VLM reliably identifies the visual attributes that VDI later aggregates into diversity distributions.
> To avoid confusion, we clarify this distinction in the revised manuscript:
> * *VDI* is measured using the 0-1 diversity score computed via the Normalized Hill Number.
> * *Accuracy* in Table 1 evaluates the VQA model’s correctness on attribute labels, which serves to validate the reliability of the evaluator and not the diversity metric.
>
> We have updated the table caption and the corresponding text in Section 4.2 to make this distinction explicit.

---

> > ### Author Response · Authors · 2025-11-20
> > **Official Comment by Authors (Continued)**
> >
> > We continue with our responses here.
> >
> > **Moreover, one high-level question. Due to GDPR, the open-sourced generative models might not be trained on EU countries' images. Do you think this can be a potential reason why "European countries (Italy, Spain, UK) exhibit more uniform contexts"?**
> >
> > We thank the reviewer for this thoughtful question. GDPR regulates the use of personal data, i.e., information relating to identifiable individuals, and does not restrict the use of non-personal imagery such as buildings, streets, landscapes, or generic daily objects. Since GDPR became enforceable in 2018, most publicly available large-scale image datasets (including those used to pretrain open-source generative models) were collected well before this period or consist primarily of non-identifiable content. Therefore, we do not expect GDPR to meaningfully reduce the presence of European countries in the training data for these models, especially given that our analysis focuses on *non-personal, everyday entities* (houses, chairs, stoves), which fall entirely outside GDPR restrictions.
> >
> > Instead, our findings suggest that the relative uniformity observed in European countries arises from model-internal priors about what “typical” European environments look like (e.g., stucco or brick façades, tiled roofs, paved surroundings), which are globally common and stylistically homogeneous. This parallels the biases we observe in portrayals of less affluent regions (e.g., dusty ground textures in Nigeria), and is more consistent with *representation biases in the training distribution than with regulatory constraints*.

---

> > > ### Comment · Reviewer_oxKU · 2025-11-22
> > >
> > > Thanks for the comprehensive revision and comments, especially the analysis related to GDPR. Since all my concerns have been addressed, I have updated my rating to 8.
> > >
> > > **Justification for the rating**: This paper, GeoDiv, is primarily a large-scale benchmark contribution rather than a methodology aimed at mitigating a specific issue. It introduces a new dataset (including its generation process) and proposes new evaluation metrics. These contributions are novel and well aligned with the “Datasets and Benchmarks” topic in the ICLR Call for Papers.

---

> > > > ### Author Response · Authors · 2025-11-23
> > > >
> > > > Thank you very much for your thoughtful follow-up and for revisiting your score. We are grateful for your feedback, which has greatly improved the clarity of our manuscript, and we are glad that the revisions successfully addressed your concerns.

---

> ### Author Response · Authors · 2025-12-02
> **Discussion Summary**
>
> We thank the reviewer once again for the constructive feedback. We appreciate the recognition of the importance and lack of research in assessing geo-diversity in generative models and the value of our large-scale benchmark. We summarize our discussions below for the ease of the AC's review.
>
> Changes made to manuscript (in blue color):
> * We restructured the Introduction, including Figure 1, to first exhibit that geographical bias exists, followed by presenting GeoDiv as a reliable method to quantify this. Similar changes can be observed in Section 3, with the addition of subsection 3.3 to define the diversity metric.
> * Table 1 and Section 4.2 have been updated to clarify accuracy related to human annotations
>
> Other clarifications made:
> * Clarified role of Q&A pairs in estimating the attribute value frequencies
> * Discussion on relevance of GDPR on generations of European countries
>
> The reviewer confirmed that all concerns were resolved and **explicitly highlighted the novelty of the contribution**.

---

### Official Review · Reviewer_WAhk · 2025-11-02

**Soundness:** 2
**Presentation:** 3
**Contribution:** 2
**Rating:** 2
**Confidence:** 5

**Summary:**

The paper investigates biases that arise from geographic diversity in text-to-image generative models. The authors propose GeoDiv, a framework designed to assess geographic representation through two complementary metrics: the Socio-Economic Visual Index (SEVI) and the Visual Diversity Index (VDI). Experiments are conducted on multiple diffusion models, including Stable Diffusion (v2.1, v3, v3.5) and FLUX.1-dev.

**Strengths:**

- The paper addresses an important issue of geographical diversity and socio-economic bias in text-to-image models, advancing beyond traditional demographic or visual diversity metrics toward a more region-aware fairness perspective.
- The study builds a large-scale benchmark comprising 160,000 generated images across 16 countries, 10 object categories, and 4 text-to-image models, enabling a comprehensive analysis of model-, country-, and attribute-level biases, further validated through strong correlations with human ratings (ρ≈0.7–0.8 for SEVI).

**Weaknesses:**

- Novelty:
   - The framework’s core methodology lacks novelty. The idea heavily relies on prompt-based probing and LLM/VLM-assisted scoring, which have been explored in prior works such as [1] and [2], making the contribution more of an application to a new bias dimension rather than a methodological advance.
   - The proposed indices (SEVI and VDI) add interpretability but do not introduce fundamentally new techniques beyond existing entropy- or attribute-based bias quantification methods. Probing variations using LLM and VQA have been extensively explored in works like [1] and [2].

- The framework’s reliance on closed-source evaluators such as Gemini-2.5 and Qwen2.5-VL introduces potential evaluator bias and limits reproducibility, while its high computational and monetary cost (~$68 per entity–country pair) further constrains accessibility and scalability for large-scale or continuous audits.


[1] TIBET: Identifying and Evaluating Biases in Text-to-Image Generative Models

[2] Generated Bias: Auditing Internal Bias Dynamics of Text-to-Image Generative Models

**Questions:**

- L160: " Manually defining a comprehensive set of attributes for each entity is infeasible, so we leverage multiple LLMs to generate candidate question-answer sets, and consolidate them into a unified list using a neutral LLM." What impact does using multiple LLMs have compared to a single model, and what is the associated computational overhead?

- L216: "Consider a question qk (related to either SEVI or VDI attributes), having a set of possible answers denoted by Ak."How is the answer set Ak defined and quantified for each question? Since the full space of possible answers is likely intractable, are you instead working with a sampled or limited subset of Ak? If so, how is the quality or representativeness of this subset ensured?

- L27: "where Pk is the answer distribution for qk, and H(·) denotes Shannon entropy." How exactly is Pk defined in this context? Does it represent a probability distribution over all possible answers or over token-level probabilities produced by the model? If it is over answers, how is this distribution estimated or normalized, given the discrete and potentially open-ended nature of responses?

---

> ### Author Response · Authors · 2025-11-20
>
> We thank the reviewer for the constructive feedback. We appreciate the reviewer’s recognition of the importance of studying geographical diversity and socio-economic bias, as well as the value of the large-scale benchmark. We address the raised concerns below. Based on the reviewer’s suggestions, we have also updated the paper (in **orange color**).
>
> **The framework’s core methodology lacks novelty. The idea heavily relies on prompt-based probing and LLM/VLM-assisted scoring, which have been explored in prior works such as [1] and [2]...**
>
> We thank the reviewer for highlighting papers [1, 2], which we have now included in the related works section. However, our contributions are fundamentally different. Although LLM/VLM-based scoring has become common, our novelty lies in the task itself: **measuring geo-diversity**, an under-explored dimension that prior work approaches only through *reference datasets with limited coverage*. Unlike [1], which studies demographic or semantic biases through *counterfactual prompt generations*, our objective is to quantify the within-country visual diversity present in the images generated for a fixed entity-country pair. Counterfactual generation, besides being substantially more computationally expensive, is not suited to capturing this type of variation. Our VDI axis models entity appearance and background appearance using dynamically generated, entity-specific and background specific visual attributes - landscape, environment, materials, which [1] and [2] do not explicitly explore. Our SEVI axis further introduces nuanced visual levels of affluence and maintenance, going well beyond the coarse socio-economic descriptors in [1]. Finally, our Hill-Number-based metric measures the distribution of these visual attributes across images, enabling a reference-free and interpretable quantification of visual and socio-economic diversity. Together, these distinctions establish GeoDiv as the first principled framework for systematically measuring geographical and socio-economic diversity in generative models.
>
> **The proposed indices (SEVI and VDI) add interpretability but do not introduce fundamentally new techniques beyond existing entropy- or attribute-based bias quantification methods. ...**
>
> The cited works [1, 2] quantify bias through counterfactual generations or representation shifts. Our task is fundamentally different: measuring geo-diversity in images generated from default entity-country prompts. GeoDiv introduces an interpretable metric in the form of Normalized Hill Numbers (Eq. 1), quantifying the effective number of visual attributes per question that are well-represented in the generations, which is necessary as geographical variation spans intertwined cues such as appearance, environment, affluence, and infrastructure; rather than discrete categories like gender [2]. GeoDiv’s diversity formulation is the first to systematically capture this multi-dimensional variation, opening a direction for broader investigation in this under-explored area.
>
> The SEVI axis is novel: we define multiple levels of Affluence and Maintenance, spanning a broad spectrum from poverty to luxury, use VLMs to categorize images along these levels, and validate them via a comprehensive human study (ρ $\approx$ 0.7). The VDI axis emphasizes the entity and background appearance, recognizing the importance of both in measuring geo-diversity. These contributions establish GeoDiv as a conceptual and quantitative advance for auditing geo-diversity in generative models.
>
> **The framework’s reliance on closed-source evaluators such as Gemini-2.5 and Qwen2.5-VL introduces potential evaluator bias and limits reproducibility...**
>
> We respectfully point out that Qwen-2.5-VL is **actually open source**. Our framework uses closed-source models such as Gemini-2.5 due to their strong, reliable performance, similar to [a, 1]. Qwen performs comparably to Gemini-2.5 in our human studies (Table 1). A detailed comparison of Gemini vs. Qwen (Appendix J.3) shows an average correlation of 0.83 between their diversity scores, indicating that GeoDiv’s results are not tied to any specific VLM. We use Gemini only for its marginal performance advantage (clearly articulated in the Limitations section), but our framework is fully modular and any VLM can be substituted, with our human-annotated SEVI and VDI axes providing a stable benchmark. All prompts are provided in Appendix H (hyperparameters added in Appendix H.1), ensuring reproducibility. Finally, while we report computational cost for transparency, it does not restrict the framework’s applicability; future audits can rely entirely on open-source models at zero cost, especially since our questions target generic, visually verifiable attributes.
>
> [a] Rassin et al., ‘GRADE: Quantifying Sample Diversity in Text-to-Image Models’, arxiv 2024.

---

> > ### Author Response · Authors · 2025-11-20
> > **Official Comment by Authors (Continued)**
> >
> > We continue with our responses here.
> >
> > **Q1: What impact does using multiple LLMs have compared to a single model, and what is the associated computational overhead**
> >
> > On average, 6 questions are generated by a single LLM for each entity. After ensembling, the count is close to 11 per entity. Manual inspection shows that most questions overlap across models, and each LLM contributes roughly **one additional unique question**, which gets added to the combined set. These candidates are then consolidated using an aggregator LLM, which combines semantically equivalent ones to ensure robust coverage regardless of model choice. For example, using Gemini and GPT for entity “bag”, the only unique questions are **What type of bag is it (e.g., backpack, handbag, tote bag)?** (Gemini) and **Is there a brand logo or label visible on the bag?** (GPT-4), whereas all others are semantically equivalent.
> >
> > As for complexity, the question generation, filtering and collation is a one time task, and the final set of questions is reused across all experiments. Thus, this step has negligible computational costs. We will release the same for the entities considered in this work to avoid repetitions. However, here is an estimate of the tokens utilized in this step:
> > * *Question Generation*: For Gemini 2.5 Pro, question generation for each entity required ~230 input tokens and produced ~144 output tokens plus ~1,300 thinking tokens. Total usage: approx. 2,300 input tokens and 14,530 output tokens (including thinking), costing $0.15.
> > * *Filtering and Collation*: We use Claude as the aggregator LLM. Across all our trial and final runs for all 10 entities, we consumed a total of ~35k tokens, costing ~$2.6.
> >
> > In cases where this cost is prohibitive, one can use an ensemble of open-source LLMs, as in our experiments they generate questions of comparable detail and coverage. For the entities studied in the paper, we will release the consolidated question sets to aid future researchers.
> >
> > **Q2: How is the answer set Ak defined and quantified for each question?**
> >
> > For the SEVI axis, the answer set $\mathcal{A}_k$ is *directly defined by the five levels of affluence and maintenance described in Sec. 3.2*. For VDI, $\mathcal{A}_k$​ is *automatically constructed using an LLM*, which first proposes candidate answers and then refines them; these curated options are provided to the VLM as multiple-choice responses to prevent free-form generation. To minimize inter-LLM variation in generating the answers, we design questions to focus on generic, visually detectable attributes, whose answers are typically binary or self-contained (e.g., for the question “Does the chair have multiple distinct legs or a single central base?”, the set of possible answers is [“multiple distinct legs” and “single central base”]). To account for missing answers, we include a “None of the Above” (NOTA) option, allowing the model to abstain when none of the answer choices are appropriate. This also reduces hallucinations (as noted in [a], models hallucinate when forced to compulsorily “guess” over acknowledging “uncertainty”). In entity-appearance diversity for instance, NOTA is observed in only 2.6% of model responses versus 2% for humans which indicates adequate coverage. When NOTA exceeds a threshold, it is treated as an additional category in the diversity calculation. We acknowledge that $\mathcal{A}_k$​ is an *approximate, LLM-sampled subset* of the full (intractable) answer space; in the revised version, we explicitly denote this as $\hat{\mathcal{A}_k}$​. The curated answer sets already reveal substantial skewness in VLM outputs, reflecting limited diversity in the generated images even on this approximate set​.
> >
> > [a] Kalai et al., ‘Why Language Models Hallucinate’, arxiv 2025
> >
> > **Q3: How exactly is Pk defined in this context? ...**
> >
> > $P_k$​ (which we revise to $\hat{P_k}$ in the updated version) is defined as the empirical answer distribution over the curated answer set $\hat{\mathcal{A}_k}$ (Sec 3.3 in the revised version). For each question $q_k$, we collect the VLM-predicted answers across all images generated for a given (country, entity, model) combination, and compute the normalized frequency of each answer in $\hat{\mathcal{A}_k}$. Thus, $\hat{P_k}$ is a discrete probability distribution over answers, not token-level probabilities, and is estimated directly from observed answer counts.
> >
> > The diversity score (Eq. 1) therefore quantifies the effective number of distinct answers expressed by the model for a given attribute, normalized by $|\hat{\mathcal{A}_k}|$. This yields an interpretable measure of how many attribute values are meaningfully represented in the generated samples, allowing model and data curators to identify systematically underrepresented visual characteristics across regions.

---

> > > ### Author Response · Authors · 2025-11-25
> > >
> > > Dear Reviewer WAhk,
> > >
> > > With only a week remaining for the discussion phase, we kindly invite you to review our rebuttal and share any further feedback or questions you may have.
> > >
> > > We look forward to the discussion!
> > >
> > > Thank you,
> > >
> > > Authors

---

> ### Author Response · Authors · 2025-12-02
> **Discussion Summary**
>
> We thank the reviewer once again for the constructive feedback. We appreciate the reviewer’s recognition of the importance of studying geographical diversity and socio-economic bias, as well as the value of the large-scale benchmark. We summarize our responses below for the ease of the AC's review:
>
> Changes made to manuscript (in orange color):
> * The referred papers have been included in the Related Work section
> * The notations of $\mathcal{A}_k$ have been revised to denote the LLM generated approximate answers $\hat{\mathcal{A}_k}$ and their corresponding distributions (Section 3.3)
> * Hyperparameter details of the LLM-VLM have been updated in Appendix H.1
>
> Other clarifications made:
> * The reviewer questioned our novelty and the use of closed-source models; we clarified that we are the first to systematically decompose geographical diversity into visual and socio-economic axes, and to use LLM/VLM world-knowledge to quantify them – substantive advances over prior demographic-bias work. We also explained how the cited works are unsuitable for measuring the nuances of geographical diversity.
> * We also showed the framework is modular and works with open-source models like Qwen (Appendix J.3), while using Gemini only for marginally better performance.
> * We have also answered questions regarding our method, including
>   * Impact of single LLM vs an ensemble for Question generation and associated computational cost
>   * How answer set $\mathcal{A}_k$ is defined and quantified
>   * How the distributions for the answer sets are calculated

---

### Official Review · Reviewer_oi9r · 2025-11-04

**Soundness:** 2
**Presentation:** 3
**Contribution:** 2
**Rating:** 4
**Confidence:** 5

**Summary:**

The paper presents GeoDiv to measure geographical diversity in text-to-image (T2I). It is driven by LLM/VLM and focusses on (a) SEVI (Socio-Economic Visual Index), and (b) VDI (Visual Diversity Index). The paper reports (1) systematic geographic/socioeconomic biases (e.g., India, Nigeria, Colombia show as poor), (2) reduced diversity in entities and backgrounds for newer models. These conclusions are arrived from 160k images from four diffusion models across 16 countries and 10 entities.

Overall, while the paper seeks to address an important problem, concerns with what exactly is being measured, and lack of statistical rigor significantly reduces the reliability of the paper's findings.

**Strengths:**

- Auditing geo-diversity of generative models is important, but has not received sufficient attention, apart from a few research publications and media articles.

- Analysis was performed on a large-scale set of generated images. The dataset involving 160k synthetic images across 16 countries × 10 entities, prompts, QA sets, and annotations could be valuable to the community.

- Human validation with reasonable Spearman correlation (about 0.7) for the evaluation.

**Weaknesses:**

- The paper mentions bias and stereotypes interchangeably. However, these two concepts are different from each other as noted by OASIS. And, in the context of generative models, we are largely interested in stereotypes. Furthermore, measuring stereotypes depends on the baseline diversity of concepts in the real world, and deviations from this baseline. However, the current paper does not take this baseline into account, so it is unclear what exactly is being measured, and if the conclusions are meaningful.

- There is heavy use of LLM/VLM to both define and measure attributes. The model's predictions themselves are potentially biased. These biases are not evaluated or accounted for.

- There is heavy use of LLM/VLM to both define and measure attributes. The paper provides point estimates of quantified metrics, and it not clear how robust or reliable these are to slight variations of the proposed approach. For example, different models (partially addressed), different prompts, number of images per attribute used for analysis, etc.

- While the proposed SEVI and VDI are interpretable metrics, they lack statistical grounding.

**Questions:**

- There have been studies into geographical stereotypes in generative models with similar conclusions (e.g., see [a] below). Could the authors clarify the contribution of the paper beyond this?

[a] https://restofworld.org/2023/ai-image-stereotypes

---

> ### Author Response · Authors · 2025-11-20
>
> We thank the reviewer for the constructive feedback. We appreciate the recognition of the importance and lack of research on assessing geo-diversity in generative models and the value of our large-scale benchmark. We address the reviewer’s concerns below. We also update the paper based on the reviewer’s suggestions, using **red color**.
>
> **The paper mentions bias and stereotypes interchangeably... it is unclear what exactly is being measured, and if the conclusions are meaningful.**
>
> We thank the reviewer for raising this important conceptual distinction. GeoDiv does not attempt to measure stereotypes or biases directly. It measures *diversity*, the breadth of visual variation a T2I model exhibits for a given (entity, country, model) combination, capturing how many distinct visual appearances the model produces along the defined dimensions: entity and background appearance, affluence, and maintenance. Any biases discussed in the paper arise only as interpretive outcomes of this analysis. As clarified in OASIS, bias refers to patterns observed directly in a set of images, while stereotypes refer to mismatches relative to real-world distributions. In our setting, real-world distributions of fine-grained visual attributes (such as road types or maintenance levels) are extremely difficult to obtain at scale, so we do not attempt to infer stereotypes. Findings like the predominance of brick houses in the UK or impoverished portrayals of Nigeria reflect relative overrepresentation within generated samples, not deviations from known ground-truth frequencies. Since our framework’s goal is to measure within-model diversity, we simplify terminology in the revised version by referring to these findings as “biases” (imbalances in model generations) rather than “stereotypes” unless explicit real-world comparisons are available. This clarification aligns the paper with OASIS definitions and avoids conflating the two concepts.
>
> **There is heavy use of LLM/VLM to both define and measure attributes. The model's predictions themselves are potentially biased. These biases are not evaluated or accounted for.**
>
> We explicitly discuss VLM/LLM biases in the limitations section. Our design choices aim to minimize their impact on GeoDiv and are discussed below.
>
> * We conduct large-scale human studies for both SEVI and VDI. The high VDI accuracy (86%) and strong SEVI correlations (country-wise average ρ = 0.72) show that the VLMs reliably perform the underlying VQA tasks from which the diversity metrics are derived.
> * For VDI, we deliberately avoid region-specific question generation to prevent introducing LLM-driven biases. Instead, we generate questions from a *global perspective with generic, visually grounded attributes* (e.g., presence of balcony, roof shape) to avoid reliance on culturally nuanced cues, which current VLMs do not model well (we observe low human correlation ≈ 0.4 for cultural grounding). Measuring cultural diversity is therefore outside the scope of this work and noted as a limitation.
> * Attribute definitions are generated using an ensemble of LLMs to capture entity-specific variations, but we acknowledge that no automated method can exhaustively enumerate all attributes. We include this issue as a limitation in the revised version.
> * Importantly, when evaluating images for diversity calculation, the VLM is never provided with the country label, preventing country-conditioned biases from influencing predictions.
> * To control VLM hallucinations, we use (a) Visibility Filtering to ensure the queried attribute is visible, reducing hallucinations, (b) Multi-Select Responses to allow multiple valid answers and avoid distortions from forced single-choice formats, and (c) None of the Above (NoTA), which lets the model abstain when no option fits, reducing hallucinations as discussed in [1].
>
> [1] Kalai et al., ‘Why Language Models Hallucinate’, arxiv 2025
>
> **There have been studies into geographical stereotypes in generative models with similar conclusions (e.g., see [a] below). Could the authors clarify the contribution of the paper beyond this?**
>
> Prior work has *qualitatively* discussed geographical stereotypes in generative models (e.g., [a]). Our work instead provides a systematic, quantitative, and interpretable framework for evaluating geo-diversity without relying on any reference dataset, making the task substantially more challenging. GeoDiv formally defines measurable dimensions: socio-economic indicators, background context, and entity-specific visual features, and offers a principled methodology to assess diversity along each of them. While [a] documents the presence of such issues, it does not provide a means to quantify them. GeoDiv fills this gap by enabling fine-grained, automated evaluation of how different attribute values are represented for each entity–country pair, yielding actionable insights for building more geographically inclusive generative models independent of curated real-world datasets.

---

> > ### Author Response · Authors · 2025-11-20
> > **Official Comment by Authors (Continued)**
> >
> > We continue our responses here.
> >
> > **There is heavy use of LLM/VLM to both define and measure attributes. The paper provides point estimates of quantified metrics, and it is not clear how robust or reliable these are to slight variations of the proposed approach...**
> >
> > We examine each of the suggested variants of the proposed approach below.
> >
> > **Robustness to underlying model:** Our human study shows highly similar accuracies across multiple VLMs (Gemini, GPT-4, Qwen), indicating agreement on the underlying attribute definitions (Table 1). To further test robustness, we repeat our evaluation with Qwen across six entities and two datasets (synthetic and real). Appendix J.3 reports a high correlation with Gemini (ρ = 0.83), suggesting that GeoDiv metrics remain stable across VLMs with sufficiently strong visual reasoning.
> >
> > **Robustness to the number of images:** For each (entity, country) pair, we originally used 250 images. We re-evaluate the metrics using three random subsets of size 10, 50, 100, 150, and 200. A budget of 100-150 images is sufficient for stable and reproducible estimates: across all axes, diversity scores converge smoothly with increasing n. Fluctuations at n < 50 shrink substantially by 50 images and become negligible after 100. Between 150-250 images, confidence intervals are extremely narrow (Appendix Fig. 18). Model and country rankings remain consistent: GeoDE maintains the highest diversity scores and FLUX the lowest across all four axes and for all n ≥ 50 (Appendix Fig. 19). The metric is therefore statistically well-behaved and convergent; 95% confidence intervals decrease monotonically with sample size, indicating predictable, low uncertainty.
> >
> > **Consistency across seeds and image batches:** Running the full pipeline three times on the same 250 SD3m-generated Indian house images yields a maximum standard deviation of 0.01 across axes (Entity: 0.009, Background: 0.001, Affluence: 0.013, Maintenance: 0.006, overall: 0.007). Generating three independent SD3m batches using different seeds produces a maximum deviation of only 0.05 (Entity: 0.018, Background: 0.044, Affluence: 0.023, Maintenance: 0.008, overall: 0.023).
> >
> > **Robustness to changing prompts:** We vary prompts in two ways. First, we perturb SEVI prompts via GPT and observe negligible changes in average affluence and maintenance scores (Appendix Table 14). Second, we modify the image-generation prompt. Our original template is “a photo of a {entity} in {country}.” We test three variants for USA, Colombia, India, and Egypt across three entities (house, chair, stove) and two models (SD2.1, FLUX.1):
> >
> > * Variant 1: “an image of a {entity} in {country}”
> > * Variant 2: “a {entity} in {country}”
> > * Variant 3: “a {entity} located in {country}”
> >
> > As with our original prompt, Variant 1 and Variant 2 remain neutral; the only difference is that the generated image need not be a photo and may be a drawing or cartoon. Variant 3 uses more sophisticated wording (“located in” instead of “in”), potentially preconditioning the models. Across all four axes, SD2.1 shows high country-rank consistency with the original prompt, indicating its diversity scores are largely insensitive to prompt changes (ρ = 0.80, 0.85, 0.80 for variants 1–3). FLUX.1 is more sensitive, with correlations of ρ = 0.65, 0.80, 0.45, still significantly high yet indicative of model-specific prompt sensitivity. Appendix J.4 further discusses our observations regarding FLUX.1’s lower correlations.
> >
> > **While the proposed SEVI and VDI are interpretable metrics, they lack statistical grounding.**
> >
> > Both SEVI and VDI are grounded in a well-established statistical framework: they use a normalized Hill Number formulation (Eq. 1), a principled family of diversity indices widely used in ecology, information theory, and diversity analysis [a, b]. Calculating the Shannon entropy within Eq. 1 requires access to the VLM-estimated answer distribution for each dimension, yielding an interpretable estimate of the effective number of distinct attribute values represented by the model. Accurate estimation of this distribution is essential, and our extensive robustness analyses, across evaluator models, prompt perturbations, image budgets, random seeds, and independent batches, consistently show that these empirical distributions are stable, convergent, and statistically well-behaved. The resulting diversity scores exhibit smooth convergence with sample size, narrow confidence intervals for n ≥ 100, and highly consistent model and country rankings.
> >
> > [a] Leinster, Tom. "Entropy and diversity: The axiomatic approach." Cambridge university press, 2021
> >
> > [b] Friedman, Dan, and Adji Bousso Dieng. "The vendi score: A diversity evaluation metric for machine learning." Transactions on Machine Learning Research, 2023.

---

> > > ### Author Response · Authors · 2025-11-25
> > >
> > > Dear Reviewer oi9r,
> > >
> > > With only a week remaining for the discussion phase, we kindly invite you to review our rebuttal and share any further feedback or questions you may have.
> > >
> > > We look forward to the discussion!
> > >
> > > Thank you,
> > >
> > > Authors

---

> ### Comment · Reviewer_oi9r · 2025-11-26
> **Thank you and a follow up**
>
> The rebuttal addresses my robustness concerns, and also clarifies that  the “goal is to measure within-model diversity”.  I have a couple of follow up questions on this goal.
>
> - There is knowledge that the generative model intrinsically knows about a given concept, and there is knowledge one can observe/extract. The latter has to do with sampling. IID sampling may not obtain samples from weak modes in the learned distribution, they are primarily samples from the strong modes. Many generative models have more knowledge than can be extracted through IID sampling. For example, see [R1,R2]. In this light, the observations in the paper may not be measuring the intrinsic within-model diversity but instead measuring what one can “observe through IID sampling”. The methodology followed in the paper is not consistent with the stated goal. The higher sensitivity observed when varying prompts, perhaps points to sampling from different modes.
>
> - If the underlying diversity in the world for a concept is low, what is the rationale for expecting the generative model to show high diversity? Do we expect the generative model to reflect the world or not? And, without comparing to the real-world, how can one determine if the diversity exhibited by the model is reflective of the world? In short, the conclusion that a generative models has “high” or “low” diversity for a given concept is not informative by itself without knowing what the real-world has.
>
> [R1] Particle guidance: non-iid diverse sampling with diffusion models, ICLR 2024
>
> [R2] DiverseFlow: Sample-Efficient Diverse Mode Coverage in Flows, CVPR 2025

---

> ### Author Response · Authors · 2025-11-27
>
> We thank the reviewer for the response and are glad to see that most of their concerns, including those on robustness, have been addressed. Below, we answer the follow-up questions about the principles of our work.
>
> **There is knowledge that the generative model intrinsically knows about a given concept, and there is knowledge one can observe/extract...**
>
> Thank you for highlighting this distinction. Our goal is to measure the geo-diversity of the images generated by a model under the most neutral prompts, ‘a photo of a {concept} in {country}’, which corresponds to the model’s default sampling behavior. This setting aligns with prior works on geographical diversity [1, 2], and follows the notion of “default-mode diversity” as described in [3]. We apologize for not clearly naming this setting earlier; in our previous comment, “within-model diversity” refers strictly to this default sampling regime, and request to note that this phrase has not been used in the manuscript.
>
> While alternative sampling strategies may uncover additional modes, these constitute interventions into the model’s sampling procedure rather than its default behavior. Our framework intentionally operates in the default setting to evaluate what the model generates with prompts without explicit attribute details and without such sampling interventions. In this standard regime, GeoDiv reveals biases through attribute distributions (e.g., lack of paved roads in Nigeria for SD3.5 or overwhelming poverty cues for India in SD2.1). As shown in Appendix G, these observations can then be leveraged to design targeted prompts that elicit more diverse and balanced generations.
>
> [1] Hall et al. 'DIG In: Evaluating Disparities in Image Generations with Indicators for Geographic Diversity', TMLR 2023
>
> [2] Hemmat, Reyhane Askari, et al. "Improving geo-diversity of generated images with contextualized vendi score guidance." ECCV 2024
>
> [3] Teotia, Revant, et al. "DIMCIM: A Quantitative Evaluation Framework for Default-mode Diversity and Generalization in Text-to-Image Generative Models." ICCV 2025.
>
> **If the underlying diversity in the world for a concept is low, what is the rationale for expecting the generative model to show high diversity? ...**
>
> Diversity can be defined in multiple ways, as noted in prior work [4]. If one aims to compare with real-world diversity, this would require (a) the true attribute distribution for each entity or (b) a reference dataset with broad country-level coverage. As noted in our earlier comment, obtaining (a) is infeasible and (b) is limited, since datasets like GeoDE [6] cover only a small subset of countries and entities. In this setting, *GeoDiv rather defines diversity from the fairness perspective, by measuring the effective percentage of attribute values a model generates for each country-entity pair, giving equal opportunity of representation to each attribute value* (consistent with OASIS, which defaults to a uniform distribution when real-world data is unavailable (Section A.1.3 [5])). It reveals clear differences across models (e.g., richer-looking images for FLUX.1, higher visual diversity in SD2.1) and across countries. For instance, the over-depiction of poverty in Nigeria in SD2.1 ignores the economic diversity within the country. From a fairness perspective, such repeated generation of only one type of appearance (e.g., “dusty road”, “broken wall”) is undesirable: it perpetuates stereotypes by failing to represent the broader plausible range, **independent of the exact real-world frequencies**. Hence, instead of collapsing to a dominant pattern, fair generative models should aim for a broader range of plausible variations. Section 6 further contextualizes this by showing that real images from the geo-diverse GeoDE dataset [6] exhibit substantially higher diversity than the synthetic ones (see Figure 1 for an example): Entity-Appearance Diversity 0.60 vs. 0.44; Background-Appearance Diversity 0.42 vs. 0.31; and similar trends for SEVI. This consistent gap confirms that synthetic generations are indeed less diverse. Thus, even without access to the real-world distribution, low diversity simply indicates that the models tend to generate narrow sets of attribute values for a concept across countries, and the persistent gap with GeoDE shows that the low diversity scores arise from the model’s default behavior.
>
> [4] Leinster, Tom. "Entropy and diversity: The axiomatic approach." Cambridge university press, 2021
>
> [5] Dehdashtian, Sepehr, Gautam Sreekumar, and Vishnu Naresh Boddeti. "Oasis uncovers: High-quality t2i models, same old stereotypes.", ICLR 2025
>
> [6] Ramaswamy et al., ‘GeoDE: a Geographically Diverse Evaluation Dataset for Object Recognition’, NeurIPS D&B 2023
>
> We hope we have been able to address the reviewer’s concerns. We would be happy to answer further questions if any.

---

> ### Author Response · Authors · 2025-12-02
> **Discussion Summary**
>
> We thank the reviewer once again for the constructive feedback. We appreciate the recognition of the importance and lack of research in assessing geo-diversity in generative models and the value of our large-scale benchmark. We summarize our discussions below for ease of the AC’s review.
>
> Changes made to manuscript (in red color):
>   * Experiments for further validating GeoDiv through Robustness Analysis: Section J.4 of the Appendix has the following new experiments:
>   * Robustness to Minimally Semantically modified Image generation Prompts
>   * Robustness to VQA Prompt Variations
>   * Robustness to varying Image Budgets
>   * Robustness to VQA re-runs and Different seed image sets
>
> Other clarifications made:
>   * We clarified that we measure geographical diversity of generated images (not stereotypes), with identified biases emerging only from analysis.
>   * Described steps we take to reduce model-internal biases.
>   * They also questioned whether we should attempt to generate weaker modes and whether comparison to real-world attribute distributions is necessary. We clarified that our framework intentionally evaluates default model behavior without sampling interventions, and that because real-world distributions for all entities/attributes are unavailable, we adopt a fairness-based definition that gives equal opportunity to each attribute value

---

### Author Response · Authors · 2025-11-20
**General Response to All Reviewers**

We sincerely thank all reviewers for their constructive feedback and thoughtful insights. We are delighted that reviewers find the problem of *auditing geo-diversity important* (**oi9r, oxKU, UsfU, WAhk**), especially given that *it has received limited attention apart from a few research publications and media articles* (**oi9r**), and that they view *our contribution as novel* (**oxKU, UsfU**), *advancing beyond traditional demographic or visual diversity metrics* (**WAhk**). Reviewers also appreciate the *nuanced definitions of diversity* (**UsfU**) introduced in the paper and the *rigorous validation of the framework* (**oi9r, WAhk, oxKU, UsfU**). Several reviewers further praised the *specific biases uncovered by our framework* and *the clarity of the writing* (**oxKU, UsfU**), as well as recognized the *large-scale benchmarking performed on 160,000 images* (**oi9r, WAhk**).

In response to the thoughtful comments we received, we have revised the paper draft with color-coded changes for each reviewer (**oi9r: red**, **WAhk: orange**, **oxKU: blue**, **UsfU: purple**) to facilitate readability. We believe the reviewers’ suggestions have significantly strengthened the work, and we will continue refining the manuscript throughout the discussion phase.

Additionally, we summarize the primary additions made to our manuscript addressing the reviewer’s comments:
1. (Reviewer **oi9r**) Experiments for further validating GeoDiv through Robustness Analysis: Section J.4 of the Appendix has the following new experiments:
    - Robustness to Minimally Semantically modified Image generation Prompts
    - Robustness to VQA Prompt Variations
    - Robustness to varying Image Budgets
    - Robustness to VQA re-runs and Different seed image sets
2. (Reviewer **WAhk**): The referred papers have been included in the Related Work section. The notations of $\mathcal{A}_k$ have been revised to denote the LLM generated approximate answers and their corresponding distributions (Section 3.3). Hyperparameter details of the LLM-VLM have been updated in Appendix H.1.
3. (Reviewer **oxKU**): We have restructured the Introduction, including Figure 1, to first exhibit that geographical bias exists, followed by presenting GeoDiv as a reliable method to quantify this. Similar changes can be observed in Section 3, with the addition of subsection 3.3 to define the diversity metric.
4. (Reviewer **UsfU**): A note on the number of entities and countries studied is added in the Limitation section, along with the challenge of being dependent on LLMs for attribute and their value generations. Details of Inter-annotator agreement have been added to Appendix J.2

---

### Author Response · Authors · 2025-12-02
**Discussion Summary**

Dear AC,

We are thankful for your coordination of the review process and briefly summarize our discussions with the reviewers below. The initial scores were *8 (UsfU)*, *6 (oxKU)*, *2 (WAhk)*, and *4 (oi9r)*, with *oxKU* updating theirs to *8* after discussion. Reviewers highlighted several strengths in our paper, and agreed that auditing geo-diversity is important (oi9r, oxKU, UsfU, WAhk) and found our contribution novel (oxKU, UsfU), advancing beyond traditional demographic or visual diversity metrics (WAhk). They further appreciated our nuanced diversity definitions (UsfU), rigorous validation (oi9r, WAhk, oxKU, UsfU), identification of specific biases, clarity of writing (oxKU, UsfU), and the scale of our 160K-image benchmark (oi9r, WAhk). We summarize the discussions with the individual reviewers below:

**Reviewer UsfU (Score 8)**: We incorporated clarifications and improvements to our manuscript based on our discussion with the reviewer who maintained their positive assessment and expressed their support for the good work (*Discussion concluded on 22 Nov*).

**Reviewer oxKU (Score 6 -> Raised to 8)**: Substantial revisions were made to the manuscript based on the discussions. The reviewer confirmed that all concerns were resolved and explicitly highlighted the novelty of the contribution (*Discussion concluded on 22 Nov*).

**Reviewer WAhk (Score 2)**: The reviewer’s concerns centered on novelty and dependence on closed evaluator models. The novelty concern, raised solely by reviewer WAhk, was based on usage of LLM/VLM in our framework. However, we clarify that we are the first to systematically decompose geographical diversity into visual and socio-economic axes, and to use LLM/VLM world-knowledge to quantify them, which is a substantive advance over prior demographic-bias works. In our rebuttal, we addressed these in detail, including clarifying a misunderstanding about Qwen2.5-VL (an open model, rather than a closed one as claimed by the reviewer). The reviewer **did not participate in the discussion phase**, but our rebuttal comprehensively addresses all novelty related queries and the questions raised.

**Reviewer oi9r (Score 4)**: We engaged in a detailed exchange and resolved most concerns (among other updates, we added a section on Robustness Analysis of our metric to the manuscript), as acknowledged by the reviewer. Two conceptual questions were also conclusively addressed in a follow-up, though the reviewer **did not respond again before Nov 28**.

All these updates have been incorporated in the revised manuscript. Detailed summary for each reviewer has been added to the individual reviewer discussion threads for ease of the AC’s review. We hope this effectively summarizes the discussion phase and the updates made to strengthen the manuscript. We greatly appreciate your time and consideration.

---

### Meta-Review · Area_Chair_LsHq · 2026-01-07

**Summary:**

Paper presents an approach for measuring geographical diversity in Text-to-Image models. Initially the paper received split recommendations of: 1 x marginally below the acceptance threshold, 1 x marginally above the acceptance threshold, 1 x reject and 1 x accept. Concerns centered mainly on the following:

(1) Confusion between bias and stereotypes [oi9r]
(2) Heavy use of LLM/VLM to both define and measure attributes, which in itself could be biased [oi9r, UsfU]
(3) Lacking statistical grounding for proposed SEVI and VDI metrics [oi9r]
(4) Lacking novelty [WAhk]
(5) Lacking exposition, which should more clearly delineate the problem from the proposed solution [oxKU]
(6) Lack of reporting fot Inter-Annotator Agreement for human validation [UsfU]

Authors have provided a rebuttal and revisions to the paper that have addressed many of these concerns. Specifically, [oi9r], [UsfU] and [oxKU] have all knowledge that their concerns were addressed, with [oxKU] commenting that the score would be raised from 6 to 8. Given the generally positive disposition of reviewers toward the paper and rebuttal (and lack of participation from WAhk), AC is recommending Acceptance.

**Reviewer Concerns:**

(1) Confusion between bias and stereotypes [oi9r]
(2) Heavy use of LLM/VLM to both define and measure attributes, which in itself could be biased [oi9r, UsfU]
(3) Lacking statistical grounding for proposed SEVI and VDI metrics [oi9r]
(4) Lacking novelty [WAhk]
(5) Lacking exposition, which should more clearly delineate the problem from the proposed solution [oxKU]
(6) Lack of reporting fot Inter-Annotator Agreement for human validation [UsfU]

**Reviewer Scores:**

Three out of four reviewers acknowledge that their concerns have been addressed and one commits to raising the score, putting the paper in favorable position for acceptance. I believe [WAhk] may have also raised a score with deliberation. Remaining two reviewers are likely to stay at their ratings.

---

### Decision · Program_Chairs · 2026-01-26

Accept (Poster)